# Surface-layer turbulence, energy-balance and links to atmospheric circulations over a mountain glacier in the French Alps.

Maxime Litt[1,2], Jean-Emmanuel Sicart[3], Delphine Six[3], Patrick Wagnon[4], and Warren D. Helgason[5]

[1]Université Grenoble Alpes, LTHE, F-38000 Grenoble, France
[2]ICIMOD, GPO Box 3226, Kathmandu, Nepal
[3]CNRS, LGGE, F-38000 Grenoble, France
[4]IRD/Université Grenoble Alpes/CNRS/G-INP, LTHE UMR 5564, Grenoble, France
[5]Civil and Geological Engineering, University of Saskatchewan, 57 Campus Drive, Saskatoon S7N 5A9, Saskatchewan, Canada

*Correspondence to:* M.Litt (maximelitt@gmail.com)

**Abstract.** Over Saint-Sorlin Glacier in the French Alps (45° N, 6.1° E, $\sim$ 3 km$^2$) in summer, we study the atmospheric surface-layer dynamics, turbulent fluxes, their uncertainties, and their impact on surface energy balance (SEB) melt estimates. Results are classified with regard to large-scale forcing. We use high frequency Eddy-Covariance data and mean air-temperature and wind-speed vertical profiles, collected in 2006 and 2009 in the glacier atmospheric surface layer. We evaluate the turbulent fluxes with the Eddy-Covariance (sonic) and the profile method, and random errors and parametric uncertainties are evaluated by including different stability corrections and assuming different values for surface roughness lengths. For weak synoptic forcing, local thermal effects dominate the wind circulation. On the glacier, weak katabatic flows with a wind-speed maximum at low height (2-3 m) are detected 71% of the time and are generally associated with small turbulent kinetic energy (TKE) and small net turbulent fluxes. Radiative fluxes dominate the SEB. When the large-scale forcing is strong, the wind in the valley aligns with the glacier flow, intense downslope flows are observed, no wind-speed maximum is visible below 5 m, TKE and net turbulent fluxes are often intense. The net turbulent fluxes contribute significantly to the SEB. The surface layer turbulence production is probably not at equilibrium with dissipation because of interactions of large-scale orographic disturbances with the flow when the forcing is strong, or low-frequency oscillations of the katabatic flow when the forcing is weak. In weak forcing when TKE is low, all turbulent fluxes calculation methods provide similar fluxes. In strong forcing when TKE is large, the choice of roughness lengths impacts strongly the net turbulent fluxes from the profile method fluxes and their uncertainties. Though, the uncertainty on the total SEB remains too high with regard to the net observed melt to be able to recommend a turbulent fluxes calculation method against another.

## 1 Introduction

Climate change might affect the hydrological regimes in glacierized mountainous catchments (Viviroli et al., 2011). There is a strong need to understand the links between large-scale atmospheric changes and glacier melt processes. The melt response of glaciers to climate change is generally assessed using calibrated temperature-index models of various complexities (Hock,

2003; Huss et al., 2010; Pellicciotti et al., 2012). Though these models perform correctly when meteorological conditions do not differ much with regard to the calibration period conditions, under changing climate, the frequency of occurrence of specific atmospheric circulation patterns may change (Corti et al., 1999), affecting relationships between temperature and melt. The physical link between glacier melt and climate is the surface energy balance (SEB), embodying all the heat exchanges occurring near the glacier surface (Oke, 1987). Surface energy balance studies under contrasted climatic conditions are required to validate temperature-index calibrations, and adapt them to changing melt processes in a changing climate.

Turbulent energy transfer can play a significant role in the SEB of glaciers (e.g. Sicart et al., 2008; Six et al., 2009), especially at high altitudes and low latitudes, where sublimation can significantly reduce the energy available for melt. Turbulent fluxes remain poorly quantified, since they are usually evaluated using parameterizations such as the profile method (often in its bulk form), based on similarity assumptions which state fluxes are constant with height in the atmospheric surface layer and that turbulent mixing scales with mean vertical gradients of wind speed, temperature and humidity near the surface. These assumptions are frequently violated over mountain glaciers (Smeets et al., 1999, 2000; Denby and Greuell, 2000; Litt et al., 2015b). Vertical flux divergence and non-stationarity of the flow (Denby and Greuell, 2000; McNider, 1982) occur under katabatic winds, a common feature of air flows above glaciers (Poulos and Zhong, 2008). Non-stationarity of the flow can be induced by oscillations of the katabatic forcing, or the interaction of outer-layer turbulent structures with the surface layer. These structures might be generated when synoptic forcing is strong and the large-scale atmospheric flow interacts with the complex orography (Smeets et al., 1999, 2000). Unmet similarity assumptions may lead to systematic biases in the flux evaluation with the BA method (Mahrt, 2008; Litt et al., 2015a).

The impact of biased turbulent heat fluxes estimates on SEB calculations has been documented for vegetation canopies (Gellens-Meulenberghs, 2005; Stoy et al., 2013), but few studies have dealt with glacier or snow surfaces (e.g. Helgason and Pomeroy, 2012; Conway and Cullen, 2013). Although numerical simulations have shown the profile method was reliable in estimating the surface fluxes in the presence of flux divergence below a wind-speed maximum (Denby and Greuell, 2000), the effects of katabatic oscillations and outer-layer interactions remain poorly documented over mountain glaciers (Smeets et al., 1999, 2000). Evaluating the accuracy of the profile method for different large-scale forcing is necessary to understand the impact of the similarity assumptions on SEB calculations over mountain glaciers, and thus, on the evaluation of glacier melt rates from meteorological measurements or from climate data.

This study on the Saint-Sorlin glacier (French Alps, Grandes Rousses massif, 45° N, 6.1° E) compares turbulent flux measurements with observed large-scale weather patterns over Europe. We investigated the impact of turbulent flux measurement errors on glacier SEB calculations, for different prevailing atmospheric circulation patterns. The study covers two melt periods during the summers of 2006 and 2009, for which dominant atmospheric conditions were contrasted. Atmospheric circulation patterns were characterized with a daily time series of weather patterns derived from the analysis of Garavaglia et al. (2010). Surface-layer flow and turbulence data were obtained from two field campaigns deployed in the ablation area of the glacier. We used data from a 6-m mast measuring vertical profiles of wind speed and temperature in 2009, a high frequency Eddy-Covariance (EC) system in 2006, and radiation, temperature, wind speed and humidity measured by automatic weather stations during both campaigns. We characterized the turbulence in the surface-layer over the glacier and calculated the tur-

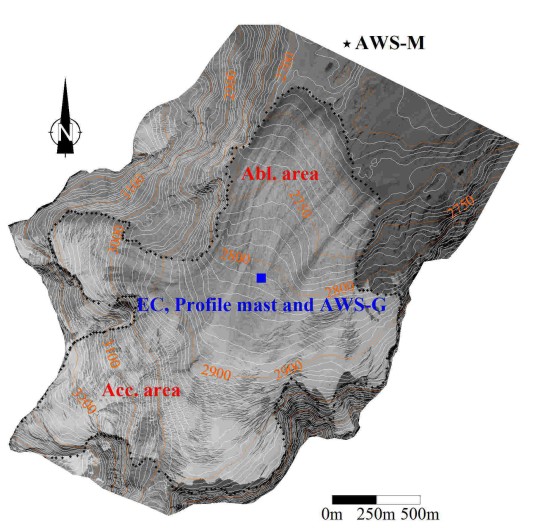

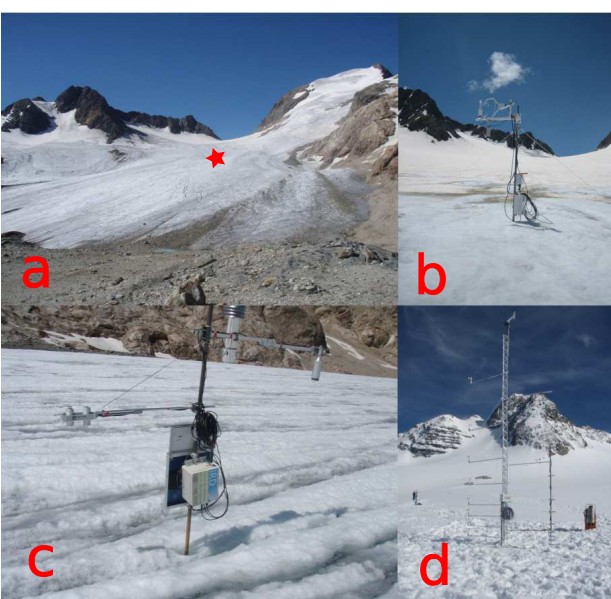

**Figure 1.** Overview of the glacier and the instruments installed during the field campaigns. (left) Topographic map of the glacier and position of the instruments. (a) Picture of the Saint-Sorlin glacier taken from the AWS-M (heading to the top of the glacier, south is at the horizon), (b-c-d) the instruments installed during the 2006 and 2009 campaigns.

bulent fluxes with both the profile method and the EC data (sonic method). We used thsese fluxes together with the measured radiative fluxes to compute the SEB. We discuss the difference on the melt as derived from the SEB calculations resulting from the use of different methods to derive the turbulent fluxes, for the two melt periods.

## 2   Site and data

### 2.1   Saint-Sorlin glacier

Saint-Sorlin Glacier is a small mountain glacier (surface area $\simeq 3$ km$^2$) which altitude ranges between 2700 and 3400 m a.s.l (Fig. 1). It is located in the western part of the French Alps in the Grandes-Rousses massif (Fig. 2). It is characterized by a large flat ablation area (slope $\simeq 5°$) between 2700 and 2900 m a.s.l. Some steep slopes ($> 35°$) are found higher. Ice flows from the South-South-West to the North-North-East, inside a valley which has the same orientation. This valley is bordered by a steep ridge on its western flank and is opened on its eastern flank. The annual mass balance of the glacier has been monitored since 1957 (Vincent et al., 2000; Six and Vincent, 2014). All mass-balance data are available at http://www-lgge.obs.ujf-grenoble.fr/ServiceObs/index.htm.

Table 1. Characteristics of sensors from the automatic weather stations (AWS-G and AWS-M), the eddy-covariance (EC) and the profile mast. We present the random errors on the measurements, as provided by the manufacturer and the random measurement errors that we used to quantify uncertainties on the turbulent fluxes. Minimum and maximum heights refer to the lowest measurement levels in the case of the profile data.

| Quantity | Instrument | Precision according to the manufacturer | Accuracy used in this study | Sensor Height (m) | | | |
| --- | --- | --- | --- | --- | --- | --- | --- |
| | | | | Mean | $\sigma$ | min | max |
| **Profile Mast (2009 campaign)** | | | | | | | |
| Aspirated air temperature, °C | Type-T thermocouple | 0.1 °C | 0.1 °C | 0.74, 0.84, 1.04, 1.24, 1.54, 1.84, 2.44, 3.14, 4.64 | 0.30 | 0.20 | 1.50 |
| Wind speed, m s$^{-1}$ | Vector A100L2 | 1% of reading | 0.1 m s$^{-1}$ | 0.83, 1.43, 2.33, 3.03, 4.03 | 0.30 | 0.35 | 1.70 |
| **Eddy Covariance System (2006 campaign)** | | | | | | | |
| High frequency wind speed components, m s$^{-1}$ | Campbell C-SAT3 | $w: \pm 0.040$ m s$^{-1}$ $u, v: \pm 0.015$ m s$^{-1}$ | $< \pm 0.04$ m s$^{-1}$ $\pm 0.015$ m s$^{-1}$ | 2.37 | 0.20 | 2.10 | 2.95 |
| High frequency sonic temperature, °C | Campbell C-SAT3 | 0.025°C | 0.025°C | 2.37 | " | " | " |
| High frequency specific humidity, % | LICOR7500 | 2% of reading | 2% of reading | 2.37 | " | " | " |
| **Automatic Weather Stations (2006 and 2009 campaigns)** | | | | | | | |
| Aspirated air temperature, °C and relative humidity, % | Vaisala HMP45C | $\pm 0.2$ °C 3% | $\pm 0.2$ °C 3% | 2.30 | 0.25 | 1.80 | 2.70 |
| Wind speed, m s$^{-1}$ | Young 05103 | 0.3 m s$^{-1}$ | 0.3 m s$^{-1}$ | 2.50 | 0.20 | 2.00 | 2.95 |
| Wind direction, deg | Young 05103 | $\pm 3°$ | $\pm 3°$ | " | " | " | " |
| Incident and reflected shortwave radiation, Wm$^{-2}$ | Kipp and Zonen CM3 | 10% on daily sums | 0.4% 0.4% | 1.20 | 0.20 | 0.80 | 2.20 |
| Incoming and outgoing longwave radiation, Wm$^{-2}$ | Kipp and Zonen CG3 | 10% on daily sums | 0.4% 0.4% | 1.20 | 0.20 | 0.80 | 2.20 |
| Surface elevation changes, m | Campbell SR50 | $\pm 0.01$ m | $\pm 0.1$ m | 1.80 | 0.20 | 1.20 | 2.45 |

## 2.2 Field campaigns

We used data from two field campaigns focused on the ablation area of the glacier at around 2800 m a.s.l (Fig. 1). The first campaign was undertaken between 9 July and 28 August 2006, featuring a tower-mounted EC system measuring the three wind-speed components, air temperature and specific humidity at 20 Hz. The second campaign was staged between 13 June and 04

September 2009. We installed 5 cup anemometers and 9 ventilated thermocouples on a 6-m mast, measuring mean temperature and wind-speed profiles. During both campaigns there was an adjacent automatic weather station (AWS-G) consisting of instruments measuring mean aspirated air temperature and humidity, mean wind speed and direction, incoming and outgoing shortwave and longwave radiation fluxes. All masts were fixed into the ice. Height changes due to melt or snow deposition were measured with a sonic height ranger. Due to high melt rates ($\simeq 6$ cm day$^{-1}$), the height of the sensors changed with time. Instruments were manually lowered every 10 to 15 days. A picture of the instruments installed on the glacier is shown in Fig. 1. We also used data from an automatic weather station located outside the glacier (AWS-M), on a nearby moraine north of the site ( 1.5 Km), measuring the same variables as the AWS-G. This station is operated in the framework of the GLACIOCLIM program (les GLACIers, un Observatoire du Climat) that undergoes a follow-up of glaciers mass balance and meteorology in order to understand glacier fluctuations in terms of climatic variations. Table 1 summarizes all the instruments used and their characteristics.

Glacier surface presented similar characteristics during both campaigns. At the beginning, the glacier surface was covered with old winter or spring snow and was smooth (Fig. 1b and d). The glacier surface was irregular at the end of the campaigns due to strong melt that caused the appearance of gullies (Fig. 1a and c). Height changes of about 20 to 30 cm were observed on horizontal scales of about 2-3 m (not shown). The glacier surface characteristics remained homogeneous in every directions on hundreds of meters from the measuring site.

## 2.3 Data processing

Data have been processed in a similar way than in Litt et al. (2015b). All data were split into 1-h runs, since the AWS-M provides only 1-h means. High frequency data from the EC system were checked for quality (Vickers and Mahrt, 1997). Problems were mainly related to precipitation events or frost on the sensor heads. Bad quality runs were discarded (30%). Remaining good-quality runs (GQR) were despiked (Vickers and Mahrt, 1997). A planar-fit rotation (Wilckzak et al., 2001) was applied on each 10 to 15 day period between field visits when they were manually lowered. From this rotation we derived the horizontal ($u$ and $v$) and vertical ($w$) wind-speed components. The sonic temperature was corrected for water vapour influences (Schotanus et al., 1983). The changes in instrument heights were taken into account using measurements from the sonic height ranger and regular field visits and controls.

## 3 Methods

### 3.1 Characterization of large-scale forcing

We characterized the large-scale forcing and the frequency of typical atmospheric circulation patterns during the campaigns using a daily time series of weather patterns (WP). We used the results of the Garavaglia et al. (2010) WP analysis which is based on the study of the shapes of the precipitation fields measured in the south-east of France, between 1956 and 1996. The shapes of the measured precipitation fields, at the daily time scale, are classified into 8 different classes. For each class, a mean

observed geopotential height field over Europe is computed. Then, for any day (inside or outside the period used to characterize the decomposition, e.g. 1956-1996) the observed geopotential height field shape over Europe is characterized by the observed geopotential height at 0h and 24h, at the 700 hPa level and the 1000 hPa level for 110 grid points (a total of 440 points). This field is compared, in this 440 dimensions space, to the 8 geopotential height fields proposed by the rain-patterns analysis. The nearest of the 8 fields provides the WP. Details of the procedure can be found in Garavaglia et al. (2010).

A WP classification cannot be separated from the object it aims to characterize. An ideal WP decomposition for surface-layer turbulence studies would be based on surface energy flux-related variables such as ground-based measurements of temperature, wind speed or humidity rather than on precipitation. However, the Garavaglia et al. (2010) decomposition is useful here for an exploratory study, in which we use the WP to identify the direction and strength of the large-scale flows, and to obtain a frequency of typical large-scale circulation patterns during each campaign. Large-scale flow direction and strength is related to the displacement of low-pressure systems and thus to the shape of geopotential height fields. Since each WP is associated with a mean geopotential height field shape in the analysis of Garavaglia et al. (2010), it provides a simple and convenient tool for this study. An approximative direction of the atmospheric flow at low level above the ground can be associated with each WP (Fig. 2).

We grouped together the WP for which the wind conditions on the glacier were expected to be similar and we conserved three subsets. Low-pressure systems coming from the West, the South west and South can cause strong winds aligned with the valley and the glacier flow (Fig. 2). These conditions are related to Atlantic waves, South-west and South Circulations and Central Depressions. We grouped together the corresponding WP (patterns 1, 3, 4 and 7, Garavaglia et al., 2010) in a Strong Forcing (SF) subset. Large-scale flows associated with WP1 are not well aligned with the glacier (Fig. 2). Nevertheless for this WP we observe strong downslope flows above the glacier. We refer to weak large-scale flows, related to the presence of high-pressure systems (WP8) as Weak Forcing (WF). The WP 2, 5 and 6 are associated with Steady Oceanic Circulations, East Returns and North-east Circulations, respectively, and are grouped together as Other Forcing (OF). They present very distinct characteristics in terms of large-scale flow directions. They are generally associated with precipitation events or freezing, low wind speeds and erratic wind directions on the glacier, thus only a small fraction of the corresponding runs are classified as good-quality runs (Table 2) and we do not analyze them in detail herein.

## 3.2 Surface turbulence characterization

To study the intensity of the turbulence in relation with the WPs, we computed the mean TKE using the EC data (overlines indicate temporal averages over a 1-h run and primes denote fluctuations around this mean):

$$\overline{\text{TKE}} = \frac{1}{2} \left( \overline{u'^2} + \overline{v'^2} + \overline{w'^2} \right), \tag{1}$$

with $u$, $v$ and $w$ the horizontal and vertical wind speed components, respectively.

The profile method, commonly used for the determination of surface turbulent heat fluxes relies, on the assumptions of Monin-Obukhov Similarity Theory (herein, MOST, Monin and Obukhov, 1954). It assumes that the production of turbulent kinetic energy (TKE) in the stable surface layer, which is defined as the layer of air above the surface for which the turbulent

fluxes do not change by more than 10% of their surface values (Stull, 1988), is related to the mean local shear. The production of turbulence must be balanced by buoyancy and by destruction by viscous dissipation. No turbulent energy must be transported from the outer layers towards the surface layer and no advection shall transfer turbulence from some other parts of the field (Monin and Obukhov, 1954). We used spectral analysis to assess the validity of MOST and identify potential surface-layer disturbing features, for the SF and WF forcing cases. We studied the Fourier spectra ($S$) of the wind-speed components and the cospectra ($Co$) of $u$ with $w$ (associated with turbulent momentum flux) and of $w$ with the potential temperature, $\theta$ (associated with sensible heat), and with the specific humidity, $q$ (associated with latent heat). Following Smeets et al. (1999), individual spectra were normalized by the variance of $w$ which led to a collapse of individual curves at high frequency. Heat flux-related cospectra were normalized by the kinematic flux ($\overline{w'\theta'}$ or $\overline{w'q'}$ ). We did not normalize $Co_{uw}$ since $\overline{u'w'}$ is expected to be small and erratic near a wind-speed maximum as observed in katabatic flows. Medians of spectral powers over subsets of runs were calculated on equally spaced logarithmic intervals.

We assumed that in an equilibrium surface layer, turbulent spectra and cospectra would compare well with the Kaimal curves (Kaimal et al., 1972) measured under ideal conditions. Deviations of the spectra from classic theoretical predictions would indicate that flux-profile relationships established in idealized circumstances are not appropriate in this environment. Deviations from these curves are expected if the surface layer is disturbed by large, outer-layer eddies. In complex terrain, several studies have shown that atmospheric flow interactions with the orography can generate such features, that they may reach the surface and enhance turbulent mixing. In such a case, mixing may not scale with local flow characteristics (Andreas, 1987; Smeets et al., 1999; Poulos et al., 2007; Helgason and Pomeroy, 2012). Near the surface, these eddies would be horizontally deformed due to its blocking effect and to the large shear in the surface layer (Högström et al., 2002): They would disturb the surface layer by inducing large horizontal velocity fluctuations, while vertical velocity would be less influenced. They would provoke an enhancement of the spectra and cospectra at lower frequency than the maximum of the Kaimal curves. This enhancement should be stronger on horizontal wind components than on vertical wind components (Högström et al., 2002). Katabatic wind oscillations would also be reflected in the spectra of the horizontal wind components by deviations at low frequency.

## 3.3 Turbulent fluxes

### 3.3.1 The EC and the profile method

High-frequency measurements were undertaken at low height ($\sim 2$ m), supposedly inside the surface layer. The turbulent fluxes of sensible ($H$) and latent heat ($LE$) can be written as:

$$H = -\rho C_p \overline{w'\theta'}_s = -\rho C_p u_* T_*, \tag{2}$$

$$LE = -\rho L_e \overline{w'q'}_s = -\rho L_e u_* q_*, \tag{3}$$

where fluxes directed towards the surface and towards the atmosphere are defined as positive and negative, respectively. The symbols $u_*$, $\theta_*$ and $q_*$ stand for turbulent velocity, temperature and humidity surface-layer scales. The subscript $s$ refers to

surface values, $\rho$ is the air density ($kg\ m^{-3}$), $C_p$ is the specific heat of humid air (1003.5 $J\ kg^{-1}\ K^{-1}$), $L_e$ is the latent heat of sublimation of the ice ($2.83 \times 10^6\ J\ kg^{-1}$) when the surface is below freezing and the latent heat of vaporization ($2.50 \times 10^6$ $J\ kg^{-1}$) when the surface is melting and $k$ is the von-Karman constant (0.4). To apply Equations 2 and 3 we used alternately the sonic and the profile methods (subscripts son and pro, respectively). The sonic method relies on direct measurements of

the fluctuating quantities $u$, $\theta$ and $q$ (based on the EC data), at some height above the surface, assuming fluxes are constant between the surface and this level.

The profile method relies on the MOST. In these context, the profile method, in its bulk version, estimates $u_*$, $\theta_*$ and $q_*$ using finite differences of mean measurements between the sensors and the surface, assuming that the fluxes scale with the mean vertical gradients:

$$u_{*_b} = k\frac{\overline{u}}{\left(\ln\left(\frac{z}{z_0}\right) - \psi_m\left(\frac{z}{L_*}\right)\right)} \tag{4}$$

$$\theta_{*_b} = k\frac{(\overline{\theta} - \overline{\theta}_s)}{\left(\ln\left(\frac{z}{z_t}\right) - \psi_t\left(\frac{z}{L_*}\right)\right)} \tag{5}$$

$$q_{*b} = k\frac{(\overline{q} - \overline{q}_s)}{\left(\ln\left(\frac{z}{z_q}\right) - \psi_q\left(\frac{z}{L_*}\right)\right)} \tag{6}$$

The $\psi_{m,h,q}$ are the stability corrections for ($m$) momentum, ($h$) temperature and ($q$) humidity, respectively. The dimensionless $z/L_*$ is a scaling parameter in the surface layer. When evaluated with the profile method, this parameter documents the stability of the layer of air between the surface and the sensor assuming TKE production scales only on the mean local vertical gradients of wind speed and temperature. The length scale $L_*$ is the Obukhov length defined as:

$$L_* = -\frac{\overline{\theta}u_*^2}{kg\theta_*} \tag{7}$$

The length scales $z_0$, $z_t$ and $z_q$ are the dynamical, thermal and humidity roughness lengths (m), respectively. Theoretically, roughness lengths can be directly related to the physical roughness of the terrain, assuming that turbulence is mainly generated by interaction of the surface-layer flow with the surface, which is generally valid above a stable surface layer under the MOST assumptions. Rugged surfaces would present higher roughness than smooth snow covered areas. One can also derive the roughness lengths by solving Equations 4, 5 and 6. In many studies roughness lengths are not known and in order for the SEB to

match the measured ablation, turbulent fluxes are adjusted using an effective roughness length $z_{\text{eff}}$ such as $z_0 = z_t = z_q = z_{\text{eff}}$ (Wagnon et al., 2003; Favier et al., 2004; Cullen et al., 2007; Six et al., 2009). According to Equations 4,5 and 6, we shall have:

$$z_{\text{eff}} = \frac{z}{exp(\sqrt{\ln z/z_0 \ln z/z_{t,q}})}; \tag{8}$$

For this study, roughness lengths were evaluated by least square iterative fitting of the temperature and wind speed vertical

profiles measured in 2009, and assuming $z_q = z_t$. Since the aspect of the surface was similar in 2006 and 2009, we used the

same values for the roughness lengths for both field campaigns. The method for roughness length determination was inspired from Andreas (2002) and developed for the tropical Zongo glacier. It is detailed in (Sicart et al., 2014). The median values are $z_0 = 10^{-3}$ m and $z_t = 10^{-5}$ m. This leads to $z_{\text{eff}} = 10^{-3.98}$ m assuming $z = 2.5$ m. The results show a large scatter, about four orders of magnitude, that can be attributed to poor accuracy of the temperature measurements leading to large random uncertainties (Sicart et al., 2014). The scatter was too large to observe significant changes in the measured roughness lengths during the 2009 campaign, in spite of snow falls or snow melt that uncovered the ice surface, or appearance of small gullies of about 0.1-0.3 m height variations on a few meters horizontal scale that could also have impacted the roughness lengths (Smeets and Van den Broeke, 2008a, b). We also evaluated the roughness length using the EC system and inverting equations 4 and 5, and selecting neutral runs. The median $z_0$ was 0.022 m, and the results for $z_t$ was $6.6 \times 10^{-6}$ m. This leads to a $z_{\text{eff}} = 10^{-3.08}$ m.

### 3.3.2 Flux calculations

The evaluation of Equations 4, 5 and 6 with the profile method requires an iterative scheme. A first evaluation of $u_{*\text{pro}}$, $\theta_{*\text{pro}}$ and $q_{*\text{pro}}$ was obtained assuming neutral stability ($z/L_{*\text{pro}} = 0$). These estimates were used to compute a value of $z/L_{*\text{pro}}$, which was used to calculate a new estimation of $u_{*\text{pro}}$, $\theta_{*\text{pro}}$ and $q_{*\text{pro}}$, and so on, until convergence was reached.

The profile formulation for the turbulent fluxes includes several variables and parameters. For reference, we first used the simple Stefan-Boltzmann law for surface temperature, assuming a surface emissivity of 0.99. Melt was observed most of the time, so that surface temperature was generally 0°C. The value of $q_s$ was derived from the surface temperature assuming air at the surface was saturated. We used log-linear stability functions (Businger et al. (1971) and Dyer (1974)). We used two sets of roughness length values: first, profile-derived roughness lengths from the 2009 campaign. We identify these fluxes with the pro subscript. The second set is calculated with an effective roughness parameter, $z_{\text{eff}}$, set to 0.001 m, calibrated in Six et al. (2009) so that the SEB matches the observed melt during the 2006 campaign. These fluxes are identified with a eff subscript.

We then performed a Monte-Carlo calculation, to assess the variability of all the other parameters (inspired by Giesen et al., 2008). Stability corrections $\psi_{m,h,q}$, for ($m$) momentum, ($h$) temperature and ($q$) humidity, respectively, were taken successively from Brutsaert (1982), Holtslag and de Bruin (1988) and Beljaars and Holtslag (1991). Temperature of the surface was derived from the outgoing longwave radiation measured at the AWS-G, using the Stefan-Boltzmann law and assuming an emissivity of 0.99 or 0.95 for the ice or snow. A correction factor of 0.6% of $SW_{i}nc$ was either applied, or not, on the $LW_{inc}$ value when assessing the surface temperature.

### 3.3.3 Errors on turbulent fluxes

We evaluated the random errors on all the resulting turbulent fluxes following the methods that Litt et al. (2015a) applied on the Zongo glacier. For the profile method, an analytical calculation was performed, propagating the uncertainties in the measurements of the meteorological variables (Table 1) through Equations 2 and 3. The large dispersion in the observed roughness lengths was taken into account in this calculation as $\delta \ln z_{0,t,q} \simeq 2.5$ (Sect. 3.3), calculated following Sicart et al. (2014). We assumed the errors on the measurements of individual variables were independent and that stability functions did

not change much under small variations of the measurements. Surface temperature error depends on the error on $LW_{out}$ which was assumed to be $0.4\%$: this corresponds to the observed dispersion of the night time measurements of $SW_{in}$ and day time melt conditions measurements of $LW_{out}$. This method provides much realistic errors on surface temperature ($\sim 0.35°K$) than when using the nominal error on radiation of 10% (leading to errors of 6-7°$K$, Litt et al., 2015a). The largest source of random error was the uncertainty on the roughness lengths, which were not precisely defined.

For the EC method, we followed Litt et al. (2015a), assuming EC measurement uncertainties on wind speed and temperature were negligible and that most random errors were due to insufficient statistical sampling of the largest eddies (Vickers and Mahrt, 1997). We applied the Mann and Lenschow (1994) method which relates the random error to the time scale $\tau$ of the largest eddies of the flow. This timescale was derived from the peaks of the cospectra of $w$ with $\theta$ and $q$ obtained by Fourier analysis (Wyngaard, 1973).

## 3.4 Surface energy balance and melt

The SEB was calculated as:

$$\text{SEB} = \sum \left( SW_{inc} - SW_{out} + LW_{inc} - LW_{out} + H + LE \right) \tag{9}$$

All fluxes are expressed in W m$^{-2}$. The symbols $SW$ and $LW$ stand for hourly mean shortwave and longwave radiation, respectively. The subscripts $inc$ and $out$ stand for incoming and outgoing radiation, respectively. Sub-surface conductive heat flux was neglected because the surface remained close to melting day and night ($T_s > -3°$C). A simulation of the conduction flux below the snow or ice surface, using the surface temperature as the only input data, showed that this flux was negligible, even when surface temperature fell below zero during short periods of a few hours. The rare events of strong surface cooling, with surface temperature reaching $\sim -11°$C during the night, were associated with precipitation events or freezing for which meteorological data quality was low due to frost or snow deposition on the sensors. These periods were discarded by the quality check procedure. The energy gains from precipitation were excluded from the analysis as they were assumed to contribute negligibly to the surface energy balance (Paterson and Cuffey, 1994; Oerlemans, 2001), and since we removed the rain and snow events period from our analysis.

Turbulent fluxes were obtained alternately from the profile method (with profile-derived or effective roughnesses) or the sonic methods. Ablation due to evaporation or sublimation at the surface was considered negligible: mean absolute latent heat fluxes remained below 10 Wm$^{-2}$ for the most turbulent subsets (Table 3), which corresponds to a daily ablation of only 0.3 mm w.e. We assumed the error on the measured daily melt was of the order of the SR50 error, i.e. $\pm 0.01$ m, which results in a 40 Wm$^{-2}$ error on the mean hourly melt energy. The error on the turbulent fluxes was translated into melt errors. Uncertainties on the net shortwave radiation were estimated by calculating the potential solar radiation following Pellicciotti et al. (2011). We first assumed no inclination and then a maximum inclination of the sensor of $10°$ in any direction. The maximum relative difference between the two values was taken as the relative net solar radiation error.

The SEB was converted to melt, assuming the ice density was 900 kg m$^{-3}$. At the beginning of the campaigns, old snow was present at the surface, with measured density of 480 kg m$^{-3}$. The precipitated water equivalent during the occasional snowfalls

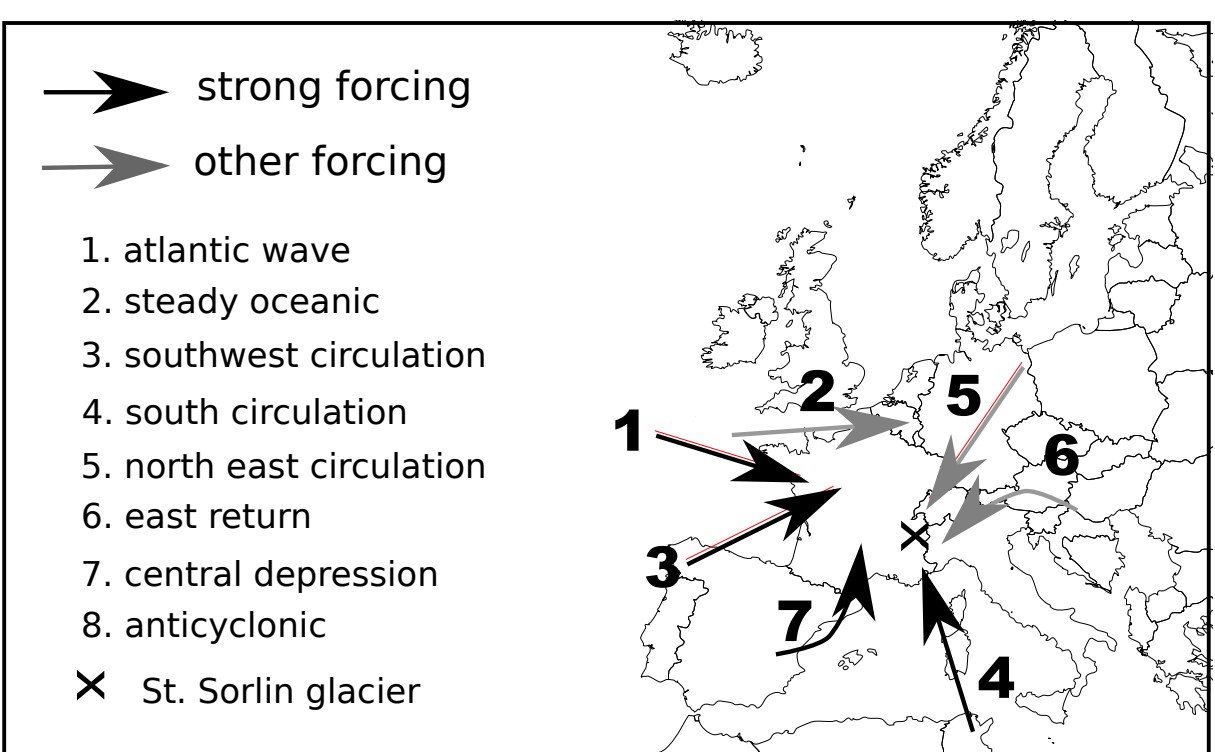

**Figure 2.** Direction of the atmospheric flow in the lower atmospheric layers for different WP (adapted from Garavaglia et al., 2010). (black arrows) Direction of the flow for the WP associated with SF and (grey arrows) direction of the flow for the other WP. No flow direction is shown for WF conditions (WP8) since atmospheric wind speed was weak and direction ill-defined for these conditions.

on the glacier were obtained from SAFRAN reanalysis data (Durand et al., 1993). The results were compared to the melt ($\text{Melt}_i$) derived from the SR50 surface height changes for continuous periods of more than 6 hours, when no measurements were discarded and characteristics of the surface layer turbulence remained similar:

$$\text{Melt} = (h_{end} - h_{start}) \times \rho_s L_f \tag{10}$$

5   Where ($h_{end,start}$) indicate the SR50 measurements at the beginning and at the end of the selected continuous periods, $\rho_s$ is the mean density of the underlying snow or ice surface ($\text{kg m}^{-3}$), $L_f$ is the latent heat of fusion of ice ($3.34 \times 10^5 \text{ J kg}^{-1}$).

## 4   Results

### 4.1   Meteorology and wind regimes

We describe the meteorological conditions of each campaign in terms of frequency of SF and WF conditions (Fig. 3). We
10   observed two types of conditions associated with WF weather patterns. One was characterized with high mean temperatures

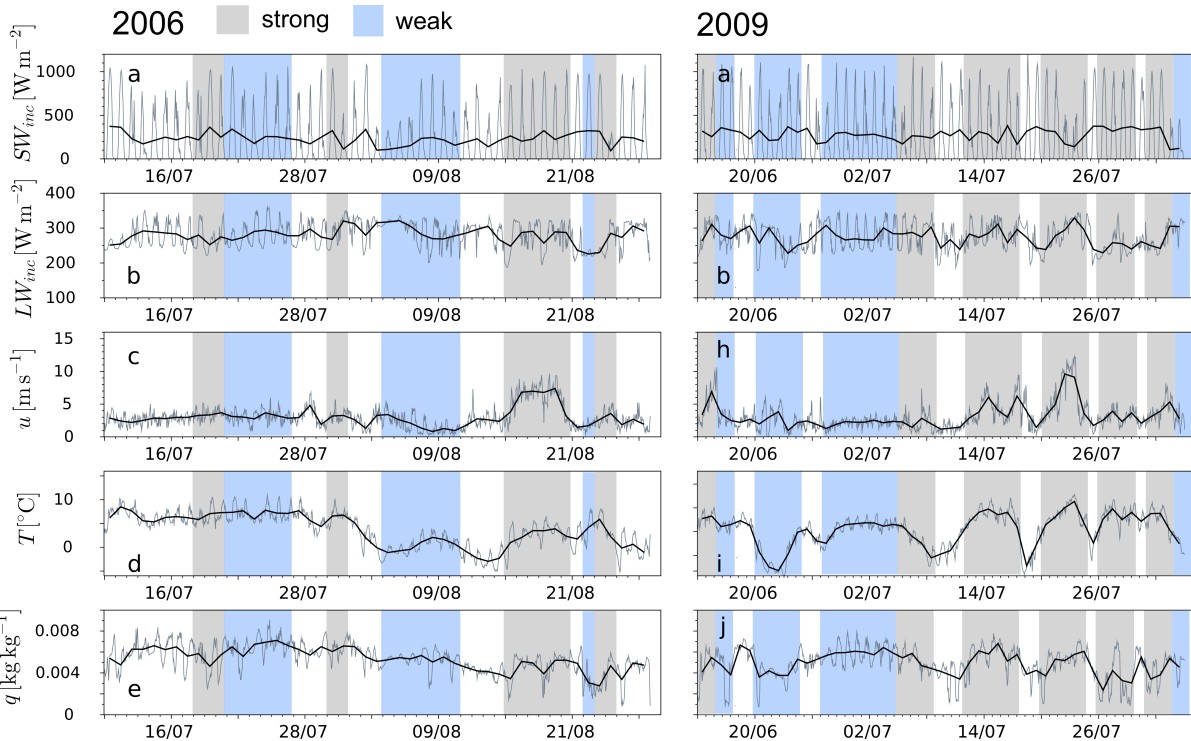

**Figure 3.** Change of the meteorological variables in the ablation zone of the glacier at the AWS-G in (left) 2006 and (right) 2009. (a,f) incoming shortwave radiation, (b,g) incoming longwave radiation, (c,h) wind speed, (d,i) air temperature and (e,j) specific humidity of the air. SF conditions are dark shaded and WF conditions are blue shaded. The dark lines in lower panels (c, d, e, h. i and j) show daily averages.

($> 5°C$), clear morning skies ($SW_{inc} > 500$ W m$^{-2}$) with clouds developing in the afternoon ($LW_{inc}$>300 W m$^{-2}$), moderate wind speed (3 to 5 m s$^{-1}$) and moist air (Table 2), as observed during the long WF event at the beginning of the 2006 campaign. The other type of conditions associated with WF was characterized by low temperatures (0-5 °C), covered sky, occasional snow falls (not shown), low wind speed (2-3 m s$^{-1}$) and dryer air. This was observed between 20 June and 25 June 2009. The WF

5    patterns are associated with high-pressure systems and to weak synoptic forcing. As a result the valley circulation seemed controlled by local thermal effects. On the glacier, wind blew mostly downslope, day and night (Fig. 4). Upslope winds were observed only 5% of the time. Over the glacier, the local katabatic forcing was dominant: a weak wind-speed maximum was observed at low heights ($\sim 2$ m s$^{-1}$ at $\sim 2$ m, Fig. 5) for $\sim 70\%$ of the 2009 WF situations (Table 2). During the day, convection probably drove the circulation in the valley: outside the glacier, the wind blew down the valley during the night and up the

10    valley during the day (Fig. 4). The TKE was generally small in these situations (Table 2). Mean sensible heat fluxes ($H_{\mathrm{son}}$) were moderately positive, they were slightly underestimated with $H_{\mathrm{pro}}$ and $H_{\mathrm{eff}}$. Latent heat fluxes ($LE_{\mathrm{son}}$) were slightly positive and $LE_{\mathrm{pro}}$ and $LE_{\mathrm{eff}}$ fit well with $LE_{\mathrm{son}}$.

Conditions of SF were characterized by high wind speeds (peaks above $> 7$ m s$^{-1}$) and moderate on-glacier air temperatures ($\sim 5°C$). Such conditions were observed around mid-august in 2006 and mid-July in 2009. SF conditions are related to low-

**Table 2.** Characteristics of the two field campaigns in terms of WP frequencies, meteorological data and turbulence characteristics: Fraction of runs recorded under each group of WPs. Fraction of good-quality runs for the EC data in 2006, inside each group, and fraction of runs, inside each group during the 2009 campaign, for which a wind-speed maximum was observed with the profile mast. Mean meteorological variables recorded by the AWS-M, mean TKE in each weather group , fraction of run in SF and WF conditions for which TKE is $> 2\ \mathrm{ms^{-2}}$ or $< 1\ \mathrm{ms^{-2}}$, and mean turbulent fluxes obtained from the different methods.

| | Strong Forcing | | Weak forcing | |
|---|---|---|---|---|
| | 2006 | 2009 | 2006 | 2009 |
| Time coverage | 26% | 46% | 47% | 33% |
| Fraction of good-quality runs (EC) | 32% | - | 51% | |
| Fraction of good-quality runs with a detected wind-speed maximum | - | 31% | - | 71% |
| Fraction of runs with $\overline{\mathrm{TKE}} < 1\ \mathrm{ms^{-2}}$ | 28% | | 68% | |
| Fraction of runs with $\overline{\mathrm{TKE}} > 2\ \mathrm{ms^{-2}}$ | 52% | | 11% | |
| $u_{AWS-M}$ [m s$^{-1}$] | 5.2 | 5.5 | 2.5 | 2.6 |
| $T_{AWS-M}$ [°C] | 7.2 | 9.1 | 7.6 | 5.3 |
| $q_{AWS-M}$ [$10^{-3}$ kg kg$^{-1}$] | 4.9 | 4.8 | 5.8 | 4.9 |
| TKE [m$^{-2}$ s$^{-2}$] | 2.6 | - | 1.0 | - |
| $H_{\mathrm{son}}$ | $45 \pm 2$ | - | $24 \pm 1$ | - |
| $H_{\mathrm{pro}}$ | $36 \pm 1$ | $47 \pm 1$ | $18 \pm 1$ | $9 \pm 1$ |
| $H_{\mathrm{eff}}$ | $50 \pm 2$ | $66 \pm 3$ | $20 \pm 1$ | $7 \pm 1$ |
| $LE_{\mathrm{son}}$ | $-9 \pm 3$ | - | $6 \pm 1$ | - |
| $LE_{\mathrm{pro}}$ | $3 \pm 1$ | $3 \pm 1$ | $6 \pm 1$ | $-1 \pm 1$ |
| $LE_{\mathrm{eff}}$ | $-6 \pm 2$ | $4 \pm 1$ | $7 \pm 1$ | $-3 \pm 1$ |

pressure atmospheric system associated with West, South-west or South circulations, roughly aligned with the glacier flow and with the valley. As a result, in the vicinity of the glacier, these conditions were associated with the largest wind speeds, and they seemed to trigger a dominant down-valley circulation, overwhelming convective circulations: on the glacier, wind was blowing downslope 99% of the time (Fig. 4). Day and night, the wind direction was similar over the glacier and outside the glacier.

5  The median wind-speed profiles were nearly logarithmic below 2 m, and the temperature inversion was marked (Fig. 5). A katabatic wind-speed maximum was not frequently observed at low heights (Table 2 and Fig. 5), either because it was located above the mast, or hidden in the background flow, or non-existent. The TKE was generally high (Table 2) for these conditions. Mean sensible heat fluxes ($H_{\mathrm{son}}$) were highly positive, they were underestimated with $H_{\mathrm{pro}}$ and slightly overestimated with $H_{\mathrm{eff}}$, but closer to $H_{\mathrm{son}}$. Latent heat fluxes ($LE_{\mathrm{son}}$) were slightly negative, $LE_{\mathrm{pro}}$ was very low, positive, and $LE_{\mathrm{eff}}$ fit well

10  with $LE_{\mathrm{son}}$.

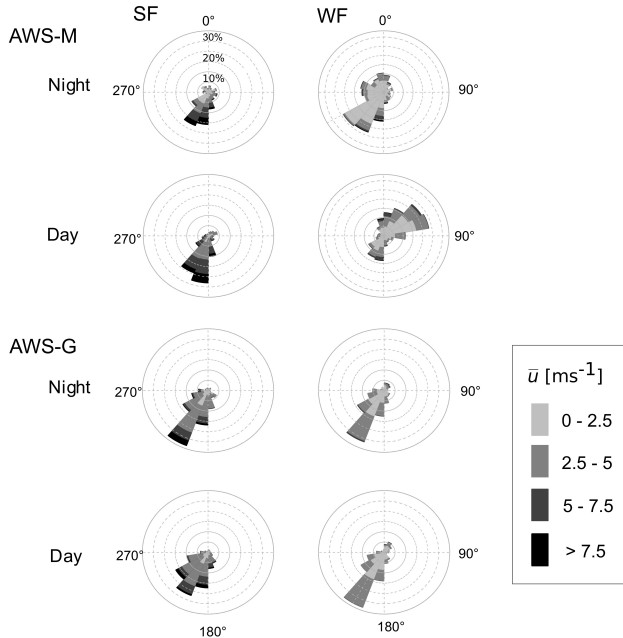

**Figure 4.** Rose diagrams of wind direction and speed measured by the AWS-G and the AWS-M for SF(left panels) and WF (right panels) conditions during the night (upper panels) and during the day (lower panels) .

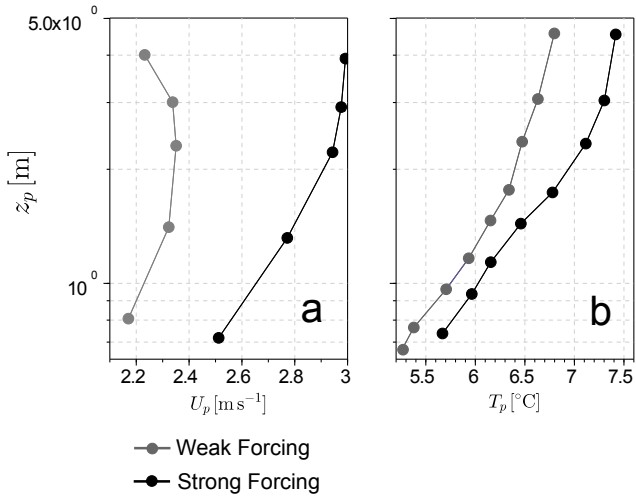

**Figure 5.** (a) Median wind-speed profiles and (b) temperature profiles during the 2009 field campaign for (gray) WF conditions and (black) SF conditions.

## 4.2 Turbulent characteristics of the surface flow

### 4.2.1 Turbulent kinetic energy

For WF and SF conditions, TKE generally was low and high, respectively (Table 2). However, some high (11%, $\overline{\mathrm{TKE}} > 2\mathrm{m}^2\mathrm{s}^{-2}$) and low (28%, $\overline{\mathrm{TKE}} < 1\mathrm{m}^2\mathrm{s}^{-2}$) TKE cases remained in the WF and SF classes, respectively (Table 2). Thus, we studied turbulent fluxes only for high or low TKE subsets of the SF or WF groups, respectively. During SF conditions, we analyzed only the most frequent runs for which wind blew downslope (direction between 90° and 270°, 85 % of the GQR), keeping only cases for which $H_{\mathrm{son}} > 5\mathrm{Wm}^{-2}$, $\overline{u} > 2\mathrm{ms}^{-1}$ and $\overline{\mathrm{TKE}} > 2\mathrm{m}^2\mathrm{s}^{-2}$ in order to filter for cases when turbulence was sufficiently developed. During WF conditions, we kept similar requirements except that we selected only runs where $\overline{\mathrm{TKE}} < 1$ $\mathrm{m}^2\,\mathrm{s}^{-2}$. Since we observed some discrepancies between RH obtained with the EC and the AWS-G (not shown), we also removed runs for which RH differed by more than 10%. This classification distinguishes typical turbulent conditions and groups together spectra and cospectra of similar shapes (Sect. 4.2.2).

### 4.2.2 Spectral analysis

The Fourier analysis of the 2006 EC wind speed components and temperature data was compared to the Kaimal et al. (1972) curves, to assess if the surface layer was in equilibrium (Sect. 3.3). Wind speed spectra in Fig. 6a and 6b show that the horizontal and the vertical wind speed oscillated at low frequency, more than expected for an equilibrium surface layer. For $n < 10^{-1}$ the horizontal and vertical wind-speed spectra were higher than the prediction of Kaimal et al. (1972). The amplitude of these low-frequency oscillations was much larger for the horizontal than for the vertical wind speed. Similar observations were made for large and low TKE runs, except that the peak in $S_u$ was observed at $n \sim 5 \times 10^{-2}$ and $n \sim 5 \times 10^{-3}$ when $\overline{\mathrm{TKE}}$ was high and low, respectively.

The low-frequency oscillations of $u$ and $w$ affected the momentum flux as shown by the $Co_{uw}$ cospectra (Fig. 6d). For high $\overline{\mathrm{TKE}}$ and $n > 10^{-2}$ the median cospectrum of $w$ with $u$ was negative (Fig. 6d). For $n < 10^{-2}$ the median cospectrum reversed its sign, a peak was observed at the same frequency as the low-frequency peak of $S_u$ ($n \sim 5 \times 10^{-2}$), and the dispersion between the individual cospectra was large. When $\overline{\mathrm{TKE}}$ was low, $Co_{uw}$ was low too. For $n > 10^{-2}$ the median cospectrum was slightly positive, indicating an upward momentum flux, which could be explained by the presence of a katabatic wind-speed maximum just below the sensor, roughly at 2 m above the ground (Fig. 5). For $n < 10^{-2}$ the median cospectra was low, but the individual cospectra were dispersed around zero (hardly visible in Fig. 6d).

Low-frequency oscillations also influenced the sensible (Fig. 6c) and the latent heat fluxes (not shown). When $\overline{\mathrm{TKE}}$ was high, all the individual cospectrum collapsed to a single curve and the dispersion was low. The median cospectrum exhibited a small plateau at the frequency of the peak in the Kaimal curve. A peak was observed above the Kaimal curve at $n \sim 5 \times 10^{-1}$. For $n < 5 \times 10^{-1}$, the median cospectrum fell to zero. When $\overline{\mathrm{TKE}}$ was low, the dispersion between the individual cospectra was high. The median cospectrum peaked at the frequency of the peak in the Kaimal curve ($n \sim 5 \times 10^{-1}$). For $n < 5 \times 10^{-1}$ the median cospectra fell to zero erratically. At the lowest frequency end, individual cospectra were dispersed around zero.

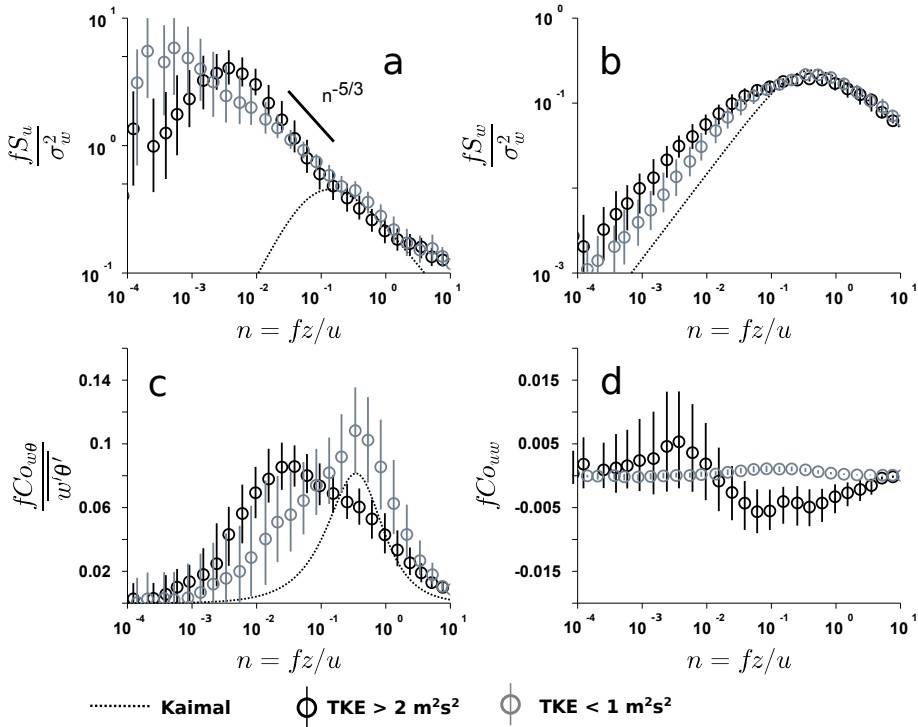

**Figure 6.** Fourier analysis of the high frequency EC data during the 2006 campaign. The median spectra and cospectra calculated over (black) the Strong forcing and high TKE and (grey) the weak forcing and low TKE subsets are presented together with (black dotted) the Kaimal et al. (1972) curve. (circles) average of the spectra or cospectra over a bin of normalized frequency and (vertical bars) interquartile range of the spectra or cospectra over the same bin. (a) Spectra of horizontal wind speed $u$, (b) spectra of $w$, (c) cospectra of $w$ with $\theta$ and (d) cospectra of $w$ with $u$.

### 4.2.3 Turbulent fluxes

Weak forcing, small TKE runs were characterized by moderate and positive sensible heat fluxes $H_{\mathrm{son}}$ (Table 3 and Fig. 7). The latent heat fluxes $LE_{\mathrm{son}}$ were slightly positive (Table 3 and Fig. 7), and net fluxes were a moderate gain of energy at the surface (Fig. 8). Mean individual fluxes from each method were comparable, except $H_{\mathrm{eff}}$. Sensible heat fluxes $H_{\mathrm{pro,eff}}$ were poorly related to the measures and latent heat fluxes $LE_{\mathrm{pro,eff}}$ showed a better relation with sonic values (Table 3, Fig. 7). Even though calculating the fluxes with effective roughnesses provided net higher fluxes on average, the different methods provided comparable results with regard to the dispersion obtained with the Monte-Carlo approach Fig. 8). Using the effective roughness length led to a larger dispersion in terms in the correlation plots (Table 3, Fig. 7) but also in the Monte-Carlo runs (Fig.8).

For strong forcing, high TKE runs, $H$ was highly positive, and $LE$ was moderately negative on average, net fluxes were a larger gain of energy for the surface than for weak forcing, low TKE runs (Table 3 and Fig. 7). The $H_{\mathrm{pro}}$ and $LE_{\mathrm{pro}}$ underestimated the magnitude of $H\mathrm{son}$, $LE_{\mathrm{son}}$ and their sum on average (Table 3 and Fig. 3). Relations between measured

**Table 3.** Characteristics of the two field campaigns in terms of turbulent fluxes measured with the EC (sonic) and modeled with the profile methods with different roughness parameters (pro, eff). Correlation, coefficients as well as normalized root mean square deviations (RMSD) from the least-square fit between each profile method fluxes and sonic fluxes are presented.

| Method | WF with $\overline{\text{TKE}} < 1$ ms$^{-2}$ | | SF with $\overline{\text{TKE}} > 2$ ms$^{-2}$ | |
|---|---|---|---|---|
| Time coverage | 9% | | 5% | |
| Fraction of good- | | | | |
| quality runs (EC) | 12% | | 7% | |
| $u_{AWS-M}$ [m s$^{-1}$] | 8.8 | | 1.9 | |
| $T_{AWS-M}$ [$^\circ$C] | 6.2 | | 10.3 | |
| $q_{AWS-M}$ [$10^{-3}$ kg kg$^{-1}$] | 1.6 | | 3.4 | |
| TKE [m$^{-2}$ s$^{-2}$] | 4.7 | | 0.44 | |
| $H_{\text{son}}$ | 25±1 | RMSE | 77±2 | RMSE |
| $H_{\text{pro}}(z_{0,t,q})$ | 28±1 | 0.4 | 48±2 | 0.6 |
| $H_{\text{eff}}$ | 34±1 | 0.6 | 72±4 | 0.3 |
| $LE_{\text{son}}$ | 10±1 | | -20±3 | |
| $LE_{\text{pro}}(z_{0,t,q})$ | 8±1 | 0.9 | -14±1 | -2.2 |
| $LE_{\text{eff}}$ | 10±1 | 0.8 | -22±2 | -1.9 |

and modeled fluxes were significant ($r > 0.68$), but they were more coherent when using effective roughnesses: in that case, regression coefficients increased significantly. Monte-Carlo-derived errors were too low to explain the discrepancy between ($H + LE_{\text{pro}}$) and the sonic results. Using the effective roughness improved the concordance but the dispersion increased dramatically.

## 5  Discussion

### 5.1  Turbulence in the surface layer.

For SF forcing when the TKE was high ($> 2\text{m}^2\text{s}^{-2}$, 52% of SF conditions), the characteristics of the wind-speed spectra were similar to that of a surface layer influenced by outer-layer turbulence: the variance of $u$ was strongly enhanced at low frequency compared to what is expected for an equilibrium surface layer, while a limited increase was observed in the $w$ spectra, a combination of observations which agrees with the parameterizations of Högström et al. (2002) for surface layers disturbed by large eddies: in the vicinity of the surface, if large-scale structures are transported from the outer-layer, they are supposedly elongated horizontally due to the surface blocking, and due to the large shear in the surface layer. As a result, they must induce large horizontal fluctuations of the horizontal wind speed while the vertical wind speed must be less affected. Also, a $n^{-5/3}$ slope was observed on the low frequency part of the $u$ spectra when TKE was large (Fig. 6a). This signature resembles that of an outer, well-mixed layer (Højstrup, 1982). Similar spectral shapes were observed over other mountain glaciers when

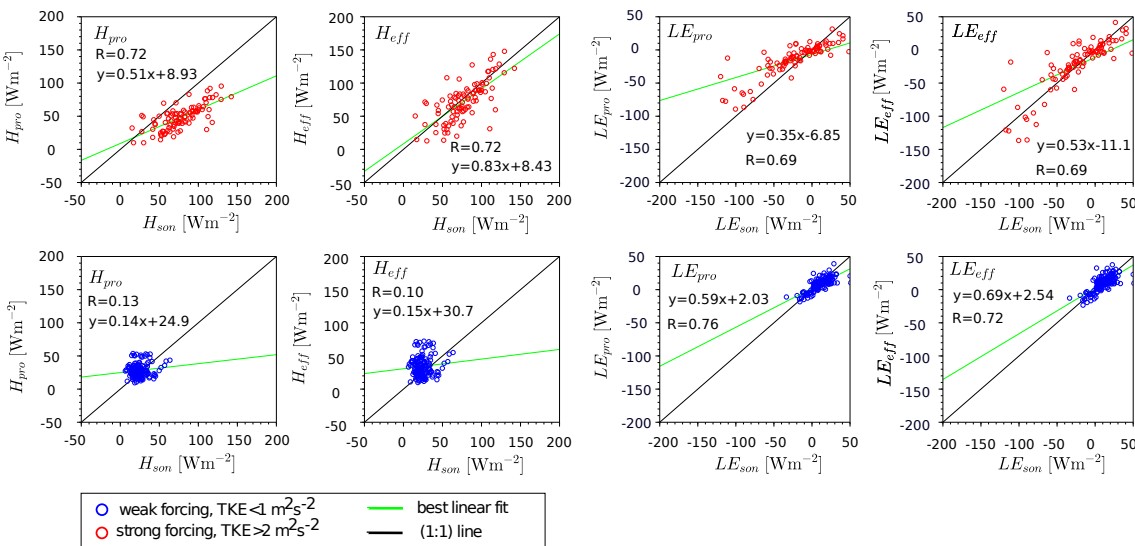

**Figure 7.** Comparison of hourly reference $H, LE_{\text{pro}}$ and $H, LE_{\text{eff}}$ fluxes from the profile method with the fluxes $H, LE_{\text{son}}$ measured with the EC, during the 2006 field campaign. Red and blue circles fluxes represent results during the SF cases when TKE was high and WF cases when TKE was low, respectively. Black lines are the 1:1 lines and green lines are the regressions. Equations for these lines and correlation coefficients are also presented.

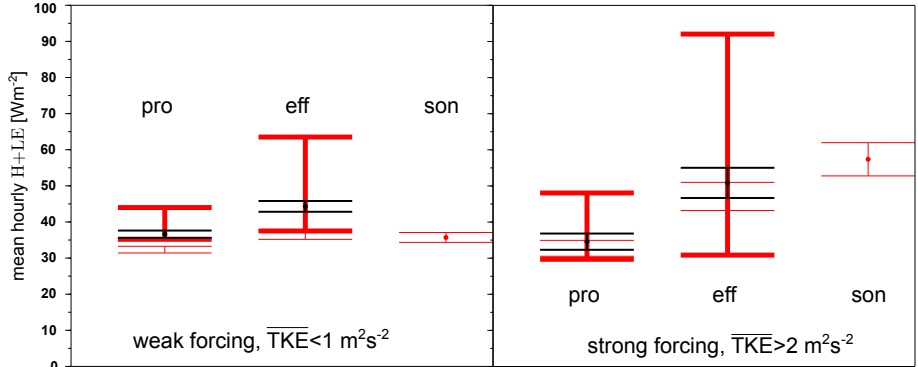

**Figure 8.** Comparison of mean net turbulent fluxes from the profile method and the sonic method. For the profile method we show the reference fluxes and their random errors in black, in thick red the maximum fluxes and in thin red minimum fluxes (with their random error) as derived from the Monte-Carlo analysis, and the net sonic fluxes to the right in thin red.

synoptic forcing was strong (Smeets et al., 1999; Litt et al., 2015b). In these studies, the observed low-frequency oscillations were attributed to the interaction of large scale coherent structures with the surface-layer turbulence. Such structures can be generated when the large-scale flow interacts with the complex mountain orography. In such situations, Smeets et al. (1999) showed that the TKE transport term was not negligible. Significant TKE can be transported from the outer layer and the surface

layer turbulence was not in equilibrium with local production. We hypothesize that such structures were influencing the surface flow on Saint-Sorlin glacier. These perturbations would have had a strong effect on the momentum flux, and would explain the shape of the cospectra of $u$ with $w$ for strong forcing, high TKE cases (Fig. 6d), which exhibited erratic contributions around zero at low frequencies. They would have affected also the turbulent heat fluxes (Fig. 6c), explaining why the cospectra

of $w$ with $\theta$ peaked and contained more energy at low frequency than the Kaimal predictions. Assuming our hypothesis was valid, these perturbations would have likely originated from various turbulent structures sharing similar time scales, since the low frequency peak was observed systematically at $n \sim 10^{-2}$, only in strong forcing, high TKE cases. The influence of these structures on the fluxes would have always be of the same sign and induced larger heat fluxes magnitudes than predicted by the Kaimal curve.

Two reasons may explain why the profile method systematically underestimated the fluxes in these conditions (Table 3), why Monte-Carlo dispersion was too low to explain the observed difference between the fluxes derived from each method, and why using the effective roughness method provided larger fluxes, that compared better to the sonic fluxes than using the profile-derived roughnesses (Table 3). First, it might be that the roughness lenghts derived during the 2009 campaign did not represent well the surface conditions of the 2006 campaign as former studies have shown that roughnesses can change by several order

of magnitudes in a course of a season. Second, it could be that since the cospectra is issued from the sonic data, the sonic method includes the fluxes above the Kaimal curve but that on the contrary the profile method must provide only the fluxes predicted by the Kaimal curve and cannot take into account the turbulent mixing that is not due to local shear production or buoyancy effects, and as a result underestimates the intensity of the heat exchanges. This suggests that the use in SEB models of an effective roughness length, larger than the profile-derived dynamic or thermal roughness lengths, respectively, in order

to increase the turbulent fluxes so that the SEB matches the melt, could actually be a way to compensate potential biases in the profile-derived turbulent fluxes due to failure of the MOST when TKE is high in SF. This is supported by the values of the sonic derived roughness lengths, considering they account for the additional mixing: the denominator $(\ln(z/z_0)\ln(z/z_t))$ in the bulk formulation of the fluxes, calculated with the EC derived roughness length or the effective roughness length is roughly identical (0.0165 and 0.0167).

During WF cases when the TKE was small ($< 1\text{m}^2\text{s}^{-2}$, 68% of WF conditions), low-frequency oscillations of the wind speed were common (Fig. 6a). These cases would have favored the development of katabatic flows, as shown by the frequent detection of a katabatic maximum in WF conditions of 2009, which lead to vertical flux divergence and to a deconnection of turbulence characteristics below and above the wind-speed maximum height. The surface layer would thus be limited to a small fraction of the wind speed maximum height (Denby and Greuell, 2000). Flow oscillations, inducing periodic oscillations

of the wind-speed and the katabatic layer depth, are also a common feature of katabatic flows (McNider, 1982). The cospectra of $w$ with $u$ was slightly positive (Fig. 6d), suggesting the wind maximum was most of the time just below the EC system, in agreement with the average observed wind maximum height of the 2009 campaign for WF conditions (Fig. 5). Assuming the flow oscillated, the depth of the katabatic layer and the height of the wind-speed maximum would have oscillated too, shifting from above to below the EC system. This could be an explanation for the presence of low frequency contributions in the wind

speed spectra, and the erratic behavior of the cospectra of $w$ with $\theta$ at low frequencies (Fig. 6): changes in the depth of the

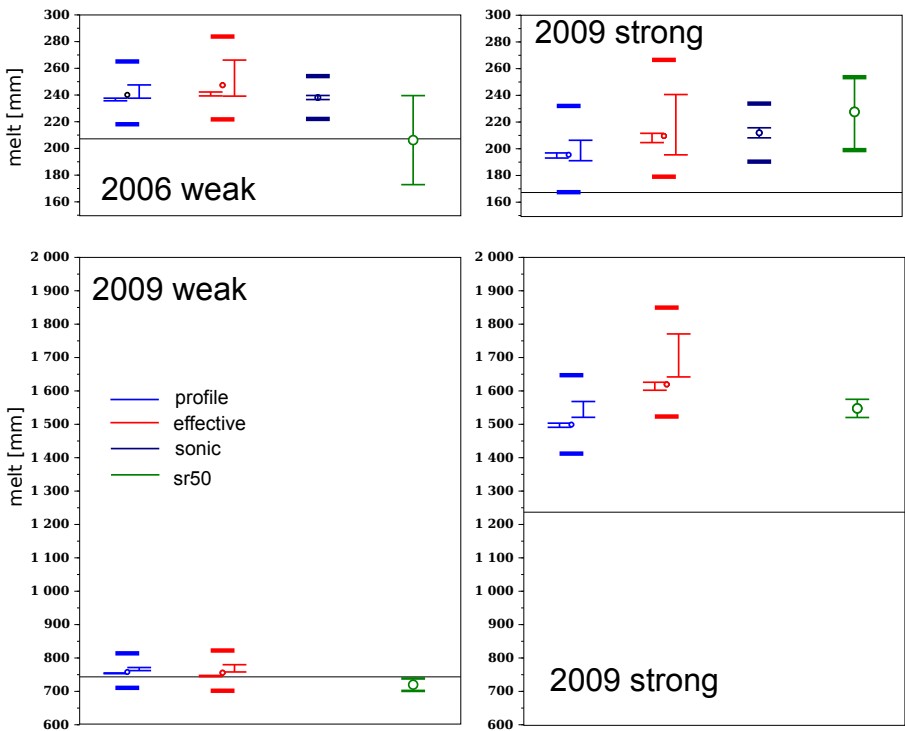

**Figure 9.** Cumulated melt from the SR50 and the SEB calculated with different methods, for WF with low TKE and SF with high TKE conditions during the 2006 campaign, and WF and SF conditions during the 2009 campaign. (black horizontal line) radiative balance, (dark green) melt measured by the SR50 and (light blue) net turbulent flux ($H + LE$) evaluated with the profile method, (red) with the effective roughness and (dark blue) with the EC method. The maximum and minimum melt values obtained with the Monte-Carlo approach are shown, with random errors due to the turbulent fluxes uncertainty. The thick horizontal bar show the maximum and minimum melt values due to the error on net shortwave radiation. The interval for the SR50 error is shown.

katabatic layer would provoke an erratic divergence of the fluxes in the first meters above the surface. The concept of a surface layer was maybe not even relevant below a wind speed maximum at less than 1.5 m. This would explain why $H$ from the profile method and the sonic method evaluated at a 2-m height were not well related (Table 3), even though on average the errors canceled out. Though, $LE$ fluxes remained well related. The profile method in its bulk form was shown to be suitable
5  below the wind-speed maximum by Denby and Greuell (2000). It is also possible that air temperature errors due to the influence of strong solar radiation inside the plate shield (Huwald et al., 2009) could have remained even though it was mechanically aspirated (Sicart et al., 2014), or simply that the random errors were relatively larger than the much smaller absolute fluxes.

## 5.2 Surface energy balance and melt

During both campaigns, the SEB was mainly controlled by large radiative fluxes, regardless of large-scale forcing (Fig. 9).
10  The contribution of turbulent fluxes to the SEB was significant only for SF. Errors on turbulent fluxes, on the radiation and

on the measured melt were large in comparison with the total SEB, and all SEB calculations compared well with the SR50 measurements within the estimated error range, for SF and WF conditions. Changes in net turbulent fluxes resulting from the choice of the calculation method remained too small in comparison with the errors to yield significant differences in the SEB estimates.

For WF, the melt was not sufficiently influenced by turbulent fluxes in order for the different methods to yield significant changes in the simulated melt. For SF conditions though, in 2009, the total melt sampled was larger than in 2006, and the difference between the melt obtained with the profile roughness lengths and the effective roughness lengths was slightly larger in comparison with 2006 cases. This suggests that if SF conditions were frequent, discrepancies between turbulent fluxes calculation methods would lead to significant changes in the melt estimates from SEB models. The errors in the turbulent fluxes also grew significantly when using effective roughness lengths (Figure 7. In these cases the low-frequency oscillations evidenced in the spectra could have been originating from large-scale orographic disturbances transporting TKE that did not scale with the mean vertical gradients in the surface layer (Sect. 4.2). This hypothesis explain the underestimation of the turbulent fluxes by the profile method, and suggests the effective roughness length method actually corrects for this effect. Simultaneous measurements of melt, wind speed and temperature profiles and sonic fluxes are needed to validate this hypothesis: the surface roughness might just have been different in 2006 and 2009.

## 5.3  Weather patterns

The analysis of meteorological conditions and wind regimes shows that using the WP decomposition of Garavaglia et al. (2010) can be useful to assess turbulent surface fluxes characteristics. The WP analysis is helpful in identifying different kind of circulations in the valley: for WF conditions local thermal winds such as katabatic flows drive the circulation whereas in SF the circulation is dominated by large-scale flows. It also provides a rough classification of the turbulence conditions in the glacier surface-layer, since SF and WF conditions are, in the glacier surface layer, likely associated with high and low TKE conditions, respectively. We show that the profile method compared differently with the sonic method for different WP (Figure 7, 8, and Table 3). More generally for SEB studies, we highlight some limitations: For weak forcing conditions only one WP is used which is always associated with weak wind speeds on the glaciers, and the SF conditions by strong winds, but other meteorological variables can be changing significantly.

## 6  Conclusions

During the summers of 2006 and 2009, field measurement campaigns were undertaken in the ablation zone of Saint-Sorlin Glacier, in the French Alps. We analysed Eddy-Covariance data from the 2006 campaign and temperature and wind-speed vertical profiles from the 2009 campaign. We characterized the wind regimes and associated surface-layer turbulent flows, in relation to the large scale forcing as characterized from the weather pattern decomposition of Garavaglia et al. (2010). We evaluated the turbulent fluxes with the profile method using observed and melt-calibrated (effective) roughness parameters and the sonic (Eddy-Covariance) method. We also calculated the surface energy balance and studied the impact of the choice of

flux calculation methods on the modeled melt, in relation with weather patterns. Errors resulting from random measurement errors, uncertainties in surface parameters (roughnesses or emissivity) and radiation measurements were assessed.

The sensible heat fluxes ($H$) were warming the surface, they were generally larger in magnitude than latent heat fluxes ($LE$) which were, on average, a small loss of energy for the glacier. When synoptic forcing was weak, local thermal effects drove the valley wind circulation. A katabatic wind-speed maximum was frequently observed at low height on the glacier (around 2 m, 71% of WF conditions). The turbulent kinetic energy (TKE) was generally low ($< 1$ m$^2$ s$^{-2}$, 68% of the cases), and both $H$ and $LE$ remained small in magnitude. When synoptic forcing was strong, under the influence of low-pressure systems, the large-scale winds roughly aligned with the glacier flow and drove the wind circulation. High wind speeds were observed on the glacier, the TKE was generally high ($> 2$ m$^2$ s$^{-2}$, 52% of the cases) and the katabatic wind-speed maximum was not frequently observed below 5 m ($<$40% of the time). Sensible heat fluxes were high (sometimes $>$100 W m$^{-2}$) in these conditions, due to high wind speeds. The magnitude of negative latent heat fluxes, mainly evaporation ($LE < 0$) increased only moderately since the air remained humid, and $LE$ did not cancel the energy gains due to $H$, so the net flux ($H + LE$) was highly positive.

For all conditions, low-frequency oscillations were observed in the wind-speed signals. For WF when TKE was low, as often observed for weak synoptic forcing, this could have been due to oscillations of the katabatic flow. Non-equilibrium of the surface-layer could explain the erratic discrepancies between sonic fluxes and profile fluxes when TKE was low in WF cases (Section 5.1). Anyway, the mean turbulent fluxes calculated with the profile or the sonic methods tended to be similar, since the differences between them compensated when averaged over several runs. When TKE was high in SF, low-frequency oscillations could have been provoked by large-scale orographic disturbances. Former studies in similar context (Smeets et al., 1999; Litt et al., 2015b) suggest that in these cases, TKE transport is non-negligible and that the surface layer turbulence is not in equilibrium with local turbulence production. Such low-frequency oscillations influenced the turbulent momentum and heat fluxes in the surface layer. This could explain systematic underestimation of the sonic fluxes by the profile fluxes was observed when TKE was large for SF cases, which could not be attributed to random or systematic measurement errors. The sonic fluxes were better estimated using a larger effective roughness.

Surface energy balance calculations compared well to the observed melt, but errors were large (Figure 9). During the 2006 campaign, using turbulent fluxes from the profile or from the sonic method did not provide significantly different melt results. During the 2009 campaign, melt (with turbulent fluxes evaluated with the profile method) was much larger than during 2006 events. As a result the difference between the melt calculated with different roughnesses was slightly larger than in 2006, even though still not significantly different due to the large uncertainties. This suggests that for frequent SF conditions, systematic biases might affect the profile method and may lead to underestimating the melt rate on alpine glaciers. The use of an effective roughness to calibrate the SEB on the observed melt increases the fluxes and thus can act as a correcting parameter.

Studies covering more melt periods are necessary to better understand how weather patterns relate to the different SEB terms and to better understand the links between large-scale forcing and mass balance in the vicinity of the glaciers, using dedicated variables to drive a weather pattern decomposition. This would help to understand the climatic processes governing inter-annual variations in the melt regimes of the glaciers. Furthermore, the potential turbulent flux biases might impact the total calculated melt significantly, when calculated over long periods. Turbulent flux error studies would be necessary on other

glaciers where turbulent fluxes dominate the SEB, e.g. at high latitudes, such as Storgalciaren (Sicart et al., 2008), or the Canadian artic (Braithwaite, 1981), to assess the effect of these potential biases on melt rate evaluation from climate data.

*Acknowledgements.* This work was funded by the French SO/SOERE GLACIOCLIM (http://www-lgge.ujf-grenoble.fr/ServiceObs/index. htm) and the ANR program TAG 05-JCJC-0135. We kindly thank EDF for providing the daily time series of WPs. We thank Romain Biron and Jean-Philipe Chazarin for the technical and field work, Adrien Gilbert and Marion Reveillet for stimulating discussions, and ana unknown reviewer for his meticulous and helpful review.

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
