# Peer review of "Surface-layer turbulence, energy-balance and links to atmospheric circulations over a mountain glacier in the French Alps."

_The Cryosphere, 2016_

## Referee Comment (RC1) · Anonymous Referee #1 · 30 May 2016

**Referee review for manuscript tc-2016-93, under review for The Cryosphere.**

"Surface-layer turbulence, energy-balance and links to atmospheric circulations over a mountain glacier in the French Alps" by Maxime Litt, Jean-Emmanuel Sicart, Delphine Six, Patrick Wagnon, and Warren D. Helgason.

**General Comments**

The paper presents an analysis of micro-meteorological data from two periods on the mountain glacier St Sorlin. These data provide the platform to assess the contribution of turbulent fluxes, and their uncertainties, to modelled glacier surface melt in the context of different weather types. The measured turbulence spectrum is presented and compared to theoretical predictions that form the basis of the widely used Monin-Obukhov similarity theory. Large deviations from theoretical predictions are found in both weakly and strongly turbulent conditions. The ability to correctly model the turbulent heat fluxes using bulk aerodynamic methods is discussed in the context of weather types and the characteristic turbulence regimes they experience. The authors find that despite the divergence of measured turbulence from theoretical predictions, simple schemes are able to model the melt to within the uncertainty in observed melt.

The paper presents useful and insightful data that support emerging conceptual models of the influence of outer layer turbulence on surface layer turbulent fluxes in glacial environments. The analysis of weather types and the associated characteristic valley circulation, wind speed and temperature profiles and TKE relationships are compelling and link well together. The analysis of turbulence spectra and co-spectra is clear and fits well into the progression of analysis.

The attempt to link these fine scales of turbulence to surface melt, through the calculation of fluxes in using the bulk aerodynamic (BA) method, is ambitious but makes sense in the conceptual framework of the paper. However, the formal links between each scale of analysis are not always well made and this reduces the confidence in the interpretations made. The large scope also makes the paper somewhat disjointed. At times analyses are presented that are not entirely relevant to the key points e.g. a time series of net turbulent heat fluxes for each period. Other analysis necessary to support the points being made are missing e.g. a direct comparison of hourly turbulent heat fluxes from EC and BA methods along with their uncertainties. Despite these omissions, the methods are well described and logical with some gaps noted in the specific comments.

There are clearly some interesting interactions occurring between the magnitude of net turbulent fluxes and the quantity of melt, which deserve further elucidating. The EC method gives the lowest melt of all methods, yet the has the same or higher net turbulent heat fluxes as the other methods. Why do we see a breakdown of theoretical predictions, yet no impact on modelled melt? Further description of the processes occurring at the hourly and daily scale

are needed to formally link uncertainties in the calculation method for turbulent heat fluxes to melt. Along with this, one of the key points introduced is the errors in the calculation of the turbulent heat flux, yet this is poorly addressed in later analysis. This would put the paper in a much stronger position to comment on the conditions in which the violation of current turbulent flux modelling methods will have an impact on melt.

The sub-setting of the analysis by weather types (i.e. Strong/ weak gradient wind forcing) is a particularly interesting approach. I find no issue in sub-setting the spectral analysis by TKE, rather than the weather types as for most of the other analysis. However, the discussion should focus around one or the other for clarity, rather than switching between the two. As it stands this switching is rather confusing.

On the whole, the paper is likely to make a useful contribution to glacier meteorology and surface energy balance with revision. Textual revisions and further analyses are needed for the manuscript to be acceptable, particularly analyses comparing hourly turbulent heat fluxes and their associated errors from each method. Some of the content is perhaps more suited to an applied meteorological journal, though I do not see that the manuscript is entirely out of the scope of *The Cryosphere* given the widespread use of energy balance modelling within the cryospheric community. Preferably, the emphasis on glacier melt and uncertainties/ validation of modelling approaches for turbulent heat fluxes can be further highlighted.

**Specific Comments**

P2 ln2. "englacial"- do you mean inside the glacier? Or rather in catchments with glaciers in them, which would be described "glacierised/ glacierized".

P2 ln11. Correct date - Anderson (2010). Also note that recent work has indicated the contribution of turbulent heat fluxes to melt was very likely overstated in this paper (see Conway and Cullen, 2016), so it is, perhaps, not the best reference to use.

P2 ln20. The work of Denby and Greuell (2000) and associated papers needs to be addressed in the introduction as this has been a key paper justifying the use of the BA method over glacier surfaces.

P3 ln18. Were any corrections for tilting of the radiometer measurements necessary given the high melt rates? If so, please detail the procedure used to correct for tilting of the mast, or if corrections were not performed please comment on the effect of tilt on the radiation values.

P4 ln5. Please explain why 1-hour runs were chosen over standard 30-minute runs?

P5 ln2. "apparition" -> appearance

P5 ln18. geopotential -> geopotential height

P5 ln19. Please explain the procedure used to 'analyse and compare' each day of the study period to the WP, particularly if this is an objective or expert judgement procedure.

P6 ln9. What do you mean by "bad" weather conditions? Please replace with a more descriptive comment.

P6 ln11. Table 3 -> Table 2

P6 ln15. More accepted acronyms are MOST or M-O theory, please use one of these throughout.

P7 ln20. What runs were chosen for the analysis of roughness lengths, and how was stability taken into account? This is especially important given the low wind speed maximum observed. Also, why was the EC data not used to analyse roughness lengths?

P7 ln25. The important small scale topography referred to by Smeets and van den Broeke (2008a) is on horizontal scales 5 to 10m, which suggests that the evolution of topography on Saint-Sorlin on the order of 20-30 cm over a few metres should have an impact on the aerodynamic roughness lengths. Also note that in another paper (Smeets and van den Broeke, 2008b) the authors find these same hummocks have a profound effect on scalar transfer. Please change this sentence to accurately reflect the papers conclusions.

P9 ln3. Please refrain from using parentheses to denote opposites and reword these sentences appropriately: i.e. "The symbols SW and LW stand for hourly mean shortwave and longwave radiation, respectively."

P9 ln10. Please justify the exclusion of heat from precipitation. This is only reasonable if the contribution of this flux can be shown to be negligible.

P10 Section 4.1.1. It is not clear how this text describing the temporal progression of the meteorology is central to the paper and should be shortened to a few sentences. Likewise, section 4.1.3 should be shortened and included in here.

P13 ln9. Please fix the use of the parentheses as per comment on p9 ln3.

P13 ln12. Please introduce the acronym GQR. Also do you mean $H_{EC} > 5$ W m$^{-2}$ given the sign convention used in the paper?

P13 ln20. Is it reasonable to use z/L from the BA method when you show later that the BA method underestimates the sensible heat flux and therefore, is likely to incorrectly represent z/L? Also, please explain why z/L was calculated from the BA method and not directly from the EC measurements.

P14 ln12 and further references: "Kaimal curve" -> "Kansas curve". This is more descriptive and is consistent with p6 ln31.

P14 Section 4.2.3 The first two paragraphs can be shortened to a few sentences as it is not clear how the temporal progression of the SEB is central to the paper. Conversely, further discussion of the results in Table 2 are needed, as is a presentation of a direct comparison of turbulent fluxes from the EC and BA methods. This would ideally take the form of scatter plots of hourly fluxes that include error bars. At the very least some descriptive statistics of the correlation between and spread within each (measured and modelled) flux are needed.

P14 ln33 and further references. For clarity, acronyms for the two BA calculation methods need to be introduced earlier and used throughout, e.g. BA1, BA2 or $BA_{pro}$ $BA_{eff}$. The descriptive names "the BA method based on the profile-derived roughness lengths" become confusing when comparing methods.

P15 ln2. The analysis of errors in the EC and BA methods needs to be presented as it is the key link between the representation of turbulence mechanisms and surface melt. This should include uncertainties on the figures given in Table 2 as well as error bars on any hourly fluxes presented.

P15 ln9. Previsions -> conclusions?

P16 ln8. The discussion here about the TKE budget is not well framed and more theoretical background is needed before the results are presented. As it is, this first sentence is unclear and needs to be broken up and reworded.

P16 ln16. "The EC method probably accounted for this" I don't understand what you mean here. The measured EC fluxes account for all of the extra turbulence observed as they are based on the same data. Perhaps you mean there is extra turbulence not captured by the EC method, or that the EC was not entirely in the surface layer. Please clarify this in the text.

P17 ln8. The use of the word "probably" here and elsewhere (p16 ln 8, section 6) suggests the interpretations and conclusions reached may not be well founded. It would be better to frame the results in the context of a certain conceptual framework, noting where the results agree or disagree with this framework.

P17 ln10. "were decorrelated from the surface fluxes". These analyses need to be shown in the paper.

P17 ln16. Where does the +/- 10% error on the surface height come from? Earlier an uncertainty of 0.1 m was stated. The uncertainty for each type is presumably some combination of the daily sum of melt for each type and the instrument uncertainty.

P17 ln33. Why were the BA fluxes not validated against the EC fluxes you already have?

P17 ln34. The effective roughness length is, in reality, smaller than the aerodynamic roughness length and larger than the scalar roughness lengths. Please change.

P18 Section 5.3. It is good reflect on the limitations of the particular weather typing method used, but the authors need to comment on if these limitations have a real bearing on the analyses made. If they do have a significant bearing on the results, then perhaps a different method needs to be chosen.

P19 ln20. 'Sublimation' -> perhaps 'evaporation' would be more appropriate given you state the surface was melting most of the period.

P19 ln28. "erratic discrepancies between EC fluxes and BA fluxes". These crucial elements of the analyses are not presented and need to be.

P20 ln2 to 8. These conclusions a very speculative considering no EC measurements are available and that both methods yield melt that is within the measurement uncertainty. Please revise.

P20 ln14. Please be more specific about which high latitude glaciers have large contributions of turbulent heat fluxes.

Table 1. It would be useful to know the maximum and minimum height for each instrument, or at least the standard deviation from the mean height.

Table 1. The accuracy of the CNR1 is listed as 0.4% - what is the justification for this figure, given the nominal accuracy should be on the order of 5 to 10%. Similarly, for the CSAT3, the accuracy is on the order of ± 2 % and ± 6% of the wind speed for attack angles of 5 and 20 degrees from horizontal. This equates to uncertainty on the order of 0.1 ms$^{-1}$ or larger for typical wind speeds. A more careful justification of the accuracy values is needed in the text.

Table 2. Why were average wind speed, air temperature etc. from the glacier AWS not included here?

Table 2. The units for q would be more simply expressed as g/kg. Also the values seem an order of magnitude too high – typical values would be 5 g/kg and here they are 50 g/kg. Please check the units of these values.

Table 2. The table would be much better split into two or more tables that can be inserted at the appropriate sections (data, results). Also you state you do not analyse the "other forcing" category further, but present it here. It would be less confusing to exclude it entirely. Similarly, the inclusion of both the weather types and the TKE bands is confusing and should be clarified.

Figure 1. It would be good for this figure to be larger and for the photos to be in colour.

Figure 3. This figure is very hard to read due to the hourly data used, small size and lack of grid lines. Perhaps either daily means of each flux can be presented, or the figures made

substantially larger. Tick marks also need to correspond to some meaningful interval (rather than 1/6th of 20 days).

Figure 7. Caption: (b) spectra of w, (c) co-spectra of w and theta

Figure 9. These error bars seem unrealistically small – especially given the large spread in H from an uncertainty in surface temperature of +/- 1 K. Further justification of these errors is needed.

Figure 9. Please use a different shading for H+LE that is not the same as for melt.

Figure 9. Please explain where the 10% error bar for melt comes from.

Figure 9. Please explain why the relatively magnitudes of melt (Figure 9) for each method differ markedly from those for the net turbulent heat flux (Table 2). e.g. EC has the highest net flux in weak forcing, but the lowest melt?

**References (not already in the manuscript)**

Conway, J. P. and Cullen, N. J.: Cloud effects on surface energy and mass balance in the ablation area of Brewster Glacier, New Zealand, The Cryosphere, 10, 313-328, 2016.

Smeets, C. J. P. P. and van den Broeke, M. R.: The Parameterisation of Scalar Transfer over Rough Ice, Boundary-Layer Meteorology, 128, 339-355, 2008b.

---

## Referee Comment (RC2) · Anonymous Referee #2 · 15 Jun 2016

Surface-layer turbulence, energy-balance and links to atmospheric circulations over a mountain glacier in the French Alps

Litt, M. et al.

The paper uses eddy covariance and vertical wind-speed and temperature profiles to quantify the errors inherent in using the bulk aerodynamic method for energy balance modeling. Errors were estimated by comparing model outputs using the three approaches over two measurement periods. The authors used large scale weather patterns to constrain their comparisons.

The authors should be commended on their use of interesting datasets. However, they

need to refocus the Results and Discussion sections to better elucidate the conclusions of the paper.

General comments

- Impact of changing sensor heights

In Section 2.2 a description of the changing sensor heights is given, as well as the manual lowering used to overcome these changes. The following remains unclear: was the 60-90 cm change accounted for in the flux calculations, or was a fixed height used (is unclear in Section 2.3)? Are the 'mean' heights given the height the sensors were changed to, the height after 5 or 7 days of change, or other? How do changing heights impact the flux results? Were different height values tested during calculation of the fluxes to understand possible errors?

- Surface properties

Sections 2.2 and 3.2 describe the changes to the surface observed in each campaign. Can the fetch properties also be added to Section 2.2 so that it is clear that the instrument heights are appropriate for the corresponding homogeneous surface (especially for the eddy covariance measurements).

- Calculation of roughness lengths

Section 3.2 describes the computation of roughness lengths using the profile data. Can you provide an explanation as to why the lengths were not calculated using the eddy covariance measurements, or, compare the lengths calculated using each method. Also, it would be useful to include an error range of the roughness length, as whilst it is stated that it 'did not change significantly', it would be useful for the reader to be able to evaluate the range (and associated error).

- Role of sublimation

Section 3.3 shows the formula used to calculate melt based on the SR50 measurements. Can you please add a comment to describe the role of sublimation, and why it is excluded from a mass loss calculation.

- Results

Currently the Results section combined comments related to both timing of events as well as classification by weather type. Please restructure to only focus on the weather patterns (as that is what is used in the Discussion section). Changing between time and pattern becomes confusing for the reader, as it becomes a lot of information to keep in mind.

- Results Section 4.1

Currently Section 4.1 is quite dense and difficult for the reader to pull apart. Please restructure in a way that is easier for a reader to follow. For example: "SF conditions are characterized by ____________. This was observed _______ (example). The impact on melt is _________." In this way, the reader can focus (and retain) the most relevant information associated with each weather pattern (rather than interchanging between time periods and weather patterns).

- Results Section 4.2.2

As the paper is directed at a general glaciology audience and not at a specialized micro-meteorology community, there needs to be a better introduction to spectral analysis within this section as a reminder. Perhaps the addition of one or two sentences describing why the analysis is necessary and what relevance it has to the theme (ie that it will help to describe the turbulent boundary layer) is necessary to ease the reader into the section. Even though it is briefly described in Section 3.2, it would be beneficial to restate the information here. Also, the term 'Kansas curve' and 'Kaimal curve' is used interchangeably throughout the document – please standardize.

- Discussion Section 5.3

If there are concerns regarding the comparison with the weather patterns, these should

be quantified.

-Results-Discussion-Conclusions disconnect

In the current form, there appears to be a disconnect between the Results, Discussion and Conclusions sections. Often, results required to support statements made in the conclusions (eg: erratic discrepancies between results during certain conditions) are not supported in the Results section, and additionally, certain sections of the Results (eg. Meteorology and wind regimes) are not fully utilized in the Discussion/Conclusions sections. I would recommend to the authors to visualize the conclusions and include only the results necessary to address those points.

Specific comments

P1 L3 Please clarify the time period in which each dataset was collected – from the description it seems that eddy covariance data and the profile data was collected during both the 2006 and 2009 study periods.

P2 L2 Change 'englacial' to 'glacierized' (or similar)

P5 L5 Why was a 1-hour period used instead of the standard 30-min run?

P5 L19 Please explain how the data was 'analyzed and compared'.

P6 L11 Change Table 3 to Table 2

P9 L4 Please minimize the use of brackets. It makes the sentence confusing for the reader.

P12 L9 Change 'again a after' to 'again after'

P13 L9 Please minimize the use of brackets. It makes the sentence confusing for the reader.

P13 L10 Change 'selecting 2 subsets' to 'selecting two subsets'

P13 L15 Please define GQR.

P15 L9 Change 'previsions' to 'conclusions' (or similar)

P19 L20 The introduction of sublimation here seems out of context, should this be evaporation? There should be some mention of this ablation fraction in the results section.

Figure 3 Please increase the size of the graphic to improve readability.

Figure 8 It would be beneficial to replace this graphic with a direct comparison of each flux for the different weather patterns (scatterplots).

Figure 9 The error bars seem to be very small, please justify.

---

## Author Comment (AC1) · 20 Aug 2016

We thank the reviewer for her-his constructive and thorough review of the manuscript. We provide an updated version of the manuscript, written according to the reviewer's comments and suggestions, as well as those from the second reviewer.

We responded to all of the general and specifics comments below (reviewers comments are in black bold, our response in blue), and provide, when relevant, a reference to the corresponding position of the related changes in the text. We provide an edited version of the paper, including track changes (in red, added or modified text, and suppressed text), and a final version. Many issues were also raised by the other reviewer, we simply provide the same answers in these cases.

**General comments**

**The paper presents an analysis of micro-meteorological data from two periods on the mountain glacier St Sorlin. These data provide the platform to assess the contribution of turbulent fluxes, and their uncertainties, to modelled glacier surface melt in the context of different weather types. The measured turbulence spectrum is presented and compared to theoretical predictions that form the basis of the widely used Monin-Obukhov similarity theory. Large deviations from theoretical predictions are found in both weakly and strongly turbulent conditions. The ability to correctly model the turbulent heat fluxes using bulk aerodynamic methods is discussed in the context of weather types and the characteristic turbulence regimes they experience. The authors find that despite the divergence of measured turbulence from theoretical predictions, simple schemes are able to model the melt to within the uncertainty in observed melt.**

**The paper presents useful and insightful data that support emerging conceptual models of the influence of outer layer turbulence on surface layer turbulent fluxes in glacial environments. The analysis of weather types and the associated characteristic valley circulation, wind speed and temperature profiles and TKE relationships are compelling and link well together. The analysis of turbulence spectra and co-spectra is clear and fits well into the progression of analysis.**

**The attempt to link these fine scales of turbulence to surface melt, through the calculation of fluxes in using the bulk aerodynamic (BA) method, is ambitious but makes sense in the conceptual framework of the paper. However, the formal links between each scale of analysis are not always well made and this reduces the confidence in the interpretations made. The large scope also makes the paper somewhat disjointed.**

We agree with this comment and we considerably changed section 4 (Results) and section 5 (discussion), in order to better link the different scales of analysis. In section 4, Table 2 has been split in two tables. The first one is describing the main characteristics of the weather pattern based and the TKE based classifications in terms of meteorology, and the second one is containing the turbulent fluxes information for each class. This second table includes previously missing information about the regression coefficients between measured and modeled turbulent fluxes. Meteorology and wind regimes (4.1) are now described differently: first for weak forcing, then for strong forcing. The subsections in 4.1 have been removed. The TKE description (4.2.1) has been simplified and the related Fig. 6 has been removed. We profoundly changed the description of the turbulent fluxes (4.2.3), we removed Fig. 8 and added a figure comparing the measured and the modelled fluxes (Fig. 7 in the new version).  The discussion about the turbulence in the surface layer (5.1) is now better framed, and we provide a better estimate of the error on the melt energy as derived with the sonic ranger (5.2).

Table 1 has been updated with more information about the sensor heights.

Minor changes have also been made in the Introduction, 2nd and 3rd paragraph know provide a better overview of the katabatic flow issues. In methods section (3), we slightly modified the TKE description (section 3.2, paragraph 1), we provide a better description of the roughness length derivation method (section 3.2, paragraph 2) and include the EC-based method for roughness length derivation (related additional results are discussed in section 5.1, 2nd paragraph). Description of the error derivation method has been improved and has now its own subsection, 3.3.

We hope the new version will be more straightforward to read.

**At times analyses are presented that are not entirely relevant to the key points e.g. a time series of net turbulent heat fluxes for each period. Other analysis necessary to support the points being made are missing e.g. a direct comparison of hourly turbulent heat fluxes from EC and BA methods along with their uncertainties. Despite these omissions, the methods are well described and logical with some gaps noted in the specific comments.**

We agree with this comment and consequently we removed the figure of the time series of the fluxes (Fig. 8 in former version) and replaced it by a direct comparison of the fluxes from the EC and the BA method (Fig. 7 in the new version). Thus, the comparison is only available for the 2006 campaign.

**There are clearly some interesting interactions occurring between the magnitude of net turbulent fluxes and the quantity of melt, which deserve further elucidating. The EC method gives the lowest melt of all methods, yet the has the same or higher net turbulent heat fluxes as the other methods. Why do we see a breakdown of theoretical predictions, yet no impact on modelled melt?**

Actually, our results show the net turbulent fluxes are higher in Strong Forcing than in Weak Forcing conditions. Focusing on $H$ ($LE$ is small or erratic due to large random errors) we show the BA method fluxes are lower than the EC method fluxes in Strong Forcing conditions. Considering the EC method is more reliable than the BA method, we consider the BA method underestimates fluxes in this cases. This effect is even more pronounced if you consider a subset of only large TKE: turbulent fluxes $H$ are quite different with the BA and the EC (Table 3, new version). The results suggest that an effective roughness length ($z_e$, introduced in former studies) can be used as a tuning parameter to correct the BA method for this underestimation.

But, since the total melt results from turbulent exchanges **and** radiative exchanges, we are not sure that we clearly understand this comment. Relating turbulent fluxes to melt cannot be done without considering the whole surface energy balance, and especially the radiative balance which plays a key role on melt. The total melt can be low if the radiative balance is low, even with strong net turbulent fluxes, and *vice versa*. That explains why we do not find a clear relationship between turbulent fluxes magnitudes and melt. Actually, the radiative balance is controlled by the albedo. In section 5.2, the first sentence states: *"During both campaigns, the SEB was mainly controlled by large radiative fluxes, regardless of large-scale forcing, but the contribution of turbulent fluxes to the SEB was significant"*.

In section 5.2 which describes Fig. 9 (former version, now Fig. 8), we added: *"Yet, changes in net turbulent fluxes resulting from the choice of calculation method remained too small in comparison with the radiative balance to yield significant differences in the SEB estimates."* We hope the issue is clearer now.

**Further description of the processes occurring at the hourly and daily scale are needed to formally link uncertainties in the calculation method for turbulent heat fluxes to melt. Along with this, one of the key points introduced is the errors in the calculation of the turbulent heat flux, yet this is poorly addressed in later analysis. This would put the paper in a much stronger position to comment on the**

**conditions in which the violation of current turbulent flux modelling methods will have an impact on melt.**

Regarding error calculation, we apply here methods to estimate the random errors resulting mainly from instrumental uncertainties that has been developed on Zongo glacier in Litt et al., (2015). We only briefly describe these methods herein, to avoid having a too long paper, but we could add more details if required by the reviewers. We encourage the reader to check the above mentioned paper where all the error calculation processes are described in detail. As presented in section 4.2.3, and discussed in section 5.1, page 18 lines 3-7 (new version), the main result here is that the random errors cannot explain the differences between the EC and the BA methods when TKE is high. We suggest the difference is due to the inability of the BA method to capture part of the flux in non-stationary conditions. We inserted the paragraph about errors into a new subsection (3.3), where we improved the method explanation. We also improved the interpretation of the results, especially for the turbulent fluxes, in relation with the turbulence characteristics, in section 5.1.

**The sub-setting of the analysis by weather types (i.e. Strong/ weak gradient wind forcing) is a particularly interesting approach. I find no issue in sub-setting the spectral analysis by TKE, rather than the weather types as for most of the other analysis. However, the discussion should focus around one or the other for clarity, rather than switching between the two. As it stands this switching is rather confusing.**

Since the classification by weather patterns doesn't always reliably work for describing turbulent conditions in the surface layer, we decided to focus on a characterization of the turbulence and turbulent fluxes using a TKE classification. Since the strong WP tend to select high TKE conditions and low WP, low TKE conditions, the TKE classification helps understanding the processes occurring in the WP-related classification. In the new version, we emphasized this point as much as possible throughout the text. In the new version, this point is evidenced in the results section 4.1, then most of the results are presented in terms of the TKE classification, and we finally discuss how that impacts the results of fluxes comparisons for the WP classification.

**Specific Comments**

**P2 ln2. "englacial"- do you mean inside the glacier? Or rather in catchments with glaciers in them, which would be described "glacierised/ glacierized".**

We meant catchments with glaciers in them. We changed to "glacierized mountain catchments".

**P2 ln11. Correct date - Anderson (2010). Also note that recent work has indicated the contribution of turbulent heat fluxes to melt was very likely overstated in this paper (see Conway and Cullen, 2016), so it is, perhaps, not the best reference to use.**

We removed the reference to Anderson and also to Gillet and Cullen. We added reference to Six et al., 2009, together with reference to Sicart et al., 2008, this must be sufficient to support the statement.

**P2 ln20. The work of Denby and Greuell (2000) and associated papers needs to be addressed in the introduction as this has been a key paper justifying the use of the BA method over glacier surfaces.**

We agree and the reference has been added. It seems that Denby and Greuell assessed the role of flux divergence, but associated papers (Denby, 1999; Denby and Smeets, 2000) do not mention effects of katabatic oscillations or non-stationnarity. Therefore, we consider this to still be an open question. To reflect this we modified the text of the introduction: *"Although numerical simulations have shown the*

*BA method was reliable in estimating the surface fluxes in the presence of flux divergence below a wind-speed maximum (Denby and Greuell, 2000), the effects of non-stationnarity and outer-layer interactions remains poorly documented over mountain glaciers (Smeets et al., 1998, 2000)"*

Added references:

Denby, B., Second-order modelling of turbulence in katabatic flows, Boundary-Layer Meteorology 92: 67–100, 1999.

Denby, B., and C.J.P.P. Smeets, Derivation of Turbulent Flux Profiles and Roughness Lengths from Katabatic Flow Dynamics, Journal of Applied Meteorology 39: 1601-1612, 2000.

**P3 ln18. Were any corrections for tilting of the radiometer measurements necessary given the high melt rates? If so, please detail the procedure used to correct for tilting of the mast, or if corrections were not performed please comment on the effect of tilt on the radiation values.**

Corrections were not performed, since tilting was not automatically measured. We assumed the potential correction on the radiometer remained low, since the sensor was rarely found to be out of alignment during each field visit (every 10 days), and it was levelled every time small deviations were noted.

**P4 ln5. Please explain why 1-hour runs were chosen over standard 30-minute runs?**

The choice was constrained by the parallel use of the off-glacier (on the moraine) meteorological station data for which sampling was set on 1-hour. Calculations of the fluxes have been done with 30 min based runs and did not changed the relative contribution of the fluxes to the SEB.

**P5 ln2. "apparition" -> appearance**

Corrected

**P5 ln18. geopotential -> geopotential height**

Corrected when necessary

**P5 ln19. Please explain the procedure used to 'analyse and compare' each day of the study period to the WP, particularly if this is an objective or expert judgement procedure.**

The weather pattern series has been kindly provided by "Electricite de France" (EDF), and we did not undergo this analysis ourselves. The paragraph provides a brief explanation of the whole method. For the specific step of analysis and comparison, each day, the shape of the observed geopotential height field over Europe is characterized by the geopotential height at 0h and 24h at the 700 hPa level and the 1000 hPa level for 110 grid points (a total of 440 points). This field is compared, in this 440 dimensions space, to the 8 geopotential height fields types which have been obtained by the statistical sorting of measured rain patterns in south-east France. The WP of the day is determined by finding the WP for which the geopotential height pattern is the nearest to the observed geopotential height field in the 440 dimension space, using the Teweles-Wobus score (Garavaglia et al., 2010).

Detailing this complex procedure is beyond the scope of this paper. We provide reference to the paper from Garavaglia et al. 2010. We modified the text to add a short description for the selection of the WP: *"Then, for any day (inside or outside the period used to characterize the decomposition, e.g. 1956-1996) the observed geopotential height field shape over Europe is characterized by the observed geopotential height at 0h and 24h, at the 700 hPa level and the 1000 hPa level for 110 grid points (a*

*total of 440 points). This field is compared, in this 440 dimensions space, to the 8 geopotential height fields proposed by the rain-patterns analysis. The nearest of the 8 fields provides the WP."*

We added the following sentence at the end of the paragraph: *"Details of the procedure can be found in Garavaglia et al., 2010."*

**P6 ln9. What do you mean by "bad" weather conditions? Please replace with a more descriptive comment.**

We improved the text to be more specific. We meant rainy or freezing conditions which evidently disturb the good quality of the measurements.

**P6 ln11. Table 3 -> Table 2**

Corrected.

**P6 ln15. More accepted acronyms are MOST or M-O theory, please use one of these throughout.**

MOST is now used throughout.

**P7 ln20. What runs were chosen for the analysis of roughness lengths, and how was stability taken into account? This is especially important given the low wind speed maximum observed. Also, why was the EC data not used to analyse roughness lengths?**

The complete procedure, developed and presented in Sicart et al., 2014, is based on an iterative fitting of profiles between wind and temperature as described in Andreas et al., 2002. The selection of runs was based on a set of criteria, including neutrality (based on Richardson-bulk parameter analysis), quality of the fits ($R^2 > 0.975$), absence of a katabatic wind-speed maximum, and some others. We agree this is not clear in the text, so we included a statement referring to the Sicart et al., 2014 paper more explicitly: *"The method for roughnesses determination was inspired from Andreas (2002) and developed for the tropical Zongo glacier. It is detailed in Sicart et al. (2014)"*

We had not included the roughness values calculated with the EC system since we thought using only one point above the ground for this calculation was not reliable enough. Anyway, the dynamic roughness length had been evaluated with the EC system, by inverting the log-linear wind speed profile relationship. The values were dispersed but we found a mean value of $z_0$ around 0.02 m and $z_t$ around $6.6 \times 10^{-6}$ m when derived from the EC system. We now mention that in the new manuscript (page 8, line 17 ):

*"We also evaluated the roughness lengths using the EC system and inverting equations 4 and 5, and selecting neutral runs. The median $z_0$ was 0.022 m, and the median $z_t$  $6.6 \times 10^{-6}$ m. We did not use these values to calculate fluxes through the BA method."*

Interestingly, the factor $(\ln(z/z_0)\ln(z/z_t))^{-1}$, which is the denominator in the bulk formulation of the fluxes (assuming neutrality), calculated with the EC derived roughness lengths or the effective roughness is roughly identical (0.0165 and 0.0167). This shows that effective roughness lengths values can be used to compensate for the BA method underestimation of the fluxes. We now mention this results in the discussion section p18 line 6:

*"This is supported by the values of the EC derived roughness lengths, considering they account for the additional mixing: the denominator $(\ln(z/z_0) \ln(z/z_t))^{-1}$ in the bulk formulation of the fluxes, calculated with the EC derived roughness lengths or the effective roughness length is roughly identical (0.0165 and 0.0167)."*

**P7 ln25. The important small scale topography referred to by Smeets and van den Broeke (2008a) is on horizontal scales 5 to 10m, which suggests that the evolution of topography on Saint-Sorlin on the order of 20-30 cm over a few metres should have an impact on the aerodynamic roughness lengths. Also note that in another paper (Smeets and van den Broeke, 2008b) the authors find these same hummocks have a profound effect on scalar transfer. Please change this sentence to accurately reflect the papers conclusions.**

Thanks for highlighting this. We updated the text according to this comment, and added references, see text added:  Actually the geometry of the gullies was probably not high enough (up to 10-30 cm cm) to stand out against the dispersion in the aerodynamic roughness measurements (roughness values were ranging between $10^{-1}$ and $10^{-5}$ m). We updated the text as follow, p 8 line 5:

*"The results show a large scatter, about four orders of magnitude. The median values are $z_0 = 0.001$ m and $z_t = 0.00001$ m. The scatter can be attributed to poor accuracy of the temperature measurements leading to large random uncertainties (Sicart et al., 2014). The scatter was too large to observe significant changes in the measured roughness lengths during the 2009 campaign, in spite of snow falls or snow melt that uncovered the ice surface, or appearance of small gullies of about 0.1 – 0.3 m height variations on a few meters horizontal scale that could also have impacted the roughness lengths (Smeets and Van den Broeke, 2008a, b)."*

**P9 ln3. Please refrain from using parentheses to denote opposites and reword these sentences appropriately: i.e. "The symbols SW and LW stand for hourly mean shortwave and longwave radiation, respectively."**

Done.

**P9 ln10. Please justify the exclusion of heat from precipitation. This is only reasonable if the contribution of this flux can be shown to be negligible.**

This is a common assumption over alpine glaciers, with significant mass turnover. Also, heat advected by precipitation is generally negligible because the temperature difference between the rain and the ice is low, the rain intensity is small, and anyway rain is rarely observed. See Paterson (1994), and Oerlemans (2001). We added references in the text.

**P10 Section 4.1.1. It is not clear how this text describing the temporal progression of the meteorology is central to the paper and should be shortened to a few sentences. Likewise, section 4.1.3 should be shortened and included in here.**

We significantly modified the text in this section 4.1 to take into account your comment together with the other reviewer comments. Meteorology and wind regimes (4.1) are now described differently: first for weak forcing, then for strong forcing. The subsections in 4.1 have been removed.  The whole text is now much shorter and clearer, the whole section 4.1 included 42 lines, it now contains only 21.

**P13 ln9. Please fix the use of the parentheses as per comment on p9 ln3.**

Done

**P13 ln12. Please introduce the acronym GQR. Also do you mean $H_{EC} > 5$ W m$^{-2}$ given the sign convention used in the paper?**

GQR (good quality runs) is now introduced in the "Data processing" section. Yes, $H_{EC} > 5$ W m$^{-2}$ with our sign convention, we changed it.

**P13 ln20. Is it reasonable to use z/L from the BA method when you show later that the BA method underestimates the sensible heat flux and therefore, is likely to incorrectly represent z/L? Also, please explain why z/L was calculated from the BA method and not directly from the EC measurements.**

This was actually an editing mistake. We had used the z/L calculated from the Eddy correlation measurements. Corrected in the text.

**P14 ln12 and further references: "Kaimal curve" -> "Kansas curve". This is more descriptive and is consistent with p6 ln31.**
Corrected.

**P14 Section 4.2.3 The first two paragraphs can be shortened to a few sentences as it is not clear how the temporal progression of the SEB is central to the paper. Conversely, further discussion of the results in Table 2 are needed, as is a presentation of a direct comparison of turbulent fluxes from the EC and BA methods. This would ideally take the form of scatter plots of hourly fluxes that include error bars. At the very least some descriptive statistics of the correlation between and spread within each (measured and modelled) flux are needed.**

Following the reviewer comment, we removed the temporal description of the turbulent fluxes, and present a description of the fluxes with regard to the SF and WF conditions. We replaced the figure with a scatter plot of fluxes as measured by the EC and as modelled by the BA method. We updated the table 2 with the correlation coefficients and parameters of regressions. The text has been updated to include these results clearly.

**P14 ln33 and further references. For clarity, acronyms for the two BA calculation methods need to be introduced earlier and used throughout, e.g. BA1, BA2 or $BA_{pro}$ $BA_{eff}$. The descriptive names "the BA method based on the profile-derived roughness lengths" become confusing when comparing methods.**

Done

**P15 ln2. The analysis of errors in the EC and BA methods needs to be presented as it is the key link between the representation of turbulence mechanisms and surface melt. This should include uncertainties on the figures given in Table 2 as well as error bars on any hourly fluxes presented.**

The errors are now included when relevant in the figures and in Table 3 (former Table 2). They are discussed in the updated text.

**P15 ln9. Previsions -> conclusions?**

In the Hogstrom paper, the authors provide a parameterization of the variances. That we called prevision. We changed the sentence to: *"[…] agrees with the **parameterization** of Hogstrom […]".*

**P16 ln8. The discussion here about the TKE budget is not well framed and more theoretical background is needed before the results are presented. As it is, this first sentence is unclear and needs to be broken up and reworded.**

We entirely removed the section regarding the TKE budget since experimental data is insufficient to support the discussion. Also figure 6 has been removed. We reframed the TKE budget discussion, where the references included are sufficient to support our conclusions. We hope the discussion is clearer as is.

**P16 ln16. "The EC method probably accounted for this" I don't understand what you mean here. The measured EC fluxes account for all of the extra turbulence observed as they are based on the same data. Perhaps you mean there is extra turbulence not captured by the EC method, or that the EC was not entirely in the surface layer. Please clarify this in the text.**

We are sorry for the confusion. We meant that the EC method fluxes included the contribution of the low frequencies, whereas the bulk method didn't. Text has been updated to clarify.

**P17 ln8. The use of the word "probably" here and elsewhere (p16 ln 8, section 6) suggests the interpretations and conclusions reached may not be well founded. It would be better to frame the results in the context of a certain conceptual framework, noting where the results agree or disagree with this framework.**

We understand the reviewer refers to section 5, discussion. We agree with the statement. We included various changes in the section, to provide stronger background and description of the expected turbulence characteristics in different cases, and how this is reflected in our data. Detailed changes can be monitored from the edited version of the manuscript.

**P17 ln10. "were decorrelated from the surface fluxes". These analyses need to be shown in the paper.**

Reference to the Table 3 (formerly Table 2), that is now updated with correlation coefficients and linear regression parameters, has been added into the text to support this statement. Also the new figure 7 illustrates more clearly this results.

**P17 ln16. Where does the +/- 10% error on the surface height come from? Earlier an uncertainty of 0.1 m was stated. The uncertainty for each type is presumably some combination of the daily sum of melt for each type and the instrument uncertainty.**

The 10 % estimate was a rough estimation of the error on the melt derived from the SR50. The error we set of 0.1 m is the error used as the height uncertainty in turbulent fluxes error calculation. This is set like this since the exact distance of the various instruments to the ground is initiated from manual measurements so it remains quite uncertain. But for the daily melt, the height differences between days are used, for which we assume the absolute height is not relevant, and so we use the traditional sensor error of 0.01 m on daily melt. Propagating this error to hourly melt energy gives an error of 40 $Wm^{-2}$. This is not far from, but slightly larger than the assumed 10%. We changed it in the graphs where we now use this 40 $Wm^{-2}$ error.

**P17 ln33. Why were the BA fluxes not validated against the EC fluxes you already have?**

We removed the statement. We actually compare BA fluxes with the $z_{eff}$ from Six et al. 2009.

**P17 ln34. The effective roughness length is, in reality, smaller than the aerodynamic roughness length and larger than the scalar roughness lengths. Please change.**

$z_{eff}$ is set to 0.001 m, which is equal to $z_0$ but higher than $z_t$, $z_q$. We changed the text to *"an effective roughness length larger than the profile-derived dynamic or thermal roughnesses, respectively"*

**P18 Section 5.3. It is good reflect on the limitations of the particular weather typing method used, but the authors need to comment on if these limitations have a real bearing on the analyses made. If they do have a significant bearing on the results, then perhaps a different method needs to be chosen.**

The way the section was written probably didn't reflect the actual advantages of the WP decomposition we used. Actually, the Strong forcing/Weak forcing classification does select between distinct wind regimes and turbulent characteristics above the glacier, and as such is quite useful. There are some limitations for SEB studies in a more general sense. We modified the text, we added: *"can be useful to assess turbulent surface fluxes characteristics"* and *"It also provides a rough classification of the turbulence conditions in the glacier surface-layer, since SF and WF conditions are, in the glacier surface layer, likely associated with high and low TKE conditions, respectively. We show that the BA method compared differently with the EC method for different WP. More generally for SEB studies, we highlight some limitations."* . We removed *"is only partly adapted for SEB studies"*.

**P19 ln20. 'Sublimation' -> perhaps 'evaporation' would be more appropriate given you state the surface was melting most of the period.**

Agreed. Sentence changed to: "*[…] magnitude of negative latent heat fluxes, mainly evaporation […]*"

**P19 ln28. "erratic discrepancies between EC fluxes and BA fluxes". These crucial elements of the analyses are not presented and need to be.**

Done. The correlation analyses and regression equation are now included in Table 3. A reference to section "turbulent fluxes" has been added herein.

**P20 ln2 to 8. These conclusions a very speculative considering no EC measurements are available and that both methods yield melt that is within the measurement uncertainty. Please revise.**

The net turbulent fluxes for SF conditions are higher in 2009 than in 2006 for the same conditions. If we assume the BA method is biased, this likely means that the turbulent fluxes in 2009 are biased similarly as reported for the 2006 campaign. We only suggest the effective roughness can be used to scale the BA method fluxes. We have updated the text to reflect this.

**P20 ln14. Please be more specific about which high latitude glaciers have large contributions of turbulent heat fluxes.**

Good examples are the Storglaciaren (Sicart et al., 2008) in Norway, and glaciers in the Canadian Arctic (Braithwaite 1981). We added these references in the text.

**Table 1. It would be useful to know the maximum and minimum height for each instrument, or at least the standard deviation from the mean height.**

We included in the table the standard deviation of the heights, as well as the minimum and maximum height observed. There was a mistake in the mean heights for the Vaisala and the SR50, which has been corrected.

**Table 1. The accuracy of the CNR1 is listed as 0.4% - what is the justification for this figure, given the nominal accuracy should be on the order of 5 to 10%.**

This comes from a previous analysis we underwent in Litt et al., 2015.b. See appendix A of this paper:

*"The Kipp & Zonen sensor notice provides an estimated error of 10 % in the irradiance (for daily sums). If we apply this as a random error in the previous equation, it yields an unrealistically large relative error in estimates of surface temperature, i.e. as large as 6–7 K, considering a melting surface temperature of 273.15 K. Random noise must be lower; analysis of SW inc and SW out measurements during the night, when they should be zero, or of LW out when melting is observed, and thus longwave emission intensity must be constant, yields typical SDs of 1 or 2 W m −2 . This is equivalent to a random*

*noise of 0.4 % around radiation measurements".* This description has been adapted and included in the text in the new section 3.3.

**Similarly, for the CSAT3, the accuracy is on the order of ± 2 % and ± 6% of the wind speed for attack angles of 5 and 20 degrees from horizontal. This equates to uncertainty on the order of 0.1 ms$^{-1}$ or larger for typical wind speeds. A more careful justification of the accuracy values is needed in the text.**

We actually don't calculate any error resulting from noise on measurements for the CSAT3. Literature shows that it is too small to have a consequent impact on the measured fluxes. Rather, we compute the uncertainty due to other sources which is larger. We use methods developed in Litt et al 2015b. See original text still present in section 3.3: *"For the EC method, we followed Litt et al. (2015a), assuming measurement uncertainties on wind speed and temperature were negligible and that most random errors were due to insufficient statistical sampling of the largest eddies (Vickers and Mahrt, 1997)."*

**Table 2. Why were average wind speed, air temperature etc. from the glacier AWS not included here?**

We just wanted to give an overall estimate of the general meteorological conditions around the glacier (and away from its thermal influence), while not overwhelming the reader with details.

**Table 2. The units for q would be more simply expressed as g/kg. Also the values seem an order of magnitude too high – typical values would be 5 g/kg and here they are 50 g/kg. Please check the units of these values.**

Thanks for the comment. They are actually expressed in g/kg but there's an order of magnitude mistake. The mistake has been corrected.

**Table 2. The table would be much better split into two or more tables that can be inserted at the appropriate sections (data, results).**

Table 2 has been split in 2 as suggested. The new table (Table 3) includes the fluxes results and correlations and regression coefficients between BA methods and the EC.

**Also you state you do not analyse the "other forcing" category further, but present it here. It would be less confusing to exclude it entirely.**

The other forcing column has been removed.

**Similarly, the inclusion of both the weather types and the TKE bands is confusing and should be clarified.**

Since the WP classification did not allow a complete selection of the different turbulent conditions, the classification on TKE has been preferred to characterize the turbulence. Important results come from this classification, so we kept it. We updated the text (see comments above, *"this point is evidenced in the results section 4.1, then most of the results are presented in terms of the TKE classification, and we finally discuss how that impact on the results for the WP classification."* ) so that this is made clearer.

**Figure 1. It would be good for this figure to be larger and for the photos to be in colour.**

Figure is now larger and pictures now in color

**Figure 3. This figure is very hard to read due to the hourly data used, small size and lack of grid lines. Perhaps either daily means of each flux can be presented, or the figures made substantially larger. Tick marks also need to correspond to some meaningful interval (rather than 1/6[th] of 20 days).**

The Figure was made larger, interval of ticks were set to 1 day, dates appear every 7 days. Daily means are now shown on the plot, and in shaded gray we kept the hourly data.

**Figure 7. Caption: (b) spectra of w, (c) co-spectra of w and theta**

Changed

**Figure 9. These error bars seem unrealistically small – especially given the large spread in H from an uncertainty in surface temperature of +/- 1 K. Further justification of these errors is needed.**

Note that these are only errors on the mean turbulent fluxes. Since they are random errors, on average over all the available runs, the error is quite reduced, whereas it can be quite large for an individual measurement (see new figure 7). The error is calculated on the basis of an error of +/- 0.35 K on Ts (see Litt et al. 2015, and updated error methods section).

**Figure 9. Please use a different shading for H+LE that is not the same as for melt.**

Done

**Figure 9. Please explain where the 10% error bar for melt comes from.**

The 10 % was a rough estimate. Calculating an error of 0.01 m on day height changes yield an hourly error on the hourly available melt energy of 40 Wm$^{-2}$. This has been updated in the text, see specific comments above.

**Figure 9. Please explain why the relatively magnitudes of melt (Figure 9) for each method differ markedly from those for the net turbulent heat flux (Table 2). e.g. EC has the highest net flux in weak forcing, but the lowest melt?**

This is because the melt energy plot in fig 9 contains radiative fluxes (in gray) plus the net turbulent fluxes (in white). Furthermore the net turbulent fluxes (in *H+LE*) are larger in strong forcing, as shown in both table 2 (now 3) and figure 9.

---

## Author Comment (AC2) · 20 Aug 2016

We kindly thank the reviewer for her/his positive review that helped us to considerably improve the paper. We provide an edited version of the paper, including track changes (in red, added or modified text, and suppressed text), and a final version. We answered to all general and specific comments (reviewers comments are in black bold, our response in blue) and made changes in the text when necessary. Many similar issues were also raised by the other reviewer, we provide the same answers in these cases.

Please also note the supplement to this comment:

[Figure]

http://www.the-cryosphere-discuss.net/tc-2016-93/tc-2016-93-AC2-supplement.pdf

[Figure]

**Supplement:**

We kindly thank the reviewer for her/his positive review that helped us to considerably improve the paper. We provide an edited version of the paper, including track changes (in red, added or modified text, and suppressed text), and a final version. We answered to all general and specific comments (reviewers comments are in black bold, our response in blue) and made changes in the text when necessary. Many similar issues were also raised by the other reviewer, we provide the same answers in these cases.

**The paper uses eddy covariance and vertical wind-speed and temperature profiles to quantify the errors inherent in using the bulk aerodynamic method for energy balance modeling. Errors were estimated by comparing model outputs using the three approaches over two measurement periods. The authors used large scale weather patterns to constrain their comparisons. The authors should be commended on their use of interesting datasets. However, they need to refocus the Results and Discussion sections to better elucidate the conclusions of the paper.**

We considerably changed section 4 (Results) and section 5 (discussion), in order to assess this overall comment and that of the first reviewer. In section 4, Table 2 has been split in two tables. The first one is describing the main characteristics of the weather pattern based and the TKE based classifications in terms of meteorology, and the second one is containing the turbulent fluxes information for each class. This second table includes previously missing information about the regression coefficients between measured and modeled turbulent fluxes. Meteorology and wind regimes (4.1) are now described differently: first for weak forcing, then for strong forcing. The subsections in 4.1 have been removed. The TKE description (4.2.1) has been simplified and the related Fig. 6 has been removed. We profoundly changed the description of the turbulent fluxes (4.2.3), we removed Fig. 8 and added a figure comparing the measured and the modelled fluxes (Fig. 7 in the new version). The discussion about the turbulence in the surface layer (5.1) is now better framed, and we provide a better estimate of the error on the melt energy as derived with the sonic ranger (5.2).

Table 1 has been updated with more information about the sensor heights.

Minor changes have also been made in the Introduction, $2^{nd}$ and $3^{rd}$ paragraph know provide a better overview of the katabatic flow issues. In methods section (3), we slightly modified the TKE description (section 3.2, paragraph 1), we provide a better description of the roughness lengths derivation method (section 3.2, paragraph 2) and include the EC-based method for roughness lengths derivation (related additional results are discussed in section 5.1, $2^{nd}$ paragraph). Description of the error derivation method has been improved and has now its own subsection, 3.3.

We hope the new version will be more straightforward to read.

**General comments**

- **Impact of changing sensor heights**

**In Section 2.2 a description of the changing sensor heights is given, as well as the manual lowering used to overcome these changes. The following remains unclear: was the 60-90 cm change accounted for in the flux calculations, or was a fixed height used (is unclear in Section 2.3)? Are the 'mean' heights given the height the sensors were changed to, the height after 5 or 7 days of change,**

**or other? How do changing heights impact the flux results? Were different height values tested during calculation of the fluxes to understand possible errors?**

The value found in Table 1 is the mean over all the campaign. The changing sensor heights were taken into account in the flux calculation and the related uncertainty was included in the random error calculation. In order to obtain the day by day height changes, measurements from the SR50 (sonic height ranger) were used. The following statement can be found in section 2.3 : *"The changing instrument heights were derived from the sonic height ranger and regular field visits and controls"*. We now provide the range of variation and the standard deviation of the heights in Table 1.

The way the uncertainties were computed is described in detail in Litt et al. (2015b), where a complete analytical method had been developed to compute the random errors on the BA method and the EC method. The paragraph about errors in section 3.2 was transformed in a new section 3.3, and the description of the error assessment has been improved.

- **Surface properties**

**Sections 2.2 and 3.2 describe the changes to the surface observed in each campaign. Can the fetch properties also be added to Section 2.2 so that it is clear that the instrument heights are appropriate for the corresponding homogeneous surface (especially for the eddy covariance measurements).**

The glacier ablation area, over which the sensors were installed, showed homogeneous surface characteristics in all directions for distances of several hundred of meters. We assumed the fluxes and the turbulent characteristics derived at the measuring site were representative of the turbulent characteristics all over the ablation area. We added, p 5 line 15, *"The glacier surface characteristics remained homogeneous in every direction on hundreds of meters from the measuring site."*

- **Calculation of roughness lengths**

**Section 3.2 describes the computation of roughness lengths using the profile data. Can you provide an explanation as to why the lengths were not calculated using the eddy covariance measurements, or, compare the lengths calculated using each method. Also, it would be useful to include an error range of the roughness length, as whilst it is stated that it 'did not change significantly', it would be useful for the reader to be able to evaluate the range (and associated error).**

We had not included the roughness lengths calculated with the EC system since we thought using only one point above the ground for this calculation was not reliable enough. Anyway, the dynamic roughness length had been evaluated with the EC system, by inverting the log-linear wind speed profile relationship. The values were scattered but we found a median value of $z_0$ of 0.02 m and $z_t$ of 6.6 × $10^{-6}$ m when derived from the EC system, values higher than the one derived from the profiles. We mention that in the new manuscript lines, p 8, line 17.

Interestingly, the factor $(\ln(z/z_0)\ln(z/z_t))^{-1}$ in the bulk formulation, calculated with the EC derived roughness or the effective roughness, is almost identical (0.0165 and 0.0167). This supports the fact that the effective roughness lengths are used to compensate for the BA method underestimation of the fluxes. We mention this result now in the discussion section, p 18 line 6.

The roughness length evaluated with the profiles are scattered between 0.1 m and 0.00001 m, which prevents the observation of significant change since the maximum observed height changes at the surface reached 0.1 m at the end of the campaigns. These large errors are likely due to large uncertainties on the temperature measurements (Sicart et al., 2014). See the following insertion, p 8 line 5:

*"The results show a large scatter, about four orders of magnitude. The median values are $z_0$ = 0.001 m and $z_t$ = 0.00001 m. The scatter can be attributed to poor accuracy of the temperature measurements leading to large random uncertainties (Sicart et al., 2014). The scatter was too large to observe significant changes in the measured roughness lengths during the 2009 campaign, in spite of snow falls or snow melt that uncovered the ice surface, or appearance of small gullies of about 0.1 – 0.3 m height variations on a few meters horizontal scale that could also have impacted the roughnesses (Smeets and Van den Broeke, 2008a, b)."*

The fluxes error calculation included an error range on the roughness lengths, as mentioned in the original manuscript, in the error calculation description. This error on $z_{0,t}$ was evaluated as the result of the errors on temperature and wind speed measurements (Sicart et al., 2014) and is set to $dlnz_0$ = 1.5. This value corresponds roughly to the standard deviation of the profile derived roughness lengths. We invite the reviewer to check the Sicart et al. (2014 ) publication for further details on the procedure used to derive the roughness parameters and the associated errors.

**-Role of sublimation**

**Section 3.3 shows the formula used to calculate melt based on the SR50 measure-ments. Can you please add a comment to describe the role of sublimation, and why it is excluded from a mass loss calculation.**

When the latent heat fluxes are at their maximum (for the high TKE subset), we find mean *LE*=-10 W m$^{-2}$ (Table 3, former Table 2). This corresponds to a daily ablation of 0.3 mm w.e., if we consider this is sublimation and the rate is sustained during a whole day. That is clearly negligible. We added a sentence in the text to explain, p10 line 5: *"Ablation due to evaporation or sublimation at the surface was considered negligible: mean absolute latent heat fluxes remained below 10 Wm$^{-2}$ for the most turbulent subsets (Table 3) which corresponds to a daily ablation of only 0.3 mm w.e."*

**-Results**

**Currently the Results section combined comments related to both timing of events as well as classification by weather type. Please restructure to only focus on the weather patterns (as that is what is used in the Discussion section). Changing between time and pattern becomes confusing for the reader, as it becomes a lot of information to keep in mind.**

Since the classification by weather patterns doesn't always reliably work for describing turbulent conditions in the surface layer, we decided to focus on a characterization of the turbulence and turbulent fluxes using a TKE classification. Since the strong WP tend to select high TKE conditions and low WP, low TKE conditions, the TKE classification helps understanding the processes occurring in the WP-related classification. We emphasized this point as much as possible throughout the text. In the new version, this point is evidenced in the results section 4.1, then most of the results are presented in terms of the TKE classification, and we finally discuss how that impact on the results for the WP classification.

**-Results Section 4.1**

**Currently Section 4.1 is quite dense and difficult for the reader to pull apart. Please restructure in a way that is easier for a reader to follow. For example: "SF conditions are characterized by ____________. This was observed ______ (example). The impact on melt is ________."  In this way, the reader can focus (and retain) the most relevant information associated with each weather pattern (rather than interchanging between time periods and weather patterns).**

We significantly modified the text in this section 4.1 to take into account your comment together with the other reviewer comments. Meteorology and wind regimes (4.1) are now described

differently: first for weak forcing, then for strong forcing. The subsections in 4.1 have been removed. The whole text is now much shorter and clearer, the whole section 4.1 included 42 lines, it now contains only 21.

**Results Section 4.2.2**

**As the paper is directed at a general glaciology audience and not at a specialized micro-meteorology community, there needs to be a better introduction to spectral analysis within this section as a reminder. Perhaps the addition of one or two sentences describing why the analysis is necessary and what relevance it has to the theme ( ie that it will help to describe the turbulent boundary layer) is necessary to ease the reader into the section. Even though it is briefly described in Section 3.2, it would be beneficial to restate the information here. Also, the term 'Kansas curve' and 'Kaimal curve' is used interchangeably throughout the document – please standardize.**

We agree with the comment, and we added a sentence, which include reference to the methods sections, p13 line 9: *"The Fourier analysis of the 2006 EC data of the wind speed components and temperature were compared to the Kansas curves, to see if the surface layer was in equilibrium (Sect. 3.2)"* And modified the following sentence to account for this inclusion.

We standardized to "Kansas curve".

**-Discussion Section 5.3**

**If there are concerns regarding the comparison with the weather patterns, these should be quantified.**

Actually the concerns are limited, but the text wasn't very clear. The weather patterns do well at distinguishing typical wind conditions, and generally reflect differences between high and low TKE. Regarding the total SEB calculation, the WP classification is maybe less efficient. We exchanged the position of the sentences to reflect this. We added, in this section: *"can be useful to assess turbulent surface fluxes characteristics"* and *"It also provides a rough classification of the turbulence conditions in the glacier surface-layer, since SF and WF conditions are, in the glacier surface layer, likely associated with high and low TKE conditions, respectively. We show that the BA method compared differently with the EC method for different WP. More generally for SEB studies, we highlight some limitations."* We removed *"is only partly adapted for SEB studies"*.

**-Results-Discussion-Conclusions disconnect**

**In the current form, there appears to be a disconnect between the Results, Discussion and Conclusions sections. Often, results required to support statements made in the conclusions (eg: erratic discrepancies between results during certain conditions) are not supported in the Results section, and additionally, certain sections of the Results (eg. Meteorology and wind regimes) are not fully utilized in the Discussion/Conclusions sections. I would recommend to the authors to visualize the conclusions and include only the results necessary to address those points.**

The inclusion of table 3 with the correlations between the fluxes from the BA method and the EC method, the change of fig 8 to a scatter plot of the turbulent fluxes, and the overall revision of the text, now provide a much clearer and focused discussion. We removed the figure related to TKE and the related discussion, which was sufficiently framed and lacked supporting data. We didn't significantly modified the conclusion, but with the other modification they now better reflect the content of the results/discussion.

**Specific comments**

**P1 L3 Please clarify the time period in which each dataset was collected – from the description it seems that eddy covariance data and the profile data was collected during both the 2006 and 2009 study periods.**

Ok - done

**P2 L2 Change 'englacial' to 'glacierized' (or similar)**

Done

**P5 L5 Why was a 1-hour period used instead of the standard 30-min run?**

The choice was constrained by the parallel use of the outside meteorological station data for which sampling was set on 1-hour. Calculations made on a 30 min basis didn't change the relative contribution of the different turbulent fluxes to the SEB.

**P5 L19 Please explain how the data was 'analyzed and compared'.**

The weather pattern series has been kindly provided by "Electricite de France" (EDF), and we did not undergo this analysis ourselves. The paragraph provides a brief explanation of the whole method. For the specific step of analysis and comparison, each day, the shape of the observed geopotential height field over Europe is characterized by the geopotential height at 0h and 24h at the 700 hPa level and the 1000 hPa level for 110 grid points (a total of 440 points). This field is compared, in this 440 dimensions space, to the 8 geopotential height fields types which have been obtained by the statistical sorting of measured rain patterns in south-east France. The WP of the day is determined by finding the WP for which the geopotential height pattern is the nearest to the observed geopotential height field in the 440 dimension space, using the Teweles-Wobus score (Garavaglia et al., 2010).

Detailing this complex procedure is beyond the scope of this paper. We provide reference to the paper from Garavaglia et al. 2010. We modified the text to add a short description for the selection of the WP: *"Then, for any day (inside or outside the period used to characterize the decomposition, e.g. 1956-1996) the observed geopotential height field shape over Europe is characterized by the observed geopotential height at 0h and 24h, at the 700 hPa level and the 1000 hPa level for 110 grid points (a total of 440 points). This field is compared, in this 440 dimensions space, to the 8 geopotential height fields proposed by the rain-patterns analysis. The nearest of the 8 fields provides the WP."*

We added the following sentence at the end of the paragraph: *"Details of the procedure can be found in Garavaglia et al., 2010."*

**P6 L11 Change Table 3 to Table 2**

Done. Note that table 2 has been split in table 2 and table 3 and table 3 now contains more results: errors on mean turbulent fluxes, regression parameters between modelled and measured fluxes, as well as correlation coefficients.

**P9 L4 Please minimize the use of brackets. It makes the sentence confusing for the reader.**

Ok, removed.

**P12 L9 Change 'again a after' to 'again after'**

This detailed description of the melt behavior has been removed

**P13 L9 Please minimize the use of brackets. It makes the sentence confusing for the**

Done.

---

## Referee Report (RR1)

**Referee review for manuscript tc-2016-93-manuscript-version4, under review for The Cryosphere.**

"Surface-layer turbulence, energy-balance and links to atmospheric circulations over a mountain glacier in the French Alps" by Maxime Litt, Jean-Emmanuel Sicart, Delphine Six, Patrick Wagnon, and Warren D. Helgason.

**General Comments**

The revised manuscript is improved in many respects from the initial submission. The removal of some analyses and inclusion of others has improved the flow of the results. Many points have been clarified and issues addressed.

The authors have done well to refine the results and discussion within a more cohesive conceptual framework. However, more care needs to be taken in the discussion and conclusions to accurately reflect on whether the statements being made are well supported by data, or more speculative in nature. In their present form, these sections contain too much speculation, which undermine other more robust arguments that are made.

The use of both weather patterns and TKE to categorize turbulence data is novel, but at times this analysis becomes ambiguous. At the moment the two categories are used somewhat interchangeably, which is not strictly correct and leads to ambiguity. Further work is needed to clarify the use of TKE categories and weather pattern categories. Alongside this, there seems to be some discrepancies between the periods used to select data in some results (e.g. Table 3 vs Figure 8).

The addition of further analyses comparing turbulent heat fluxes measured by eddy-covariance and modelled using the bulk aerodynamic method (Figure 8) is useful. However, these new analyses have also highlighted the limitations in the datasets available, namely:

- Large uncertainties in measured melt that make it hard to show a significant improvement of one method or another when calculating melt using the SEB method. Other avenues need to be explored to illustrate where better flux calculations matter.
- A large scatter in the BA fluxes compared to the EC fluxes, particularly for latent heat: individual points that are divergent in sign (-100 vs +50 W m$^{-2}$) point to deficiencies in measured gradients of air temperature or humidity. This deserves further scrutiny as it severely limits the confidence in the latent and net turbulent heat flux data presented.
- The lack of concurrent flux and profile measurements limit the conclusions that can be drawn around the flux-profile relationship. Because the profile derived roughness lengths were calculated during a different season to the EC flux measurements, no unambiguous statements can be made about the performance of the BA method using these profile data.

The authors do not always seem aware of these limitations and make many inferences about the flux – profile relationship that are not supported. For instance – they interpret the underestimation of sensible heat fluxes by the bulk method in 2006 (using parameters determined in 2009) as confirmation that bulk method does not resolve a low-frequency contribution to the sensible heat flux, yet have no profile data to show this. It is well established that the roughness length for momentum can change by several orders of magnitude over the course of a season on a single glacier, so it may be equally likely that a profile determined roughness length was simply different in 2006 versus 2009, especially in the context of a large (4 orders of magnitude) scatter in the measured roughness length.

The introduction of an error analysis is encouraging, but this only takes into account random errors, while systematic errors are also likely and will affect the comparison of methods to a large degree. It is also well known that the BA method is extremely sensitive to the choice of roughness lengths, stability parameterization and surface temperature scheme used, but this is not assessed. A thorough comparison between EC and BA fluxes, taking into account of the full range of systematic and parametric uncertainty (roughness lengths, stability functions, treatment of surface temperature) needs to be made.

In short, the authors need to present a more careful analysis and discussion for the manuscript to be acceptable. This would address the issues surrounding the limitations of datasets, a more thorough treatment of errors and a consistent and clear treatment of weather pattern or TKE classification.

Critically, the authors also need to show some example of where the method used to calculate the turbulent fluxes impacts significantly on melt, as this is the fundamental premise of the paper (as outlined in the first two sentences of the abstract). As it stands, Figure 8 does little to confirm to readers that they should consider the effect of deviations of turbulent heat fluxes from theoretical predictions when calculating melt using the SEB method. Perhaps breaking the analyses further into a daily or multi-day periods with the same weather pattern may highlight these effects? Using multi-day periods would reduce uncertainty in the back-calculated melt energy by increasing the absolute magnitude of melting for the same uncertainty in the sonic ranger measurements.

Alternatively, the authors may wish to present an analysis of the utility of the weather patterns for diagnosing temporal variations in the SEB and include the turbulent flux calculations as part of the uncertainty. The paper does present a good summary of how large scale weather patterns are related to surface layer turbulence and corresponding characteristic deviations from MOST. This provides a good base from which to present and discuss the temporal variation of SEB components in more detail. This may prove to be a more useful analysis given the limitations to the datasets noted above.

A further general comment is that there is a general inconsistency in the use of abbreviations for various terms that at times makes the arguments hard to follow.

I would recommend the authors carefully assess what can and cannot stated confidently with the datasets available, revise the results accordingly, position the discussion within this context and conduct a more thorough internal review for clarity and consistency (of text and figures) before resubmission.

I have structured the detailed comments by first addressing the response of the authors to my earlier revision before making line comments on the revised text and tables in the new manuscript.

**Comments on the response of the authors to comments on the first manuscript.**

Note that, for brevity, comments are only shown where the initial comments were not fully addressed. Most comments were well addressed. The original comments are shown in bold type, author responses in blue type and the new comments as normal indented type.

**The attempt to link these fine scales of turbulence to surface melt, through the calculation of fluxes in using the bulk aerodynamic (BA) method, is ambitious but makes sense in the conceptual framework of the paper. However, the formal links between each scale of analysis are not always well made and this reduces the confidence in the interpretations made.**

> As discussed above, this remains to be a central issue with the manuscript. Several weaknesses in the comparison of the BA and EC methods and their effect on melt (outlined above and discussed in more detail in comments below) need to be resolved.

**The large scope also makes the paper somewhat disjointed.**

> The paper is now less disjointed and provides a more cohesive conceptual framework for the reader.

**There are clearly some interesting interactions occurring between the magnitude of net turbulent fluxes and the quantity of melt, which deserve further elucidating. The EC method gives the lowest melt of all methods, yet the has the same or higher net turbulent heat fluxes as the other methods. Why do we see a breakdown of theoretical predictions, yet no impact on modelled melt?**

Actually, our results show the net turbulent fluxes are higher in Strong Forcing than in Weak Forcing conditions. Focusing on H (LE is small or erratic due to large random errors) we show the BA method fluxes are lower than the EC method fluxes in Strong Forcing conditions. Considering the EC method is more reliable than the BA method, we consider the BA method underestimates fluxes in this cases. This effect is even more pronounced if you consider a subset of only large TKE: turbulent fluxes H are quite different with the BA and the EC (Table 3, new version). The results suggest that an effective roughness length (ze, introduced in former studies) can be used as a tuning parameter to correct the BA method for this underestimation.

But, since the total melt results from turbulent exchanges and radiative exchanges, we are not sure that we clearly understand this comment. Relating turbulent fluxes to melt cannot be done without considering the whole surface energy balance, and especially the radiative balance which plays a key role on melt. The total melt can be low if the radiative balance is low, even with strong net turbulent fluxes, and vice versa. That explains why we do not find a clear relationship between turbulent fluxes magnitudes and melt. Actually, the radiative balance is controlled by the albedo. In section 5.2, the first sentence states: "During both campaigns, the SEB was mainly controlled by large radiative fluxes, regardless of large-scale forcing, but the contribution of turbulent fluxes to the SEB was significant".

In section 5.2 which describes Fig. 9 (former version, now Fig. 8), we added: "Yet, changes in net turbulent fluxes resulting from the choice of calculation method remained too small in comparison with the radiative balance to yield significant differences in the SEB estimates." We hope the issue is clearer now.

> Perhaps I could have made myself clearer here, as the authors have not addressed my comment. My reference here was to Figure 9 in the original manuscript which detailed the melt predicted for each SEB method: 2006- strong forcing showed the EC method gave the lowest melt of all methods (which all have the same radiative fluxes). This seemed at odds with EC method having

larger fluxes than the BA method. Is this because of large *LE* that cancels out a larger value of *H*, or is this because of how EC fluxes were used to calculate melt?

On closer inspection I now see the magnitude of *H+LE* is different in the figure (20-30W m$^{-2}$) compared to that predicted for the same period in Table 3 (30-40 W m$^{-2}$). Please explain clearly how the estimates of SEB in Figure 8 are made. Are the same time periods used? How have poor quality EC data been treated (gap filled? Excluded?).

I also note that in the updated Figure 8, the values of *H+LE* for the EC method and the average melt rate have changed between revisions. Please also explain why these net turbulent fluxes are different in the figure and table and why these have changed between revisions.

Regarding the authors response to my comment here that – *"changes in net turbulent fluxes resulting from the choice of calculation method remained too small in comparison with the radiative balance to yield significant differences in the SEB estimates."* As presented this really does limit the usefulness of the analyses presented here to the glacier modelling community. Further analyses describing situations where it matters to get the fluxes right is needed.

**Further description of the processes occurring at the hourly and daily scale are needed to formally link uncertainties in the calculation method for turbulent heat fluxes to melt. Along with this, one of the key points introduced is the errors in the calculation of the turbulent heat flux, yet this is poorly addressed in later analysis. This would put the paper in a much stronger position to comment on the conditions in which the violation of current turbulent flux modelling methods will have an impact on melt.**

Regarding error calculation, we apply here methods to estimate the random errors resulting mainly from instrumental uncertainties that has been developed on Zongo glacier in Litt et al., (2015). We only briefly describe these methods herein, to avoid having a too long paper, but we could add more details if required by the reviewers. We encourage the reader to check the above mentioned paper where all the error calculation processes are described in detail. As presented in section 4.2.3, and discussed in section 5.1, page 18 lines 3-7 (new version), the main result here is that the random errors cannot explain the differences between the EC and the BA methods when TKE is high. We suggest the difference is due to the inability of the BA method to capture part of the flux in non-stationary conditions. We inserted the paragraph about errors into a new subsection (3.3), where we improved the method explanation. We also improved the interpretation of the results, especially for the turbulent fluxes, in relation with the turbulence characteristics, in section 5.1.

While the manuscript is improved, the it still does not present a thorough comparison between EC and BA fluxes, taking into account of the full range of systematic and parametric uncertainty (roughness lengths, stability functions, treatment of surface temperature), which are known to have a large impact (see for example Giesen et al, 2008).

**P3 ln18. Were any corrections for tilting of the radiometer measurements necessary given the high melt rates? If so, please detail the procedure used to correct for tilting of the mast, or if corrections were not performed please comment on the effect of tilt on the radiation values.**

Corrections were not performed, since tilting was not automatically measured. We assumed the potential correction on the radiometer remained low, since the sensor was rarely found to be out of alignment during each field visit (every 10 days), and it was levelled every time small deviations were noted.

> Please note in the text that corrections were not made. Also, the effect of even a few degrees' tilt on incoming shortwave radiation can be significant, so the assumption may not be justified. Please estimate the additional uncertainty introduced by the maximum estimated tilt angle and detail in the text.

**P4 ln5. Please explain why 1-hour runs were chosen over standard 30-minute runs?**
The choice was constrained by the parallel use of the off-glacier (on the moraine) meteorological station data for which sampling was set on 1-hour. Calculations of the fluxes have been done with 30 min based runs and did not changed the relative contribution of the fluxes to the SEB.

> Please note this in the text.

**P7 ln20. What runs were chosen for the analysis of roughness lengths, and how was stability taken into account? This is especially important given the low wind speed maximum observed. Also, why was the EC data not used to analyse roughness lengths?**
The complete procedure, developed and presented in Sicart et al., 2014, is based on an iterative fitting of profiles between wind and temperature as described in Andreas et al., 2002. The selection of runs was based on a set of criteria, including neutrality (based on Richardson-bulk parameter analysis), quality of the fits ($R_2 > 0.975$), absence of a katabatic wind-speed maximum, and some others. We agree this is not clear in the text, so we included a statement referring to the Sicart et al., 2014 paper more explicitly: *"The method for roughnesses determination was inspired from Andreas (2002) and developed for the tropical Zongo glacier. It is detailed in Sicart et al. (2014)"*
We had not included the roughness values calculated with the EC system since we thought using only one point above the ground for this calculation was not reliable enough. Anyway, the dynamic roughness length had been evaluated with the EC system, by inverting the log-linear wind speed profile relationship. The values were dispersed but we found a mean value of $z_0$ around 0.02 m and $z_t$ around $6.6 \times 10_{-6}$ m when derived from the EC system. We now mention that in the new manuscript (page 8, line 17 ):
*"We also evaluated the roughness lengths using the EC system and inverting equations 4 and 5, and selecting neutral runs. The median $z_0$ was 0.022 m, and the median $z_t$ $6.6 \times 10_{-6}$ m. We did not use these values to calculate fluxes through the BA method."*
Interestingly, the factor $(\ln(z/z_0)\ln(z/z_t))_{-1}$, which is the denominator in the bulk formulation of the fluxes (assuming neutrality), calculated with the EC derived roughness lengths or the effective roughness is roughly identical (0.0165 and 0.0167). This shows that effective roughness lengths values can be used to compensate for the BA method underestimation of the fluxes. We now mention this results in the discussion section p18 line 6:

*"This is supported by the values of the EC derived roughness lengths, considering they account for the additional mixing: the denominator $(ln(z/z_0) \ ln(z/z_t))_{-1}$ in the bulk formulation of the fluxes, calculated with the EC derived roughness lengths or the effective roughness length is roughly identical (0.0165 and 0.0167)."*

The additional data presented here serve to illustrate the shortcomings of the profile technique, rather than the bulk aerodynamic method per se. The deviations of the spectra from classic theoretical predictions indicate that flux-profile relationships established in idealised circumstances are not appropriate in this environment. Thus, to correctly characterise fluxes, the roughness length should be diagnosed from EC measurements, rather than profile measurements. The authors need to frame their arguments in this light.

**P9 ln10. Please justify the exclusion of heat from precipitation. This is only reasonable if the contribution of this flux can be shown to be negligible.**

This is a common assumption over alpine glaciers, with significant mass turnover. Also, heat advected by precipitation is generally negligible because the temperature difference between the rain and the ice is low, the rain intensity is small, and anyway rain is rarely observed. See Paterson (1994), and Oerlemans (2001). We added references in the text.

The exclusion of the heat advected from precipitation may be justified when considering total glacier-wide annual mass balance, but is not correct for shorter time periods, especially summer periods where convective rainfalls can add significantly to melt energy (Neale and Fitzharris, 1997). It is an oft used assumption, but is used for its convenience more than its robustness and should be stated as such. None of the references provide an analysis of the contribution (or lack thereof) of rain heat to the SEB, therefore can only be treated as reflecting the current practice, rather than established reality. Two of the papers (Six et al., 2009 and Oerlemans et al., 2009) simply cite the earlier text books, therefore should be removed. A more appropriate statement would read something like,

"The energy gains from precipitation were excluded from the analysis as they were assumed to contribute negligibly to the surface energy balance (Paterson and Cuffey, 1994; Oerlemans, 2001).

**Figure 9. These error bars seem unrealistically small – especially given the large spread in H from an uncertainty in surface temperature of +/- 1 K. Further justification of these errors is needed.**

Note that these are only errors on the mean turbulent fluxes. Since they are random errors, on average over all the available runs, the error is quite reduced, whereas it can be quite large for an individual measurement (see new figure 7). The error is calculated on the basis of an error of +/- 0.35 K on Ts (see Litt et al. 2015, and updated error methods section).

The effect of potential systematic biases on the average turbulent heat fluxes needs to be considered here, as they can be substantial and have a large impact on calculated fluxes – these will not cancel out when considering the uncertainty on the mean values. As noted earlier a more thorough treatment of systematic variations in roughness lengths, stability corrections, the treatment of surface temperature is

needed to establish a more realistic uncertainty on both the average and hourly turbulent heat fluxes. The use of a Monte Carlo method may be useful way to include this uncertainty.

**Further specific comments on text and tables in the revised manuscript**

P10 ln14 - change *"wet"* to "moist"

P11 ln1 - *"10 June"* do you mean 20 June? The period 10 June to 20 June includes other and strong forcing.

P11 ln8 - *"moderate air temperatures (5 C)"* – add "on-glacier"

P11 ln9 - *"generally cloudy conditions"* – this is not apparent in the figure. Please either clearly show this or modify the statement.

P12 ln2 - change *"inexistent"* to "non-existent"

P12 ln6 - *"However, some high and low TKE cases remained in the classification (Table 2)".* It is unclear what is being referred to here. Do you mean that after sub setting the 2006 data by TKE, some low (28%) and high (11%) cases remained in the SF and WF classes, respectively? If so, please state this.

P13 ln3 - *"-5 W m$^{-2}$"* – do you mean +5 W m$^{-2}$

P13 ln10, P14ln1, P15 ln3 – consistency in terms – *"horizontal", "longitudinal, "u".* Please choose one term and stick with it. Similar when introducing spectra and co-spectra ($S_u$, $Co_{uw}$).

P13 ln12 - Need to either explain the dependency on $z/L$, or remove the reference to it here.

P15 ln3 – change to *"…. affected the co-spectra of the momentum flux ($Co_{uw}$) (Figure 6d)"*

P15 ln17 – change *"low"* to "small"

P15 ln19 – *"…did not provide significantly different results"* – do you mean just for the average values of $LE$ here? The $BA_{eff}$ method gives an average value for $H$ that is significantly larger than the EC flux, taking into account the uncertainty. Also the correlation (r) between EC and BA fluxes is different between methods (range 0.31 to 0.67) and this would indicate that there may be statistically significant differences in the fit, not just the mean value, which is relevant here. Please make sure your statements are consistent with the data presented.

P15 ln21 – *"random errors were too low to explain such a discrepancy".* Further elaboration is needed on this point.

P15 ln23 – *"… as result of random errors on both methods…".* Figure 7 indicates that many points lie well outside the range expected by random errors alone for LE. Please modify the statement and discuss further.

P16 ln1 – *"…better correlation … were found with $BA_{eff}$."* – The r values in Table 3 are identical for these methods. Please revise.

P17 ln9 – please use consistent terms for each co-spectra and Kaimal…

P17 ln13 – please clarify that the peak in heat flux occurred at n= $10^{-2}$ in high TKE cases only.

P17 ln16 – *"this explains why the $BA_{pro}$ method systematically underestimated…"*. Unfortunately, this result is speculative, as concurrent profile and EC measurements are not available. Please modify the statement.

P18 ln3 – *"This suggests that the use in SEB models of an effective roughness length, larger than the profile-derived dynamic or thermal roughness lengths, respectively, in order to increase the turbulent fluxes so that the SEB matches the melt, is actually a way to compensate potential biases in the BA turbulent fluxes due to failure of the MOST when TKE is high.* The discussion needs to reflect on the use of an effective roughness length as a common practice. Also I note that the effective roughness length used here (0.001 m) is the same as the dynamic roughness length derived from the profile measurement. Please revise.

P18 ln10 – *"(Fig. 6a)"*

P18 ln16 – please pose as a hypothesis *"…wind speed maximum would be expected to oscillate too… This would explain…"*

P18 ln20 – *"As a result, fluxes from … were de-correlated"* Further analyses would be needed to show that poor correlation of EC and BA fluxes in low TKE conditions are a result of the hypothesized oscillations in wind speed maximum. The poorer correlation is to be expected given that the smaller absolute values of the fluxes are more effected by measurement uncertainty. Please provide additional analyses, or modify the statement.

P18 ln27 – *"…too small to …. yield significant differences in the calculated SEB."* Figure 7 seems to show that the SEB estimates are significantly different from each other, just not significantly different from the melt. Please modify this statement.

P19 ln6 – *"In these cases low-frequency oscillations…led to an underestimation of the turbulent heat fluxes by the BApro method"* – This statement is extremely speculative. No data are available to show what the actual turbulent heat fluxes were in 2009, while in 2006 no profile data are available to show the mean gradients. This may be occurring, but needs to be phrased as a working hypothesis.

P20 ln7 – *"we show that the BA method compared differently"* – Please point to the data that indicate this.

P20 ln 9 – *"for which temperature and cloud cover can be significantly different."* This is an important points and deserves further discussion.

P20 ln10 – *"cloud covered conditions"* – this was not shown in the data (see earlier comment). Please show more clearly the increase in cloud in SF conditions, or revise the statement.

P21 ln2 – *"The non-equilibrium of the surface layer let to…"* Again, this is speculative and needs to be phrased as hypothesis.

P21 ln3 – *"both methods"* please clarify which methods in the text.

P21 ln4 – *"A systematic underestimation… in magnitude than the EC fluxes"* Again this is speculation as the data cannot show that the differences were not simply due to a systematic changes or errors in the dynamic or temperature roughness lengths, stability correction functions, or surface temperature. Please present these as hypothesis, or provide further analysis to support the statements.

P21 ln8 – *"During the 2006 campaign, using turbulent fluxes from the BA or from the EC method did not provide significantly different results…. Hence, the turbulent fluxes from both methods were small and not significantly different on average.""* – This statement is confusing as the fluxes from $BA_{eff}$ and $BA_{pro}$ look to be significantly different to each other (Table 3 and Figure 8). Also later in the paragraph it is stated that *"The turbulent fluxes calculated with $BA_{pro}$ underestimated fluxes calculated with $BA_{eff}$"* – do you mean here that you interpret Figure 8 as showing significant differences, but only 2009? Please clarify which data indicate statistically significant relationships, and which data are used to illustrate or support the potential importance of processes during certain weather conditions (i.e. 2009 SF).

P21 ln13 *"the $BA_{pro}$ method could not account… surface layer…"* Again, this is speculation and needs revised.

Table 2.

- Please make sure the use of acronyms is consistent. Good quality runs -> GQR, SF, WF etc.
- It would be preferable to use the abbreviation TKE throughout rather than introduce the term e. Also low-TKE and high-TKE would be clearer terms to use to rather than $e < 1$ $m^2$ $s^{-2}$ etc.
- "2006-TKE" is confusing – perhaps use "TKE classes (2006 only)"

Table 3.

- What do the bold values in the last column indicate?
- Again please use consistent acronyms – $H$ ($BA_{pro}$) $H$ (EC) etc., rather than introducing new acronyms $H_{EC}$, $H_b$ etc.
- Same comments as Table 2 regarding TKE.
- Some measure of the spread of results is needed (RMSE or mean absolute error) to characterize uncertainty in the fit.
- The regression coefficients for situations with very low correlation are meaningless and should be removed.

Figure 3.

    Change *"(a-f)"* to "(a, f)" etc.
- Change *"light shaded"* to "blue shaded"?
- Change *"gray lines"* to "dark lines"

Figure 4.

- Please use AWS-G, AWS-M, SF and WF for consistency

Figure 6.

- Check colours in legend.

- Change *"Kaimal"* to "Kansas curves"

Figure 7.

- Figure 7 is a good addition to the manuscript, but it would be better to provide separate plots for H and LE and similar figures for both $BA_{pro}$ and $BA_{eff}$ (i.e 4 sub plots)
- Please use the abbreviations $BA_p$ and $BA_{eff}$ in the caption, rather the longer descriptions.
- Please clarify what the *"error values"* (+/- 1 standard deviations, 2 standard deviations?) are in the caption.
- Please check the dates used for the SF and WF conditions: 10 - 26 July, 2006 seems to include a period of SF? Also it is ambiguous what dates/conditions are actually used – are the blue and red points selected using a combination of TKE and WP? I think the analysis needs to choose one or the other to avoid ambiguously and overly selecting data to compare. For a robust comparison, it would be preferable to present all data greater than/ less than certain TKE limits for the full 2006 period.

Figure 8.

- Please use consistent terms $BA_{eff}$ etc
- Please clarify that *"mean measured melt"* here refers to the magnitude of melt energy back-calculated from surface height changes at the AWS, not the actual melt (which is measured is mm w.e. or similar).
- Why have the magnitudes of turbulent fluxes changed from the original manuscript? Have the analyses been updated? If so the changes that have been made should be carefully documented.

**References**

Giesen, R. H., van den Broeke, M. R., Oerlemans, J., and Andreassen, L. M.: Surface energy balance in the ablation zone of Midtdalsbreen, a glacier in southern Norway: Interannual variability and the effect of clouds, Journal of Geophysical Research, 113, 2008.

Neale, S. M. and Fitzharris, B. B.: Energy balance and synoptic climatology of a melting snowpack in the Southern Alps, New Zealand, International Journal of Climatology, 17, 1595-1609, 1997.

---

## Author Response (AR2)

**Referee review for manuscript tc-2016-93-manuscript-version4, under review for The Cryosphere.**

"Surface-layer turbulence, energy-balance and links to atmospheric circulations over a mountain glacier in the French Alps" by Maxime Litt, Jean-Emmanuel Sicart, Delphine Six, Patrick Wagnon, and Warren D. Helgason.

**General Comments**

The revised manuscript is improved in many respects from the initial submission. The removal of some analyses and inclusion of others has improved the flow of the results. Many points have been clarified and issues addressed.

The authors have done well to refine the results and discussion within a more cohesive conceptual framework. However, more care needs to be taken in the discussion and conclusions to accurately reflect on whether the statements being made are well supported by data, or more speculative in nature. In their present form, these sections contain too much speculation, which undermine other more robust arguments that are made.

We wish to sincerely thank the reviewer to have provided such a detailed review. We propose herein a revised version which takes into account all the new comments and provides more explanations when necessary. We changed considerably some parts of the selection procedures and some calculations, and we set-up a new analysis (including a Monte-Carlo approach for the turbulent fluxes, as suggested). In this final version we had to modify profoundly some parts of text, including the abstract and conclusion to correctly reflect the results of our modified and/or new analysis.

The use of both weather patterns and TKE to categorize turbulence data is novel, but at times this analysis becomes ambiguous. At the moment the two categories are used somewhat interchangeably, which is not strictly correct and leads to ambiguity. Further work is needed to clarify the use of TKE categories and weather pattern categories.

We agree that this point was either not clear or led to ambiguity. The main underlying reason for the use of two different categories is that the weather pattern classification works fine at defining wind regimes (katabatic/anabatic), but only partly, at sorting high/low TKE runs: We formerly introduced the high/low TKE classes since we observed, for weak weather patterns, more frequent low TKE cases. For strong weather patterns high TKE cases were the more frequent. We thought that characterizing the flow directly in terms of TKE was more reliable than only on weather patterns.

According to the reviewer's comment, we modified the classification scheme. We now first classify hourly data with regard to weak/strong weather pattern and study the general characteristics of these two subsets in terms of local wind regimes: strong or weak winds, presence of katabatic winds, etc… (Mostly using 2009 data)  Then, in order to study turbulence characteristics (only for the 2006 campaign during which we had EC data), we refine the selection procedure: we analyze only the weak forcing cases for which $TKE<1$ $m^2s^{-2}$, $H>5$ $Wm^{-2}$, wind-speed$>2$ $ms^{-1}$ , and for the strong forcing subset, we sub-select only the cases for which $TKE>2$ , $H>5$ $Wm^{-2}$, wind-speed$>2$ $m$ $s$-1. According to one of the reviewer's comments, we

identified discrepancies in the RH values from the different instruments, and our selection procedure includes now an RH criteria. This is new selection process is described in section 4.2.1.

These data are used for the updated Figures 7 and 8 (in the manuscript, Figure 6 and 7 herein).

Alongside this, there seems to be some discrepancies between the periods used to select data in some results (e.g. Table 3 vs Figure 8).

The periods used to select data were the same in Table 3 and Figure 8. The reason why the displayed values are slightly different was that in figure 8, the full boxes presented the total melt as calculated with the surface energy balance (SEB, equ. 8). The shaded part of the box was the mean sum of the radiation components. The remaining part was considered to be the contribution of the turbulent fluxes, which is not strictly correct since the overall melt calculated from the SEB is considered equal to 0 when SEB is negative. Thus, the white boxes heights actually represented $(SEB|_{seb>0\&Ts=0})$ - mean(SWin-SWout+LWin-LWout), which is not equal to the mean sum of H and LE presented in Table 3.

Figure 8 has been replaced by two figures (8 and 9), the first one presenting the mean hourly H+LE for the new classes (our new classification that considers strong forcing with high TKE and weak forcing with low TKE cases) and calculated with the different methods, the second presenting the melt cumulated over all the periods when continuous measurements of EC and bulk data were available. See the answer to the specific comments below.

The addition of further analyses comparing turbulent heat fluxes measured by eddy-covariance and modelled using the bulk aerodynamic method (Figure 8) is useful. However, these new analyses have also highlighted the limitations in the datasets available, namely:

- Large uncertainties in measured melt that make it hard to show a significant improvement of one method or another when calculating melt using the SEB method. Other avenues need to be explored to illustrate where better flux calculations matter.

We present now in Figure 9 melt calculated with the SEB method on periods for which there is no data interruption, and we compare that to the integrated melt observed over the related period, with errors derived from the Monte-Carlo approach. The presented data suggests that discrepancies appear in the calculations whether profile-derived roughnesses or effective roughnesses are used but an unambiguous statement cannot be made. We agree that in terms of total melt, the use of either method for turbulent fluxes or the other doesn't provide significant changes, but we do show that the total turbulent fluxes significantly change (Figure 8). This shows that in the case of high altitude or very dry sites, where the contribution of the turbulent fluxes to the overall energy balance can be more significant (include references here), the choice of the method might be important. We changed the abstract, introduction and conclusion to better reflect these results.

Regarding uncertainties on measured melt, we provided extreme error estimates, but our surface energy balance calculations actually do well at representing the melt as derived from the sonic height ranger, as shown in the Figure 9.

- A large scatter in the BA fluxes compared to the EC fluxes, particularly for latent heat: individual points that are divergent in sign (-100 vs +50 W m$^{-2}$) point to deficiencies in measured gradients of air temperature or humidity. This deserves further scrutiny as it severely limits the confidence in the latent and net turbulent heat flux data presented.

We agree that the scatter is large between the measured sonic fluxes and the evaluated profile fluxes. First of all, in the former figure 7 there was an issue in our data treatment - we were not comparing datasets at the same time – this was corrected. See the updated figure 7 attached, where we selected the data with as described above. Scatter remains, and we identify 2 main reasons for that:

1) Thanks to the comment by the reviewer we checked the humidity measurements. This do highlights issues: The comparison of the humidity measured by the AWS sensor and that obtained from the LICOR shows large dispersion, growing with humidity. When both instruments provided difference in RH larger than 10%, we removed the concerned runs.
2) Overestimation of the temperature by the AWS-G due to solar radiation contamination (Huwald et al., 2009), even though the shield was mechanically aspirated (also seen in Sicart et al., 2014).

Filtering for RH<60 % filters for outliers. Some scatter remains though, probably due to the second point above, but statistically, the tendency is shown.

The lack of concurrent flux and profile measurements limit the conclusions that can be drawn around the flux-profile relationship. Because the profile derived roughness lengths were calculated during a different season to the EC flux measurements, no unambiguous statements can be made about the performance of the BA method using these profile data.

We agree with the reviewer that in the case of different seasons with different surface states one cannot transfer roughness values from one period to another. We highlighted this in the text and have mitigated our conclusions according to this comment. Though, in our case, they are various arguments that justify our approach. We provide these in the answer to the specific comment below, which relates to the same issue.

The authors do not always seem aware of these limitations and make many inferences about the flux – profile relationship that are not supported. For instance – they interpret the underestimation of sensible heat fluxes by the bulk method in 2006 (using parameters determined in 2009) as confirmation that bulk method does not resolve a low-frequency contribution to the sensible heat flux, yet have no profile data to show this.

We agree, this might be a bit speculative. Though, the hypothesis is solid and since it is supported by other studies in stable layers. It has been shown , not only on glaciers (Andreas, 1987, Smeets et al., 1998, 1999) that when outer layer large scale turbulent eddies interact with the surface turbulent flow, the turbulence no longer scales with surface parameters (roughness, temperature) since turbulent transport is non-negligible. The presence of outer-layer interactions is supported by the shape of our spectra which compares, at low-frequency to results obtained in similar conditions in other studies (Hogstrom 2002, Hojstrup 1987, Smeets et al, 1998…). We introduced more clearly these concept in the script and method, to better illustrate our discourse, and we mitigate the discussion and our conclusions to take the reviewer comment in consideration.

It is well established that the roughness length for momentum can change by several orders of magnitude over the course of a season on a single glacier, so it may be equally likely that a profile determined roughness length was simply different in 2006 versus 2009, especially in the context of a large (4 orders of magnitude) scatter in the measured roughness length.

We agree with the reviewer on this point, but observations during the campaigns show that the surface was quite similar and that its evolution during each campaign cannot lead to such drastic changes in the roughness parameters. We provide herein the results of our roughness lengths (Figure 9) derivation from the profiles, and pictures (Figures 1-8) of the surface during each campaign. We clarify our discourse by defining clearly the effective roughness ($z_{eff}$) concept (new Equation 8).

We set $z_{eff}$ such as $\ln(z/z_0)\ln(z/z_{0t}) = \ln(z/z_{eff})\ln(z/z_{eff})$, $z$ = measurement height.

So we can calculate the first part of this equation using the profile derived $z_0$ and $z_{0t}$, and find an effective roughness and compare it to the $z_{eff}$ set in order for the bulk fluxes to match the eddy fluxes. Such **zeff** in our case is 0.001 m. Our measurements with profiles in 2009 show a predominant value of and $z_0$=0.001 and $z_{0t}$=0.00001 (even for the most chaotic surfaces), this leads to a $z_{eff}$ of about 0.0001 considering 2m measurements. The surface was quite similar (see Figure 1-4 herein) during both campaigns and it evolves through the campaign, but $z_{0t}$ derived from the profile never goes above $10^{-4}$ m – we think it doesn't make sense that the actual $z_{0t}$ in 2006 was 2 orders of magnitude higher than in 2009, whereas the surfaces remained similar.

[Figure]

**Figure 1: The surface state on installation, 10 July 2006.**

[Figure]

**Figure 2: The surface on 19 July 2006.**

[Figure]

**Figure 3: The surface on 2006, 01 august**

**Comment from field report, 2006 12 August:**

"Glacier clean and covered with 10 to 15 cm of snow"

[Figure]

**Figure 4: The surface on 22 aout 2006**

[Figure]

**Figure 5: 2009 installation on 12 June $z_0$ profiles = 0.001 $z_{0t}$ profiles =$10^{-4.5}$**

[Figure]

**Figure 5: 2009 surface on 21 July $z_0$ profiles =0.001 $z_{0t}$ profiles =$10^{-4.5}$**

[Figure]

**Figure 7: 2009 - 25 july $z_0$ profiles $=10^{-3}$ $z_{0t}$ profiles $= 10^{-4.5}$**

[Figure]

**Figure 8: 2009 – 4 August $_{z0}$ profiles $=10^{-3}$ $z_{0t}$ profiles $= 10^{-4}$**

We include herein a barplot of the $z_0$ and $z_{0t}$ derived with the profiles in 2009 (Figure 9). The mentioned dispersion of 4 orders of magnitude was the actual range of remaining $z_0$ values after selection but the standard deviation $d\ln(z_0)$ is equal to about 2.5. This is the value we use for random error calculations. So the way dispersion on the roughness might impact the fluxes is taken into account in the random error. We updated the discussion according to these findings.

[Figure]

**Figure 9: dispersion of the roughness measurements from the profiles in 2009**

The introduction of an error analysis is encouraging, but this only takes into account random errors, while systematic errors are also likely and will affect the comparison of methods to a large degree. It is also well known that the BA method is extremely sensitive to the choice of roughness lengths, stability parameterization and surface temperature scheme used, but this is not assessed. A thorough comparison between EC and BA fluxes, taking into account of the full range of systematic and parametric uncertainty (roughness lengths, stability functions, treatment of surface temperature) needs to be made.

The former error study included an error on the emissivity of the surface = 0.01 that led to an uncertainty of 0.35K on the surface temperature (though this was not described). But anyway we now do a Monte-Carlo approach on top of which we add a random error calculation on the final fluxes. We calculate the whole possible combinations of:

1)  Two Ts parameterization: blocking at 0 degrees – or one including a correction to include the effects of shortwave radiation on the longwave sensor: we remove 6% of the reflected longwave radiation.
2)  (emissivity of the surface) set once at  e=0.99 and once at  e=0.95
3)  $z_0$ fixed (0.001) but  $z_{0t}$=0.00001 or 0.001 ($z_{eff}$)
4)  different stability function parameterizations:
    log-linear (a=5 and a=7), Brutsaert,  Holtslag and de Bruin, Beljaars and Holtslag.
5)     Inclusion of a SW error calculation scheme (based on the observed difference between the calculated potential radiation for  non-tilted and tilted sensor)

On top, the resulting different turbulent fluxes are dispersed with random error calculation like in Litt et al. AMTD 2015. This now provide a broader analysis of the systematic errors, results are shown in figure 8 (new figure) which compares the mean H+LE bulk fluxes with the resulting

dispersion with those derived from the Eddy covariance measurements. This is derived for the two subsets described above: weak forcing with low TKE and strong forcing with high TKE, for which we remove the cases for which RH values from the EC and the Vaisala values differ by more than 10%. The actual dispersion is large, larger than the random error on the classical log-linear method. In weak conditions the bulk method as well as the EC method provide similar net fluxes and the dispersion is too large to show any clear difference. In strong forcing, the EC method shows slightly larger fluxes than the pro method, but this could partly be due to some errors. The effective roughness method can explain properly the EC fluxes.

In weak forcing, the minimum values are obtained from log-linear with a=5 stability functions, epsilon = 0.95, and blocking method for $T_s$. The maximum values are obtained with Beljaars and Holtslag, emissivity=0.99, blocking method for $T_s$. In strong forcing, the maximum values are found for the Beljaars and Holtslag functions, emissivity=0.99, blocking $T_s$ at zero method. Minimum values are found for emissivity = 0.95, blocking scheme for $T_s$, and log-linear a=7 when $z_{eff}$ is used and Holtslag and de Bruin when $z_{0t}$ profile is used.

In short, the authors need to present a more careful analysis and discussion for the manuscript to be acceptable. This would address the issues surrounding the limitations of datasets, a more thorough treatment of errors and a consistent and clear treatment of weather pattern or TKE classification.

We agree with the comment, we now present the limitations more clearly into the discussion and conclusions. Also, as mentioned above: We now classify data as weak/strong cases, and for the weak subset, we analyze only the weak forcing cases for which TKE<1, and for the strong subset, we analyse only the strong forcing cases for which TKE>2, so that our analysis is now unambiguous.

Critically, the authors also need to show some example of where the method used to calculate the turbulent fluxes impacts significantly on melt, as this is the fundamental premise of the paper (as outlined in the first two sentences of the abstract).

We do question the fact that choice of methods impact significantly the melt. In our dataset such a period of time where turbulent fluxes calculation method really impacts the ablation is lacking, still, if we take the period of strong forcing cases with high TKE we can see that the $z_0$ likely fails to capture the total turbulent fluxes (Figure 8). The error analysis on the melt (Figure 9 in the manuscript) shows that this result is not unequivocal and can be discussed, so the abstract-introduction-conclusions have been rewritten accordingly. The results show that situations where turbulent fluxes calculations should be taken care of are those for which very high winds and large sublimation can be observed. Such situations can be found on high altitude low latitude glaciers.

As it stands, Figure 8 does little to confirm to readers that they should consider the effect of deviations of turbulent heat fluxes from theoretical predictions when calculating melt using the SEB method.

True, we updated figure 8 by providing two figures (8 and 9). Figure 8 presents only the net turbulent fluxes with the results of the Monte Carlo approach, while figure 9 presents the melt

obtained using only sections of more than 6 hour for which data were continuous, as suggested in the next comment.

Perhaps breaking the analyses further into a daily or multiday periods with the same weather pattern may highlight these effects? Using multi-day periods would reduce uncertainty in the back-calculated melt energy by increasing the absolute magnitude of melting for the same uncertainty in the sonic ranger measurements.

See response to the comment above. We now calculate the melt-fluxes over period for which the data set is continuous. But still this doesn't provide better evidence.

Alternatively, the authors may wish to present an analysis of the utility of the weather patterns for diagnosing temporal variations in the SEB and include the turbulent flux calculations as part of the uncertainty. The paper does present a good summary of how large scale weather patterns are related to surface layer turbulence and corresponding characteristic deviations from MOST. This provides a good base from which to present and discuss the temporal variation of SEB components in more detail. This may prove to be a more useful analysis given the limitations to the datasets noted above.

Apart from the wind conditions, it is hard to find a consistent tendency in the other parameter with the strong/weak forcing classification (see figures 10, 11 and 12 of this document). On the other side, we can show that the participation of the net turbulent fluxes is more important in the case of strong forcing: We compute the RATIO=mean(H+LE)/mean(SWinc-SWout+LWinc-LWout) for weak/strong subset and different methods for H+LE. Results suggest that strong forcing favors the participation of H+LE to the total flux exchange. So if errors affect them this might have some importance in these case. The ratio is much smaller in 2009 weak cases than in 2006 weak cases probably because the LW incoming is much larger than in 2006 for these conditions.

<table>
<tr><td style="color:red">RATIO</td><td style="color:red">z0-zt</td><td style="color:red">ze</td><td style="color:red">ec</td></tr>
<tr><td style="color:red">2006 strong</td><td style="color:red">0.20</td><td style="color:red">0.27</td><td style="color:red">0.25</td></tr>
<tr><td style="color:red">2009 strong</td><td style="color:red">0.24</td><td style="color:red">0.34</td><td style="color:red">NA</td></tr>
<tr><td style="color:red">2006 weak</td><td style="color:red">0.18</td><td style="color:red">0.21</td><td style="color:red">0.20</td></tr>
<tr><td style="color:red">2009 weak</td><td style="color:red">0.05</td><td style="color:red">0.05</td><td style="color:red">NA</td></tr>
</table>

A further general comment is that there is a general inconsistency in the use of abbreviations for various terms that at times makes the arguments hard to follow.

We apologize for the inconvenience, and updated the draft with respect to that comment.

I would recommend the authors carefully assess what can and cannot stated confidently with the datasets available, revise the results accordingly, position the discussion within this context

and conduct a more thorough internal review for clarity and consistency (of text and figures) before resubmission.

According to this final comments, we reviewed more carefully the actual conclusions and discuss more clearly the issues to highlight the actual results of our study. The text has been clarified as demanded.

I have structured the detailed comments by first addressing the response of the authors to my earlier revision before making line comments on the revised text and tables in the new manuscript.

We provide below an answer to these comment to comments, in red.

[Figure]

**Figure 10: Mean daily evolution of the radiation components for weak and strong forcing conditions**

[Figure]

Figure 11: Mean daily variation of windspeed, temperature, and humidity, in 2006 and 2009.

[Figure]

**Figure 12: Mean daily variation of turbulent fluxes as measured with the bulk method and z0 derived from the profile method.**

**Comments on the response of the authors to comments on the first manuscript.**

Note that, for brevity, comments are only shown where the initial comments were not fully addressed. Most comments were well addressed. The original comments are shown in bold type, author responses in blue type and the new comments as normal indented type.

**The attempt to link these fine scales of turbulence to surface melt, through the calculation of fluxes in using the bulk aerodynamic (BA) method, is ambitious but makes sense in the conceptual framework of the paper. However, the formal links between each scale of analysis are not always well made and this reduces the confidence in the interpretations made.**

As discussed above, this remains to be a central issue with the manuscript. Several weaknesses in the comparison of the BA and EC methods and their effect on melt (outlined above and discussed in more detail in comments below) need to be resolved.

Our improved analysis shows that we cannot really conclude that the choice of the sonic or profile method really impacts on the melt calculated – only suggests that errors might exist and should be taken care of on the longer term, and that strong forcing cases might be the cases for which that imports more. We updated our analysis, and the discussion and conclusions.

**The large scope also makes the paper somewhat disjointed.**

The paper is now less disjointed and provides a more cohesive conceptual framework for the reader.

**There are clearly some interesting interactions occurring between the magnitude of net turbulent fluxes and the quantity of melt, which deserve further elucidating. The EC method gives the lowest melt of all methods, yet the has the same or higher net turbulent heat fluxes as the other methods. Why do we see a breakdown of theoretical predictions, yet no impact on modelled melt?**

Actually, our results show the net turbulent fluxes are higher in Strong Forcing than in Weak Forcing conditions. Focusing on H (LE is small or erratic due to large random errors) we show the BA method fluxes are lower than the EC method fluxes in Strong Forcing conditions. Considering the EC method is more reliable than the BA method, we consider the BA method underestimates fluxes in this cases. This effect is even more pronounced if you consider a subset of only large TKE: turbulent fluxes H are quite different with the BA and the EC (Table 3, new version). The results suggest that an effective roughness length (ze, introduced in former studies) can be used as a tuning parameter to correct the BA method for this underestimation.

But, since the total melt results from turbulent exchanges and radiative exchanges, we are not sure that we clearly understand this comment. Relating turbulent fluxes to melt cannot be done without considering the whole surface energy balance, and especially the radiative balance which plays a key role on melt. The total melt can be low if the radiative balance is low, even with strong net turbulent fluxes, and vice versa. That explains why we do not find a clear relationship between turbulent fluxes magnitudes and melt. Actually, the radiative balance is controlled by the albedo. In section 5.2, the first sentence states: "During both campaigns, the SEB was mainly controlled by large radiative fluxes, regardless of large-scale forcing, but the contribution of turbulent fluxes to the SEB was significant".

In section 5.2 which describes Fig. 9 (former version, now Fig. 8), we added: "Yet, changes in net turbulent fluxes resulting from the choice of calculation method remained too small in comparison with the radiative balance to yield significant differences in the SEB estimates." We hope the issue is clearer now.

Perhaps I could have made myself clearer here, as the authors have not addressed my comment. My reference here was to Figure 9 in the original manuscript which detailed the melt predicted for each SEB method: 2006- strong forcing showed the EC method gave the lowest melt of all methods (which all have the same radiative fluxes). This seemed at

odds with EC method having larger fluxes than the BA method. Is this because of large *LE* that cancels out a larger value of *H*, or is this because of how EC fluxes were used to calculate melt?

On closer inspection I now see the magnitude of *H+LE* is different in the figure (20-30W m$^{-2}$) compared to that predicted for the same period in Table 3 (30-40 W m$^{-2}$). Please explain clearly how the estimates of SEB in Figure 8 are made. Are the same time periods used? How have poor quality EC data been treated (gap filled? Excluded?).

The same period were used but results from Table 3 cannot be compared directly to the results from the SEB. In Table 3 we only showed mean turbulent fluxes for different periods. The sum of their mean is not equal to the mean of their sum. Furthermore in figure 8 (now 9) the SEB is sometimes found to be slightly negative whereas melt was observed, these values were forced to zero – this explains why the results of Table 3 and Figure 8 are slightly different. Poor quality EC data had been excluded. In the new melt comparison figure (i.e. figure 9) we used the new selection of both weather patterns together with TKE (see comments above).

I also note that in the updated Figure 8, the values of *H+LE* for the EC method and the average melt rate have changed between revisions. Please also explain why these net turbulent fluxes are different in the figure and table and why these have changed between revisions.

Values in the table are identical in between both version. In figure 8 values changed slightly because we wanted to include only full days of data. In first version we had included half days. We didn't change for Table 3.

Regarding the authors response to my comment here that – *"changes in net turbulent fluxes resulting from the choice of calculation method remained too small in comparison with the radiative balance to yield significant differences in the SEB estimates."* As presented this really does limit the usefulness of the analyses presented here to the glacier modelling community. Further analyses describing situations where it matters to get the fluxes right is needed.

Such results smight interest not only the glacier modelling community, but are also important for the atmospheric science community. Also, our results for Saint-Sorlin glacier in 2009 show that if the SF conditions are really frequent, differences between the flux calculation methods can grow, as well as the errors. Note that large sensible heat fluxes can observed on Scandinavian glacier, or very high sublimation at dry, high altitudes sites. In this latter cases better understanding of such situations can also have an impact on the mass balance calculations.

**Further description of the processes occurring at the hourly and daily scale are needed to formally link uncertainties in the calculation method for turbulent heat fluxes to melt. Along with this, one of the key points introduced is the errors in the calculation of the turbulent heat flux, yet this is poorly addressed in later analysis. This would put the**

**paper in a much stronger position to comment on the conditions in which the violation of current turbulent flux modelling methods will have an impact on melt.**

Regarding error calculation, we apply here methods to estimate the random errors resulting mainly from instrumental uncertainties that has been developed on Zongo glacier in Litt et al., (2015). We only briefly describe these methods herein, to avoid having a too long paper, but we could add more details if required by the reviewers. We encourage the reader to check the above mentioned paper where all the error calculation processes are described in detail. As presented in section 4.2.3, and discussed in section 5.1, page 18 lines 3-7 (new version), the main result here is that the random errors cannot explain the differences between the EC and the BA methods when TKE is high. We suggest the difference is due to the inability of the BA method to capture part of the flux in non-stationary conditions. We inserted the paragraph about errors into a new subsection (3.3), where we improved the method explanation. We also improved the interpretation of the results, especially for the turbulent fluxes, in relation with the turbulence characteristics, in section 5.1.

> While the manuscript is improved, the it still does not present a thorough comparison between EC and BA fluxes, taking into account of the full range of systematic and parametric uncertainty (roughness lengths, stability functions, treatment of surface temperature), which are known to have a large impact (see for example Giesen et al, 2008).

> We thank the reviewer for this useful reference, we now include a Monte Carlo approach for determining the impact of the choice of this different parameters on the overall melt. This is described in the new section 3.3.2.

**P3 ln18. Were any corrections for tilting of the radiometer measurements necessary given the high melt rates? If so, please detail the procedure used to correct for tilting of the mast, or if corrections were not performed please comment on the effect of tilt on the radiation values.**

Corrections were not performed, since tilting was not automatically measured. We assumed the potential correction on the radiometer remained low, since the sensor was rarely found to be out of alignment during each field visit (every 10 days), and it was levelled every time small deviations were noted.

> Please note in the text that corrections were not made. Also, the effect of even a few degrees' tilt on incoming shortwave radiation can be significant, so the assumption may not be justified. Please estimate the additional uncertainty introduced by the maximum estimated tilt angle and detail in the text.

> We now include into our analysis an error on the shortwave radiation. This is calculated by estimating the potential solar radiation (Pellicciotti et al., 2011) for untilted cases and for a maximum tilt of 10 degrees. The error is estimated as the relative difference between the two calculations.

**P4 ln5. Please explain why 1-hour runs were chosen over standard 30-minute runs?**

The choice was constrained by the parallel use of the off-glacier (on the moraine) meteorological station data for which sampling was set on 1-hour. Calculations of the fluxes have been done with 30 min based runs and did not changed the relative contribution of the fluxes to the SEB.

Please note this in the text.

Ok, this has been added.

**P7 ln20. What runs were chosen for the analysis of roughness lengths, and how was stability taken into account? This is especially important given the low wind speed maximum observed. Also, why was the EC data not used to analyse roughness lengths?**

The complete procedure, developed and presented in Sicart et al., 2014, is based on an iterative fitting of profiles between wind and temperature as described in Andreas et al., 2002. The selection of runs was based on a set of criteria, including neutrality (based on Richardson-bulk parameter analysis), quality of the fits ($R_2 > 0.975$), absence of a katabatic wind-speed maximum, and some others. We agree this is not clear in the text, so we included a statement referring to the Sicart et al., 2014 paper more explicitly: *"The method for roughnesses determination was inspired from Andreas (2002) and developed for the tropical Zongo glacier. It is detailed in Sicart et al. (2014)"*

We had not included the roughness values calculated with the EC system since we thought using only one point above the ground for this calculation was not reliable enough. Anyway, the dynamic roughness length had been evaluated with the EC system, by inverting the log-linear wind speed profile relationship. The values were dispersed but we found a mean value of $z_0$ around 0.02 m and $z_t$ around

$6.6 \times 10_{-6}$ m when derived from the EC system. We now mention that in the new manuscript (page 8, line 17 ):

*"We also evaluated the roughness lengths using the EC system and inverting equations 4 and 5, and selecting neutral runs. The median $z_0$ was 0.022 m, and the median $z_t$ $6.6 \times 10_{-6}$ m. We did not use these values to calculate fluxes through the BA method."*

Interestingly, the factor $(\ln(z/z_0)\ln(z/z_t)^{-1}$ which is the denominator in the bulk formulation of the fluxes (assuming neutrality), calculated with the EC derived roughness lengths or the effective roughness is roughly identical (0.0165 and 0.0167). This shows that effective roughness lengths values can be used to compensate for the BA method underestimation of the fluxes. We now mention this results in the discussion section p18 line 6:

*"This is supported by the values of the EC derived roughness lengths, considering they account for the additional mixing: the denominator $(\ln(z/z_0) \ln(z/z_t))_{-1}$ in the bulk formulation of the fluxes, calculated with the EC derived roughness lengths or the effective roughness length is roughly identical (0.0165 and 0.0167)."*

The additional data presented here serve to illustrate the shortcomings of the profile technique, rather than the bulk aerodynamic method per se. The deviations of the spectra from classic theoretical predictions indicate that flux-profile relationships established in idealised circumstances are not appropriate in this environment.

That is exactly what we meant. The log-linear profile is valid when turbulence in the surface layer only result from the friction at the surface, and thus the aerodynamic roughness parameter has a physical sense and can be linked to the geometrical roughness, when turbulence does not depend only on interaction with the surface, the

roughness parameter and the profile method fail. We added the comment in the text into the method section about turbulence characterization.

Thus, to correctly characterise fluxes, the roughness length should be diagnosed from EC measurements, rather than profile measurements. The authors need to frame their arguments in this light.

We indeed say that the profile fluxes correspond better to the EC fluxes if we take the EC-derived roughnesses to calculate the turbulent fluxes. Calculating the roughnesses from the EC is equivalent to calibrating the bulk aerodynamic method so that it fits to the observed fluxes.

**P9 ln10. Please justify the exclusion of heat from precipitation. This is only reasonable if the contribution of this flux can be shown to be negligible.** This is a common assumption over alpine glaciers, with significant mass turnover. Also, heat advected by precipitation is generally negligible because the temperature difference between the rain and the ice is low, the rain intensity is small, and anyway rain is rarely observed. See Paterson (1994), and Oerlemans (2001). We added references in the text.

The exclusion of the heat advected from precipitation may be justified when considering total glacier-wide annual mass balance, but is not correct for shorter time periods, especially summer periods where convective rainfalls can add significantly to melt energy (Neale and Fitzharris, 1997). It is an oft used assumption, but is used for its convenience more than its robustness and should be stated as such. None of the references provide an analysis of the contribution (or lack thereof) of rain heat to the SEB, therefore can only be treated as reflecting the current practice, rather than established reality. Two of the papers (Six et al., 2009 and Oerlemans et al., 2009) simply cite the earlier text books, therefore should be removed. A more appropriate statement would read something like,

"The energy gains from precipitation were excluded from the analysis as they were assumed to contribute negligibly to the surface energy balance (Paterson and Cuffey, 1994; Oerlemans, 2001).

We thank the reviewer for this clarification, we updated the statement as suggested. Note that we anyway remove cases of precipitation from the analysis since they correspond low quality measurements. This has been added in the text.

**Figure 9. These error bars seem unrealistically small – especially given the large spread in H from an uncertainty in surface temperature of +/- 1 K. Further justification of these errors is needed.** Note that these are only errors on the mean turbulent fluxes. Since they are random errors, on average over all the available runs, the error is quite reduced, whereas it can be quite large for an individual measurement (see new figure 7). The error is calculated on the basis of an error of +/- 0.35 K on Ts (see Litt et al. 2015, and updated error methods section).

The effect of potential systematic biases on the average turbulent heat fluxes needs to be considered here, as they can be substantial and have a large impact on calculated fluxes – these will not cancel out when considering the uncertainty on the mean values. As noted earlier a more thorough treatment of systematic variations in roughness lengths, stability corrections,

the treatment of surface temperature is needed to establish a more realistic uncertainty on both the average and hourly turbulent heat fluxes.  The use of a Monte Carlo method may be useful way to include this uncertainty.

Our newly introduced Monte-Carlo approach addresses this issue. We now take into account different stability functions, surface temperature parameterizations and different emissivities, and potential errors due to tilting of the radiometer.  Note that random errors on roughness lengths takes into account their large dispersion.

**Further specific comments on text and tables in the revised manuscript**

P10 ln14 - change *"wet"* to "moist"

Done.

P11 ln1 - *"10 June"* do you mean 20 June? The period 10 June to 20 June includes other and strong forcing.

Yes we meant 20 June, but there we are talking about weak forcing, which was observed between 20-25. This has been updated.

P11 ln8 - *"moderate air temperatures (5 C)"* – add "on-glacier"

Ok, this has been added.

P11 ln9 - *"generally cloudy conditions"* – this is not apparent in the figure. Please either clearly show this or modify the statement.

Ok, this has been removed.

P12 ln2 - change *"inexistent"* to "non-existent"

Ok, this has been changed.

P12 ln6 - *"However, some high and low TKE cases remained in the classification (Table 2)".* It is unclear what is being referred to here. Do you mean that after sub setting the 2006 data by TKE, some low (28%) and high (11%) cases remained in the SF and WF classes, respectively? If so, please state this.

Yes. We state that within the weak forcing runs 11% of the runs show TKE>2 and within strong forcing runs 28% of runs show TKE<1, this was changed to be clearer.

P13 ln3 - *"-5 W m$^{-2}$"* – do you mean +5 W m$^{-2}$

Yes, this was corrected.

P13 ln10, P14ln1, P15 ln3 – consistency in terms – *"horizontal", "longitudinal, "u".* Please choose one term and stick with it. Similar when introducing spectra and co-spectra ($S_u$, $Co_{uw}$).

Ok, we changed it to horizontal. Actually horizontal components refer to u (longitudinal) and v (lateral) components. We thus had to remove some "*lateral*" terms. These were simply noted as horizontal. The components symbols "u" and "v" are now used to differentiate the longitudinal and lateral cases.

P13 ln12 - Need to either explain the dependency on *z/L*, or remove the reference to it here.

Surface layer stability must control the position of the maximum spectra and cospectra by shifting it towards higher frequencies when increasing. We removed the statement to lighten the text.

P15 ln3 – change to *".... affected the co-spectra of the momentum flux ($Co_{uw}$) (Figure 6d)"*

We changed to: "*affected the momentum flux as shown by the $Co_{uw}$ cospectra (Figure 6d)"*, we hope the reviewer is ok.

P15 ln17 – change *"low"* to "small"

Done. Note that we now refer to weak forcing cases with low TKE.

P15 ln19 – *"…did not provide significantly different results"* – do you mean just for the average values of *LE* here? The $BA_{eff}$ method gives an average value for *H* that is significantly larger than the EC flux, taking into account the uncertainty. Also the correlation (r) between EC and BA fluxes is different between methods (range 0.31 to 0.67) and this would indicate that there may be statistically significant differences in the fit, not just the mean value, which is relevant here. Please make sure your statements are consistent with the data presented.

We agree with the reviewer. This point as changed considerably, since we modified the selection procedure before studying the fluxes. We invite the reviewer to look at the new figure 8. With the complete Monte-Carlo approach, one cannot differentiate between the mean net fluxes from each method in the weal forcing, low TKE subsets. Regarding the issue with the statistical significance, we agree with the reviewer, but the classification has been changed and the results updated.

P15 ln21 – *"random errors were too low to explain such a discrepancy".* Further elaboration is needed on this point.

We are not sure to understand what the reviewer actually asks through this comment. We actually meant that the random error calculation on the turbulent fluxes could not explain the difference observed between the profile and the sonic method's net turbulent fluxes. Actually this is not still the case with the new classification and with the dispersion resulting from the Monte-Carlo approach (Figure 8 in the manuscript).

P15 ln23 – *"… as result of random errors on both methods…"*. Figure 7 indicates that many points lie well outside the range expected by random errors alone for LE. Please modify the statement and discuss further.

Figure 7 has been removed and this has been changed.

P16 ln1 – *"…better correlation … were found with $BA_{eff}$."* – The r values in Table 3 are identical for these methods. Please revise.

There was a mistake: referring to SF conditions, we wanted to refer to high TKE conditions. We admit that presentation of data was confusing. This sentence is not anymore relevant in term of the new classification. It was changed according to the new results.

P17 ln9 – please use consistent terms for each co-spectra and Kaimal…

We are not sure to understand the comment. Here we meant cospectra of u **with** w, that was changed. Otherwise all "*Kansas curve*" occurrences were replaced by "*Kaimal curve*".

P17 ln13 – please clarify that the peak in heat flux occurred at n= $10^{-2}$ in high TKE cases only.

The beginning of this paragraph was slightly changed to answer the query.

P17 ln16 – *"this explains why the $BA_{pro}$ method systematically underestimated…"*. Unfortunately, this result is speculative, as concurrent profile and EC measurements are not available. Please modify the statement.

We updated the discussion in order to reflect this. We know present the statement as one of the possible explanations for the underestimations. We also add as an hypothese that the roughness length were simply different.

P18 ln3 – *"This suggests that the use in SEB models of an effective roughness length, larger than the profile-derived dynamic or thermal roughness lengths, respectively, in order to increase the turbulent fluxes so that the SEB matches the melt, is actually a way to compensate potential biases in the BA turbulent fluxes due to failure of the MOST when TKE is high.* The discussion needs to reflect on the use of an effective roughness length as a common practice.

We introduced a better description of the effective roughness in the methods.

Also I note that the effective roughness length used here (0.001 m) is the same as the dynamic roughness length derived from the profile measurement. Please revise.

Yes, but this effective roughness is not the same as the thermal roughness length derived from the profiles, it is two orders of magnitude larger. In the methods we describe now more clearly the roughness length issues.

P18 ln10 – *"(Fig. 6a)"*

Ok, this has been added.

P18 ln16 – please pose as a hypothesis *"…wind speed maximum would be expected to oscillate too… This would explain…"*

The text has been changed to reflect the fact that these remain hypotheses.

P18 ln20 – *"As a result, fluxes from … were de-correlated"* Further analyses would be needed to show that poor correlation of EC and BA fluxes in low TKE conditions are a result of the hypothesized oscillations in wind speed maximum. The poorer correlation is to be expected given that the smaller absolute values of the fluxes are more effected by measurement uncertainty. Please provide additional analyses, or modify the statement.

We agree with the reviewer comment. The text has been updated.

P18 ln27 – *"…too small to …. yield significant differences in the calculated SEB."* Figure 7 seems to show that the SEB estimates are significantly different from each other, just not significantly different from the melt. Please modify this statement.

New figures (8 and 9), with new classification provides different results. The text here has been significantly modified.

P19 ln6 – *"In these cases low-frequency oscillations…led to an underestimation of the turbulent heat fluxes by the BApro method"* – This statement is extremely speculative. No data are available to show what the actual turbulent heat fluxes were in 2009, while in 2006 no profile data are available to show the mean gradients. This may be occurring, but needs to be phrased as a working hypothesis.

We agree with the comment, the text was significantly modified, and all these statements are now presented as hypothesis.

P20 ln7 – *"we show that the BA method compared differently"* – Please point to the data that indicate

this.

Done

P20 ln 9 – *"for which temperature and cloud cover can be significantly different."* This is an important points and deserves further discussion.

P20 ln10 – *"cloud covered conditions"* – this was not shown in the data (see earlier comment). Please show more clearly the increase in cloud in SF conditions, or revise the statement.

Regarding these two comments, we agree this is important but actually this was not really evident (as shown by Figures 11 and 12 of this document). Our classification provides a distinction between high and low wind speed but not really for the other meteorological variables (different behavior between years). Another WP classification

would be needed to more accurately represent the changes in the SEB. We updated the text in this section accordingly.

P21 ln2 – *"The non-equilibrium of the surface layer let to…"* Again, this is speculative and needs to be phrased as hypothesis.

See response to comments above: the text was significantly modified, and all these statements are now presented as hypothesis.

P21 ln3 – *"both methods"* please clarify which methods in the text.

Done

P21 ln4 – *"A systematic underestimation… in magnitude than the EC fluxes"* Again this is speculation as the data cannot show that the differences were not simply due to a systematic changes or errors in the dynamic or temperature roughness lengths, stability correction functions, or surface temperature. Please present these as hypothesis, or provide further analysis to support the statements.

This statement is actually reinforced with the data presented in the new figure 8. We kept it, but the discussion and conclusion are now more prudent on the explanations of this behavior.

P21 ln8 – *"During the 2006 campaign, using turbulent fluxes from the BA or from the EC method did not provide significantly different results…. Hence, the turbulent fluxes from both methods were small and not significantly different on average.""* – This statement is confusing as the fluxes from $BA_{eff}$ and $BA_{pro}$ look to be significantly different to each other (Table 3 and Figure 8). Also later in the paragraph it is stated that *"The turbulent fluxes calculated with $BA_{pro}$ underestimated fluxes calculated with $BA_{eff}$"* – do you mean here that you interpret Figure 8 as showing significant differences, but only 2009? Please clarify which data indicate statistically significant relationships, and which data are used to illustrate or support the potential importance of processes during certain weather conditions (i.e. 2009 SF).

We have updated the text since the Figure 8 has changed (now Figure 8 and 9). In this paragraph we refer to Figure 9 (before to figure 8). We invite the reviewer to check the updated text in order to understand the new results. We included reference to Figure 9.

P21 ln13 *"the $BA_{pro}$ method could not account… surface layer…"* Again, this is speculation and needs revised.

See response to comments above: the text was significantly modified, and all these statements are now presented as hypothesis.

Table 2.

- Please make sure the use of acronyms is consistent. Good quality runs -> GQR, SF, WF etc.
- It would be preferable to use the abbreviation TKE throughout rather than introduce the term e.

That has been changed throughout the text.

Also low-TKE and high-TKE would be clearer terms to use to rather than e < 1 m² s⁻² etc.
– "2006-TKE" is confusing – perhaps use "TKE classes (2006 only)"

We hope the editing enables to clarify these issues

Table 3.

- What do the bold values in the last column indicate?
  They indicated the higher values of the regression coefficients. This has been removed.
- Again please use consistent acronyms – $H$ (BA$_{pro}$) $H$ (EC) etc., rather than introducing new acronyms $H_{EC}$, $H_b$ etc.
  Done
- Same comments as Table 2 regarding TKE.
  Done
- **Some measure of the spread of results is needed (RMSE or mean absolute error) to characterize uncertainty in the fit.**
  Table 3 now includes the RMSE. The values support the results of Fig. 8.
- **The regression coefficients for situations with very low correlation are meaningless and should be removed.**

  We believe this still provides some information

Figure 3.

Change *"(a-f)"* to "(a, f)" etc.
- Change *"light shaded"* to "blue shaded"?
- Change *"gray lines"* to "dark lines"

  Done

**Figure 4.**

- Please use AWS-G, AWS-M, SF and WF for consistency

  Done

**Figure 6.**

- Check colours in legend.
  We are not sure to understand the comment.
- Change *"Kaimal"* to "Kansas curves"

  We decided to keep Kaimal, as we homogenized all the manuscript by replacing "Kansas curve" with "Kaimal" curve.

**Figure 7.**

- Figure 7 is a good addition to the manuscript, but it would be better to provide separate plots for H and LE and similar figures for both $BA_{pro}$ and $BA_{eff}$ (i.e 4 sub plots)

  Done we have now 8 subplots since we separate for H and LE too.

- Please use the abbreviations $BA_p$ and $BA_{eff}$ in the caption, rather the longer descriptions.

  Ok

- Please clarify what the *"error values"* (+/- 1 standard deviations, 2 standard deviations?) are in the caption.

  This is not relevant anymore with the new figure.

- Please check the dates used for the SF and WF conditions: 10 - 26 July, 2006 seems to include a period of SF? Also it is ambiguous what dates/conditions are actually used – are the blue and red points selected using a combination of TKE and WP? I think the analysis needs to choose one or the other to avoid ambiguously and overly selecting data to compare. For a robust comparison, it would be preferable to present all data greater than/ less than certain TKE limits for the full 2006 period.

  We now consistently present data out of our new selection procedure, for all figures.

**Figure 8.**

- Please use consistent terms $BA_{eff}$ etc

  Ok, done.

- Please clarify that *"mean measured melt"* here refers to the magnitude of melt energy back calculated from surface height changes at the AWS, not the actual melt (which is measured is mm w.e. or similar).

  Since the figure has been changed, the issue is no more relevant. We know present the actual melt.

- Why have the magnitudes of turbulent fluxes changed from the original manuscript? Have the analyses been updated? If so the changes that have been made should be carefully documented.

  We don't understand why the reviewer states the magnitude of the turbulent fluxes has changed in this figure. They are the same in both manuscripts.

**References**

Giesen, R. H., van den Broeke, M. R., Oerlemans, J., and Andreassen, L. M.: Surface energy balance in the ablation zone of Midtdalsbreen, a glacier in southern Norway: Interannual variability and the effect of clouds, Journal of Geophysical Research, 113, 2008.

Neale, S. M. and Fitzharris, B. B.: Energy balance and synoptic climatology of a melting snowpack in the Southern Alps, New Zealand, International Journal of Climatology, 17, 1595-1609, 1997.

We added reference to Giesen et al., since it inspired our Monte-Carlo approach. The second reference was not included since it was not directly used, even though interesting.